# UTSD: UNIFIED TIME SERIES DIFFUSION MODEL

## ABSTRACT

Transformer-based architectures have achieved unprecedented success in time series analysis. However, facing the challenge of across-domain modeling, existing studies utilize statistical prior as prompt engineering fails under the huge distribution shift among various domains. In this paper, a Unified Time Series Diffusion (UTSD) model is established for the first time to model the multi-domain probability distribution, utilizing the powerful probability distribution modeling ability of Diffusion. Unlike the autoregressive models that capture the conditional probabilities of the prediction horizon to the historical sequence, we use a diffusion denoising process to model the mixture distribution of the cross-domain data and generate the prediction sequence for the target domain directly utilizing conditional sampling. The proposed UTSD contains three pivotal designs: (1) The condition network captures the multi-scale fluctuation patterns from the observation sequence, which are utilized as context representations to guide the denoising network to generate the prediction sequence; (2) Adapter-based fine-tuning strategy, the multi-domain universal representation learned in the pretraining stage is utilized for downstream tasks in target domains; (3) The diffusion and denoising process on the actual sequence space, combined with the improved classifier free guidance as the conditional generation strategy, greatly improves the stability and accuracy of the downstream task. We conduct extensive experiments on mainstream benchmarks, and the pre-trained UTSD outperforms existing foundation models on all data domains, exhibiting superior zero-shot generalization ability. After training from scratch, UTSD achieves comparable performance against domain-specific proprietary models. In particular, UTSD shows stable and reliable time series generation, and the empirical results validate the potential of UTSD as a time series foundational model. The source codes of UTSD are publicly available on `https://anonymous.4open.science/r/UTSD-1BFF`.

## 1 INTRODUCTION

Time Series (TS) data widely exist in many real-world fields (Bengio et al., 2015; Sezer et al., 2019; Fan et al., 2023), such as power (Wang et al., 2022), weather (Schultz et al., 2021), transportation (Thissen et al., 2003), finance (Chi & Chi, 2022), etc. The wide application of time series analysis makes it of vital research significance to many practical fields. Empirical practice illustrates that time series data from different domains perform shifted statistical properties (Wang et al., 2023; Yuan & Qiao, 2024), such as period, frequency, data distribution, number of features, and fluctuation patterns, which poses a critical challenge to the generalizability and robustness of time series analysis.

With the continuous development of deep learning, models based on DNN (Zeng et al., 2023; Yi et al., 2023), RNN (Shi et al., 2015), CNN (Wu et al., 2023; Wang et al., 2023) and Transformer (Wu et al., 2021; Nie et al., 2023), have made remarkable achievements in many tasks of time series analysis. With the success of generative pre-trained diffusion models in the vision domain (Esser & Kulal, 2024; Peebles & Xie, 2022; Liu et al., 2024b), diffusion-based time series forecasting has also shown promising results. Early efforts like TimeGrad (Rasul et al., 2021) employed RNNs to capture temporal patterns, thereby predicting future series in an autoregressive fashion. Furthermore, CSDI (Tashiro et al., 2021) and TimeDiff (Shen & Kwok, 2023) predict all time points simultaneously to mitigate the issue of error accumulation in long-term series forecasting. However, current methods often focus on training domain-specific models tailored to individual datasets, limiting their ability to generalize well to a variety of new, unseen time series domains.

Figure 1: Illustration of the proposed UTSD architecture. **(a)** In the diffusion process, the input original sequence $X_0$ is progressively noised until degenerating into the gaussian noise $X_T$. **(b)** In the context learning phase, the mixed different domain sequences are utilized as input to the UTSD . The condition net captures cross-domain temporal fluctuation patterns as conditional variables to guide the generation process. **(c)** In the denoising phase, the model accepts representations from multiple domains to reconstruct the fusion distribution from gaussian distribution. Forecasting the actual sequence by iterative denoising process.

The success of unified Large Language Models (LLMs) (Touvron et al., 2023) has inspired the development of a unified time series model. Trained on time series data from various domains, the unified model aims to achieve strong generalization capabilities and robustness to deliver satisfactory zero-shot inference performance on previously unseen domains. Previous attempts to develop unified time series models can be divided into two main categories: the *LLM-based approach* and the *multi-domain generalization approach*. The LLM-based approach leverages the alignment of time series modalities with natural language processing (NLP), utilizing a pretrained LLM, potentially with further fine-tuning, to enhance generalization capabilities. OneFitsAll (Zhou et al., 2023) fine-tunes a subset of the weights from pre-trained LLMs on specific time series dataset and customizes distinct output layers for various downstream tasks. To mitigate the cross-modality challenges encountered during fine-tuning, TimeLLM (Jin et al., 2023) employs mathematical and statistical information as part of its prompt engineering strategy to refine the LLM. However, the LLM-based approach requires fine-tuning model weights for each individual time series domain, and the inherent differences between NLP and time series modalities can result in concept drift and misalignment of representation dimensions (Yang et al., 2024). On the other hand, the multi-domain generalization approaches (Woo et al., 2024; Goswami et al., 2024) aim to train a broadly applicable model from scratch using data from multiple time series domains. UniTime (Liu et al., 2024a), Timer (Liu et al., 2024c), and Moirai (Woo et al., 2024) have focused on designing a generic architecture and training from scratch on comprehensive datasets with several temporal domain characteristics. Existing multi-domain generalization methods (e.g., Timer, etc.) rely on autoregressive mechanisms to establish connections between observed and predicted sequences. However, these methods are prone to error accumulation in long-sequence predictions and often face challenges with domain confusion. In contrast, models such as UniTime and Moirai attempt to directly learn the projection from the historical horizon to the future. Nevertheless, different domains exhibit varying data characteristics, making it nearly impossible to design a shared encoder capable of effectively handling time series from domains with distinct semantics. This limitation significantly restricts the zero-shot and cross-domain generalization capabilities of such models.

This paper establishes the unified time series diffusion model for the first time. Taking advantage of the diffusion model's excellent capability to model probability distributions, our approach directly produces diverse and high-quality forecasts by modelling a fusion probability distributions over multiple time series domains without establishing any inter-series projections. Further, the contextual information embedded in the observation sequence is captured as a conditional variable that guides the process of reconstructing the forecast results from gaussian noise, enhancing the stability and accuracy. Due to the excellent cross-domain generalisation capability and robustness of the diffusion approach, without any fine-tuning strategy, the pre-trained model exhibits better performance than the existing LLM-based methodologies on the all benchmarks. Besides, taking advantage of the excellent probability distribution modelling capability of the diffusion approach, this paper models the multidomain fusion distribution directly from the integrated data domain and generates diversified high-quality forecast results directly, thus avoiding the inferior of the autoregressive paradigm with respect to the cumulative error and predictive coherence.

Figure 1 shows the architecture of UTSD , which contains three pivotal novel designs: the innovative condition-denoising architecture, the execution of the reverse noise reduction process in the actual

sequence space, and the conditional generation strategy based on the classifier-free guidance. First, the condition-denoising architecture is designed, which contains both Condition Net and Denoising Net components. (a) In the context learning stage, observation sequences from different domains are fused together as inputs, from which the condition net captures a multi-scale representation of fluctuation patterns as context to guide the conditional generation at different levels. For example, shallow conditional variables will guide the trend part of the generated sequence, and deep representations will guide the multi-periodic patterns of the generated sequence. This ensures that the conditional information of the input data can be fully utilized to generate high-quality sequence samples. (b) In the denoising stage, the contextual information embedded in the observation sequence is captured as a conditional variable that guides the process of reconstructing the forecast results from gaussian noise, enhancing stability and accuracy. Another novel design is that, UTSD models in the actual sequence space instead of the latent space. Since the inverse denoising process often goes through a large number of iterative denoising (Li et al., 2024), each iteration of the denoising stage causes an accumulation of errors in the latent space, which are further amplified during the alignment of the latent space to the actual sequence space. Therefore, this paper proposes to perform the diffusion and denoising processes in the actual sequence space. While ensuring low time overhead, iterative denoising directly in the original sequence space can alleviate the dithering problem and improve the prediction accuracy. Furthermore, we propose the improved classifier free guidance as the conditional generation strategy, which ensures that UTSD has sufficiently strong generalization ability. Finally, we also design the efficient fine-tuning module Transfer-Adapter, which can generate high-quality sequence samples in a specific domain by fine-tuning only 5% of the parameters while retaining the fluctuation information learned by pre-training.

The key contributions of our work are as follows.

- We propose a novel condition-denoising architecture, an actual-space diffusion and denoising procedure, and a classifier-free guidance conditional generation strategy, which improvements greatly improve the prediction accuracy of the diffusion model.

- This paper establishes the time series foundation model based on a diffusion model for the first time. Due to the excellent cross-domain generalization capability, the model shows the potential to become an entirely new paradigm in time series.

- The proposed UTSD achieves SOTA results. Extensive experiments validate the effectiveness of UTSD, with overall performance improvements of **19.6%** and **21.2%** compared to existing foundation and proprietary baselines.

## 2 PRELIMINARIES

In the long-term forecasting task, $X^0_{-L+1:0} \in \mathbb{R}^{d \times L}$ and $X^0_{1:H} \in \mathbb{R}^{d \times H}$ are utilized to represent the observed series and the future series, respectively, where $d$ denotes the number of channels of the multivariate time series, $L$ and $H$ denote the lookback window and forecast horizon.

### 2.1 DIFFUSION MODEL

The Denoising Diffusion Probabilistic Models (DDPM) (Ho et al., 2020) consists of two processes, the forward diffusion and the reverse denoising processes, see appendix for details.

**Forward Process.** In the forward process, A set of real time series samples $X^0_{1:H} \sim q(X)$ are gradually noised until they degenerate into gaussian distribution $X^T_{1:H} \sim N(0, \mathrm{I})$. The complete diffusion process is regarded as the Markov chain, and the diffusion process at time step $t \in [1, T]$ is represented as $q\left(X^t_{1:H} \mid X^{t-1}_{1:H}\right) = N\left(X^t_{1:H}; \sqrt{1 - \beta_t} X^{t-1}_{1:H}, \beta_t \mathrm{I}\right)$, where $\beta_t \in (0, 1)$ is the diffusion coefficient and $T$ is the length of the Markov chain. The diffusion result $X^t_{1:H}$ corresponding to any number of steps $t$ can be directly computed from $X^0_{1:H}$ via the formula $X^t_{1:H} = \sqrt{\prod_{i=1}^t (1 - \beta_i)} \cdot X^0_{1:H} + \sqrt{1 - \prod_{i=1}^t (1 - \beta_i)} \cdot \varepsilon, \varepsilon \sim N(0, \mathrm{I})$.

**Reverse Process.** In the reverse process, the deep model is utilized to progressively denoise from gaussian distribution. The denoising process at time step $t$ is represented as $p_\theta\left(X^{t-1}_{1:H} \mid X^t_{1:H}, c\right)$, where $c$ represents the condition variable calculated from the observation sequence, $\mu_\theta(X^t_{1:H}, t)$ represents the denoising model established at the diffusion timestep $t$, $\theta$ represents the model parameters, and $\sigma_t$ serves as a hyperparameter. Following the design in DDPMs, we calculate the mean squared error between the $\mu_\theta(X^t_{1:H}, t)$ and the mean $\mu\left(X^t_{1:H}, X^0_{1:H}\right)$ of the posterior distribution $q\left(X^{t-1}_{1:H} \mid X^0_{1:H}, X^t_{1:H}\right)$ as the loss function $L\left(X^0_{1:H}\right) = \sum_{t=1}^T E_{q\left(X^t_{1:H} \mid X^0_{1:H}\right)} \| \mu\left(X^t_{1:H}, X^0_{1:H}\right) - \mu_\theta\left(X^t_{1:H}, t\right) \|^2$.

## 2.2 Classifier-Free Guidance for Condition Time Series Generation

To improve the capability of diffusion models for forecasting based on the conditional context of observed sequences, a potential classifer-free guidance mechanism is introduced to establish guided diffusion models. The context representation captured by the condition net from the observation sequence is denoted as the condition variable $c$, so that the goal of the reverse denoising process can be described as $p_\theta\left(X_{1:H}^{0:T} \mid c\right) = p_\theta\left(X_{1:H}^{T}\right) \Pi_{t=1}^{T} p_\theta\left(X_{1:H}^{t-1} \mid X_{1:H}^{t}, c\right)$, Where $X_{1:H}^{T} \sim N(0, \mathrm{I})$ represents the initial state obtained by sampling from gaussian distribution. Furthermore, according to the Bayesian formula (Ho, 2022), we have:

$$p_\theta\left(X_{1:H}^{t-1} \mid X_{1:H}^{t}, c\right) \cdot p_\theta\left(c \mid X_{1:H}^{t}\right) = p_\theta\left(X_{1:H}^{t-1} \mid X_{1:H}^{t}\right) \cdot p_\theta\left(c \mid X_{1:H}^{t-1}, X_{1:H}^{t}\right). \tag{1}$$

To obtain the probability score function to control the condition generation process, we run gradient update on $X_{1:H}^{t-1}$ and the score function as follows:

$$\nabla_{X_{1:H}^{t-1}} \log p_\theta\left(X_{1:H}^{t-1} \mid X_{1:H}^{t}, c\right) = \nabla_{X_{1:H}^{t-1}} \log p_\theta\left(X_{1:H}^{t-1} \mid X_{1:H}^{t}\right) + \nabla_{X_{1:H}^{t-1}} \log p_\theta\left(c \mid X_{1:H}^{t-1}, X_{1:H}^{t}\right). \tag{2}$$

Where $\nabla_{X_{1:H}^{t-1}} \log p_\theta\left(X_{1:H}^{t-1} \mid X_{1:H}^{t}\right)$ and $\nabla_{X_{1:H}^{t-1}} \log p_\theta\left(c \mid X_{1:H}^{t-1}, X_{1:H}^{t}\right)$ represent the gradients of the pretrained denoising model and classifier, respectively, which are used to approximate the sampling result $\log p_\theta\left(X_{1:H}^{t-1} \mid X_{1:H}^{t}, c\right)$ expressing the posterior distribution. This strategy of intervening based on a classifier to make the generated results more consistent with the user's intention is called classifier guidance. The classification guidance mechanism is widely used to achieve conditional generation in existing time series research. However, there are some inherent drawbacks. Firstly, the classifier may ignore many important details in the input sequence, thus providing incomplete conditional signals. Second, the gradient calculated by the classifier for $X_{1:H}^{t-1}$ may point in any direction, which leads to instability in conditional generation.

During training and inference, we build a high-quality conditional probabilistic diffusion model based on the classifier-free guidance strategy. First of all, by the Bayesian formula we have $\log p_\theta\left(c \mid X_{1:H}^{t-1}, X_{1:H}^{t}\right) = \log p_\theta\left(X_{1:H}^{t-1} \mid X_{1:H}^{t}, c\right) + \log p_\theta\left(c \mid X_{1:H}^{t}\right) - \log p_\theta\left(X_{1:H}^{t-1} \mid X_{1:H}^{t}\right)$. Then we run gradient update on $X_{1:H}^{t-1}$ and get $\nabla_{X_{1:H}^{t-1}} \log p_\theta\left(c \mid X_{1:H}^{t-1}, X_{1:H}^{t}\right) = \nabla_{X_{1:H}^{t-1}} \log p_\theta\left(X_{1:H}^{t-1} \mid X_{1:H}^{t}, c\right) - \nabla_{X_{1:H}^{t-1}} \log p_\theta\left(X_{1:H}^{t-1} \mid X_{1:H}^{t}\right)$. Subsequently, this equation is substituted into the equation 2 , classifier-free guidance is formulated as:

$$\begin{aligned} \nabla_{X_{1:H}^{t-1}} \log p_\theta\left(X_{1:H}^{t-1} \mid X_{1:H}^{t}, c\right) = & \nabla_{X_{1:H}^{t-1}} \log p_\theta\left(X_{1:H}^{t-1} \mid X_{1:H}^{t}\right) \\ & + \tau\left(\nabla_{X_{1:H}^{t-1}} \log p_\theta\left(X_{1:H}^{t-1} \mid X_{1:H}^{t}, c\right) - \nabla_{X_{1:H}^{t-1}} \log p_\theta\left(X_{1:H}^{t-1} \mid X_{1:H}^{t}\right)\right). \end{aligned} \tag{3}$$

Where $\log p_\theta(X_{1:H}^{t-1} \mid X_{1:H}^{t}, c)$, $\log p_\theta(X_{1:H}^{t-1} \mid X_{1:H}^{t})$ , and $\log p_\theta(X_{1:H}^{t-1} \mid X_{1:H}^{t}, c)$ represents the final output, unconditional output, conditional output of the denoising net, respectively. In the implementation, condition net first accepts the observation sequence as input and subsequently outputs the captured multi-scale representation as the observation sequence context. Then, two identical initial samples are sampled from the Gaussian noise, and two output sequences are generated by denoising net iterative denoising with the observation sequence context and zero vector as conditional variables, respectively, which are conditional- and unconditional- output. Finally, we calculate the final output of the model based on the weight specified by user.

# 3 UTSD Architecture

The establishment of unified diffusion model in time series faces the challenge of learning the fusion distribution from multiple data domains, while the long-term forecasting requires the model to learn the enough temporal information from the observation series of the specified domain as a condition context to generate forecast series that comply with the distribution of the domain. Existing condition diffusion models (Tashiro et al., 2021; Li et al., 2024; Yuan & Qiao, 2024) often use simple neural network layer to capture the conditional variable on single scale. However, sequences from several domains often have different multi-scale latent representations (Shabani et al., 2022), such as sampling rate, periodic frequency characteristics, multi-periodic patterns (Ma et al., 2024), etc. Due to the difficulty for the model to learn enough fluctuation pattern information from the lookback window and the randomness of the initial state, diffusion model demonstrates low accuracy in the prediction task (Alcaraz & Strodthoff, 2022). UTSD contains three pivotal novel designs: the innovative condition-denoising architecture, the execution of the reverse noise reduction process in the actual sequence space, and the conditional generation strategy based on the classifer-free guidance.

## 3.1 Input Observation Instance

Based on the channel independent design, the input observation sequence $X_{-L+1:0}^{0} \in \mathbb{R}^{B \times d \times L}$ is first processed as $X_{-L+1:0}^{0}{}' \in \mathbb{R}^{B \cdot d \times L}$, where $B$ denotes the batch size, and then following the patching instance strategy with no padding and no overlap, each univariate sequence of length $L$ is rerepresented as $X_{-L+1:0}^{0}{}'' \in \mathbb{R}^{B \cdot d \times P_L \times P_d}$, where $P_d$ is a hyperparameter representing the dimension of tokens, $P_L$ is the number of tokens contained in a single sequence, and the product of $P_d$ and $P_L$ is equal to $L$. Subsequently,

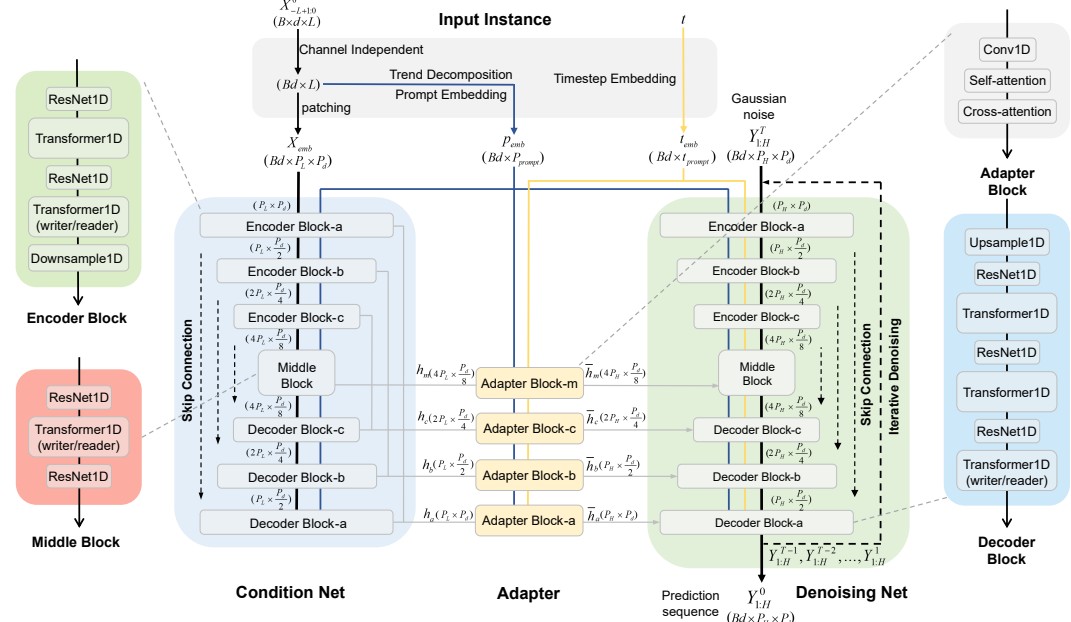

Figure 2: The overall framework of UTSD . Specifically, the observation sequence $X^0_{-L+1:0}$ and the diffusion timestep $t$ are processed by the input instance module to obtain the lookback embedding, trend-prompt embedding, and time embedding, which serve as inputs to the condition-denoising net and the adapter. The condition net captures the multi-scale representations $h_{a,b,c,m}$ and the adapter transforms those into the context variables $\overline{h}_{a,b,c,m}$ which utilized to guide conditional generation process for the prediction task.

observation tokens $X_{emb} \in \mathbb{R}^{B \cdot d \times P_L \times P_d}$ are computed from $X^{0}_{-L+1:0}'' \in \mathbb{R}^{B \cdot d \times P_L \times P_d}$ by the Embedding Layer based on 1D convolution.

In addition to the observation sequence, UTSD accepts two input data. Firstly, the trend part of the historical sequence is considered to be very critical information for the forecasting performance, so this paper proposes to use the embedding $p_{emb}$ of the trend part obtained by decoupling as prompt vector. Besides, since the reverse denoising process requires the diffusion timestep $t$ as guidance information. This paper proposes to compute the embedding of $t$ via a time encoder. See the appendix for details of the embedding layer.

## 3.2 CONDITION-DENOISING STRUCTURE

Firstly, the architecture composition of the conditional learning module and the denoising generation module are introduced. For convenience, the term Block is utilized to refer to a set of consecutive neural network, such as Encoder Block, Middle Block, Decoder Block, and Adapter Block, etc., which are reused as important components in building the Unet structure. As shown in Figure 2, condition net and denoising net both follow the encoder-decoder design, where the encoder and decoder are composed of Encoder Block-a,b,c and Decoder Block-a,b,c, respectively. Besides, the Middle Block is designed to connect the encoder and decoder. Unet consists of seven Blocks, where Encoder Block-a and Decoder Block-a have the same feature dimension $P_d/2$ and skip connections are established between the two blocks, similarly, the other blocks have dimensions $P_d/4$ and $P_d/8$, respectively.

In the training and inference of UTSD , the reverse process is divided into a context learning stage and a denosing generation stage, corresponding to condition-denoising net, respectively.

**Context Learning Stage.** In the context learning stage, condition net accepts observation tokens $X_{emb}$ as input, and Decoder Block-a at the end does not output any result. Specifically, the deep representation included in the observation sequence undergoes 4 consecutive Blocks (Middle Block and Decoder Block) to obtain the multi-scale historical fluctuation pattern $\{h_{m,a,b,c}\}$, as shown in Figure 2. This set of tempotal representations passes through Adapter Blocks with the same structure to obtain a set of condition variables $\{\overline{h}_{m,a,b,c}\}$. This is passed as a condition context to several Blocks in denoising net, thus guiding the condition generation process.

**Denoising Generation Stage.** In the denoising generation stage, $Y^T_{1:H} \in \mathbb{R}^{B \cdot d \times P_H \times P_d}$ obtained by sampling from gaussian distribution $N(0, \mathbf{I})$ is utilized as the initial input, where $P_H = H/P_d$. In each round of denoising iteration, the denoising net accepts the diffusion timestep $t$ and $Y^t_{1:H}$ as input, and its output is utilized to calculate the sample $Y^{t-1}_{1:H}$ for the next round of denoising iteration. After $T$ rounds of denoising process, $Y^0_{1:H} \in \mathbb{R}^{B \cdot d \times P_H \times P_d}$ undergoes the flatten and channel independent to obtain the prediction result

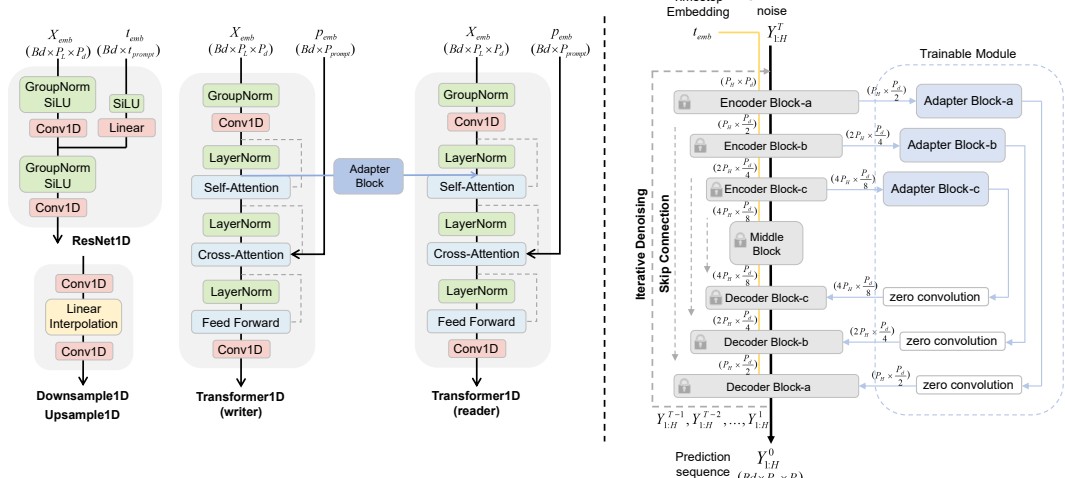

Figure 3: $(a)$: Illustration of two submodules, where Transformer1D contains two versions that are utilized to establish condition-denoising net, respectively. $(b)$: The unconditional generation paradigm of UTSD can be divided into two phases, the multi-domain pre-training phase and downstream data domain fine-tuning phase.

$Y_{1:H}^0 \in \mathbb{R}^{B \times d \times H}$. In particular, during inference, condition net only needs to learn a set $\{\overline{h}_{m,a,b,c}\}$ from the observation sequence, and in the subsequent all $T$ rounds of iterations, denoising net reuses this set of historical pattern representations to predict noise from samples with different timesteps $t$.

## 3.3 BLOCKS IMPLEMENTATION

All Blocks included in condition-denoising net are built from two smaller modules, ResNet1D Module and Transformer1D (writer/reader) Module. The ResNet1D module accepts the embedding $t_{emb}$ of diffusion timesteps $t$ (only in the denoising generation stage) and the latent representation $X_{emb}$ as input data, which contains two 1d convolutions. After the first convolution layer, the latent representation $X_{emb}$ is added to the timestep embedding $t_{emb}$ through the linear layer, and the output is obtained by second convolution layer (the normalization layer, activation layer, etc., are ignored in the description). The Transformer1D module has two versions, writer and reader, which constitute Blocks in condition and denoising net, respectively. Transformer1D-writer accepts the latent representation $X_{emb} \in \mathbb{R}^{B \cdot d \times P_L \times P_d}$ as input, where self-attention mechanism is utilized to capture dependencies between global patches in the lookback window. Transformer1D-reader accepts the condition variable $\{\overline{h}_{m,a,b,c}\}$, noised sample $Y_{1:H}^t \in \mathbb{R}^{B \cdot d \times P_H \times P_d}$, and the trend part embedding $p_{emb}$ (as prompt) as input, where self-attention concatenates the context into the key-value vector, thereby using historical fluctuation patterns as contextual information to guide the denoising process. In the subsequent cross-attention mechanism, the historical trend information contained in $p_{emb}$ is used to model the long-term trend of future sequences. The proposed condition-denoising net establishes the direct connection between the generated sequence space and the observed sequence through Transformer1D-writer,reader, ensuring that UTSD has strong generalization capabilities to address the challenge of cross-domain probability distribution modeling. In addition, each Encoder Block has Downsample1D as the end layer and each Decoder Block has Upsample1D as the first layer, as shown in Figure 2. The implementation details of ResNet1D, Transformer1D, Downsample1D, and Upsample1D are given in the Figure 3.

## 3.4 TRANSFER-ADAPTER MODULE

For pretrained models, directly fine-tuning with limited data at full weight or continuing training leads to catastrophic forgetting, mode collapse, and overfitting (Hu et al., 2022; Ruiz et al., 2023). Existing time series models avoid forgetting by freezing the original model weights and adding a small number of new parameters (Zhou et al., 2023), or low-rank adaptation prevents catastrophic forgetting by learning parameter shifts of a low-rank matrix (Jin et al., 2023). However, in order to deal with cross-domain challenges, it is necessary to design fine-tuning strategies that can adapt to the diffusion and denoising process.

UTSD's hybrid architecture supports naturally efficient fine-tuning through the 'plug-and-play' Adapter. Specifically, pre-training to obtain a Condition-Denoising Net with a large number of weights is completely frozen, and only a small number of weights in the Adapter component need to be optimised. The effectiveness of the fine-tuning strategy can be intuitively explained by the fact that the pre-trained Condition Net is responsible for capturing generic fluctuation patterns from observed sequences as conditional information, the Denoising Net is required to reconstruct sequence samples from noise in the target domain based on specific fluctuation patterns, and the fine-tuning-enabled Adapter is used to connect the unified representation space with the proprietary representation space.

Table 1: **Experimental Tasks.** We validate the proposed UTSD on four forecasting tasks.

| Forecasting Scenarios | Descriptions | Baselines | Metrics |
|---|---|---|---|
| Trained across-domain | Deep model trained on multi-domain fusion dataset and subsequently tested on each small-scale dataset | UniTime, GPT4TS, Moirai | MSE, MAE |
| Trained from scratch | Deep model trained from scratch and tested on each small-scale dataset | TimeLLM, LLM4TS, GPT4TS, PatchTST, TimesNet, DLinear, FEDformer, Autoformer, Informer | MSE, MAE |
| Zero-shot | Deep model trained on dataset-A and subsequently tested on other dataset-B | TimeLLM, LLMTime, GPT4TS, PatchTST, TimesNet, DLinear, Autoformer | MSE, MAE |
| Probability | Probabilistic model trained on each small-scale dataset and demonstratd with repeated sampling on the test set | DiffusionTS, LDT, CSDI, TimeGrad | MSE-a,t,m,l, Stability |

In additional, In Adapter, the innovative **$1 \times 1$ Conv1D** is designed to align the number of tokens in the observation space and the forecasting space. Adapter utilizes the attention mechanism to capture the dependencies between all tokens, which establishes a connection between context learning and noise reduction reconstruction. These hybrid structures ensure that UTSD has the flexibility to generate high-quality prediction sequences of arbitrary length. Adapter is a bridge between the conditional and denoising networks, supporting flexible input or output of prediction sequences of arbitrary time steps.

## 4 EXPERIMENTS

For different forecasting scenarios, four task paradigms are designed, which include **across-domain** pretraining, training from **scratch**, **zero-shot** learning and **probability** distribution modeling, as shown in Table 1.

To evaluate the performance of the proposed method, we extensively experiment with several popular real-world datasets, including: **ETT-h1,h2,m1,m2** (Zhou et al., 2021), **Exchange** (Lai et al., 2017), **Weather** (Wetterstation, 2015), **Electricity** (Trindade, 2015) and **Traffic** (PeMS, 2015). All forecasting scenarios have the same settings: lookback window $L = 336$, forecast horizon $H = \{96, 192, 336, 720\}$. In particular, all results shown in the paper are calculated based on single sampling, which demonstrates the satisfactory generation stability of UTSD as a probabilistic model. Table 1 shows the details of various forecasting scenarios, where all the utilized baselines include the following four components: **(1)** LLM-based model: GPT4TS (Zhou et al., 2023), LLMTime (Gruver et al., 2024), TimeLLM (Jin et al., 2023); **(2)** Unified time series model: UniTime (Liu et al., 2024a), Moirai (Woo et al., 2024); **(3)** Deep model: PatchTST (Nie et al., 2023), TimesNet (Wu et al., 2023), DLinear (Zeng et al., 2023), FEDformer (Zhou et al., 2022), Autoformer (Wu et al., 2021) and Informer (Zhou et al., 2021); **(4)** Diffusion model: DiffusionTS (Yuan & Qiao, 2024), LDT (Li et al., 2024), CSDI (Tashiro et al., 2021) and TimeGrad (Rasul et al., 2021). See the Appendix section for details on the Benchmark, the experimental environment, and the training hyperparameters.

### 4.1 ACROSS-DOMAIN AND SCRATCH FORECASTING

Across-domain prediction is defined as pre-training UTSD on a mixed dataset for learning pivotal information from multiple domains and subsequent inference on a specified dataset. Regarding the establishment of the mixed dataset and the multi-domain pre-training, we follow the experimental setup of UniTime (Liu et al., 2024a), which is described in the Appendix section E.1. The left part of Table 2 shows the results of cross-domain pre-training, which validates the ability of the proposed method to model multi-domain probability distributions. Compared to SOTA time series foundation models, UTSD achieves an overall better performance than them. Specifically, the average MSE of the proposed UTSD is reduced by **14.2%**, **20.1%** and **27.6%** compared to the existing Moirai, UniTime and GPT4TS, respectively, which demonstrates the potential of UTSD as a unified temporal spreading model.

The right part of Table 2 demonstrates the results of training from scratch on each particular dataset, the average MSE is reduced by **17.9%**, **18.6%** and **22.4%** compared to the existing TimeLLM, LLM4TS and GPT4TS, which indicates that the proposed method can fully utilize a small amount of data for efficient training. (TimeLLM et al. utilize huge corpus (more than 15,000,000 million timesteps) in pre-train, UTSD is only pre-trained on a mixed dataset (only 27.5 million timesteps)) Table 2 demonstrates the overall performance of the proposed method. Overall, the scratch UTSD achieves comparable performance to SOTA model TimeLLM, and excitingly, the cross-domain UTSD achieves overall better results than the existing foundation model.

### 4.2 ZERO-SHOT FORECASTING

Zero-shot forecasting is defined as first training the model on data domain A and subsequently forecasting on other "never seen" data domains. Specifically, Table 3 shows the results of the long-time forecasting task under zero-shot setting. An encouraging result is that UTSD shows strong generalization ability on the zero-shot scenarios. The potential advantages include, the first being that UTSD serves as a thorough foundation model on time series, whereas LLM-based models generally face cross-modality challenges. Another advantage is the utilization of unique probability distribution modeling rather than regression modeling, which ensures that UTSD can learn pivotal information from multiple domains and efficiently migrate it to never-before-seen target domains.

Table 2: Comparison of the performance from diverse prediction lengths on **Across-domain and Scratch Forecasting**. We boldface the best performance on two scenarios, respectively.

| Method | | Models Trained Across Datasets | | | | | | | | Models Trained From Scratch | | | | | | | | | | | | | | |
|---|---|---|---|---|---|---|---|---|---|---|---|---|---|---|---|---|---|---|---|---|---|---|---|---|---|
| | | Ours | | Moirai | | UniTime | | GPT4TS | | Ours | | TimeLLM | | LLM4TS | | GPT4TS | | PatchTST | | TimesNet | | DLinear | | FEDformer | |
| | | MSE | MAE | MSE | MAE | MSE | MAE | MSE | MAE | MSE | MAE | MSE | MAE | MSE | MAE | MSE | MAE | MSE | MAE | MSE | MAE | MSE | MAE | MSE | MAE |
| **ETTm1** | 96 | 0.357 | 0.378 | 0.404 | 0.383 | **0.322** | **0.363** | 0.509 | 0.463 | 0.299 | **0.333** | **0.272** | 0.334 | 0.285 | 0.343 | 0.292 | 0.346 | 0.344 | 0.373 | 0.338 | 0.375 | 0.345 | 0.372 | 0.379 | 0.419 |
| | 192 | **0.354** | **0.386** | 0.435 | 0.402 | 0.366 | 0.387 | 0.537 | 0.476 | **0.304** | 0.358 | 0.310 | **0.358** | 0.324 | 0.366 | 0.332 | 0.372 | 0.367 | 0.386 | 0.374 | 0.387 | 0.380 | 0.389 | 0.426 | 0.441 |
| | 336 | **0.363** | **0.388** | 0.462 | 0.416 | 0.398 | 0.407 | 0.564 | 0.488 | **0.312** | **0.365** | 0.352 | 0.384 | 0.353 | 0.385 | 0.366 | 0.394 | 0.392 | 0.407 | 0.410 | 0.411 | 0.413 | 0.413 | 0.445 | 0.459 |
| | 720 | **0.370** | **0.403** | 0.490 | 0.437 | 0.454 | 0.440 | 0.592 | 0.504 | **0.317** | **0.368** | 0.383 | 0.411 | 0.408 | 0.419 | 0.417 | 0.421 | 0.464 | 0.442 | 0.478 | 0.450 | 0.474 | 0.453 | 0.543 | 0.490 |
| | Avg | **0.361** | **0.389** | 0.448 | 0.410 | 0.385 | 0.399 | 0.551 | 0.483 | **0.308** | **0.356** | 0.329 | 0.372 | 0.343 | 0.378 | 0.352 | 0.383 | 0.392 | 0.402 | 0.400 | 0.406 | 0.403 | 0.407 | 0.448 | 0.452 |
| **ETTm2** | 96 | 0.195 | 0.289 | 0.205 | 0.282 | **0.183** | **0.266** | 0.229 | 0.304 | 0.191 | 0.284 | **0.161** | **0.253** | 0.165 | 0.254 | 0.173 | 0.262 | 0.177 | 0.260 | 0.187 | 0.267 | 0.193 | 0.292 | 0.203 | 0.287 |
| | 192 | **0.241** | 0.322 | 0.261 | 0.318 | 0.251 | **0.310** | 0.287 | 0.338 | 0.221 | 0.306 | **0.219** | 0.293 | 0.220 | **0.292** | 0.229 | 0.301 | 0.246 | 0.305 | 0.249 | 0.309 | 0.284 | 0.362 | 0.269 | 0.328 |
| | 336 | **0.286** | **0.345** | 0.319 | 0.355 | 0.319 | 0.351 | 0.337 | 0.367 | **0.235** | **0.314** | 0.271 | 0.329 | 0.268 | 0.326 | 0.286 | 0.341 | 0.305 | 0.343 | 0.321 | 0.351 | 0.369 | 0.427 | 0.325 | 0.366 |
| | 720 | **0.371** | **0.404** | 0.415 | 0.410 | 0.420 | 0.410 | 0.430 | 0.416 | **0.283** | **0.353** | 0.352 | 0.379 | 0.350 | 0.380 | 0.378 | 0.401 | 0.410 | 0.405 | 0.408 | 0.403 | 0.554 | 0.522 | 0.421 | 0.415 |
| | Avg | **0.273** | 0.340 | 0.300 | 0.341 | 0.293 | **0.334** | 0.321 | 0.356 | **0.233** | 0.314 | 0.251 | **0.313** | 0.251 | 0.313 | 0.267 | 0.326 | 0.285 | 0.328 | 0.291 | 0.333 | 0.350 | 0.401 | 0.305 | 0.349 |
| **ETTh1** | 96 | **0.364** | 0.404 | 0.375 | **0.402** | 0.397 | 0.418 | 0.449 | 0.424 | **0.274** | **0.301** | 0.362 | 0.392 | 0.371 | 0.394 | 0.376 | 0.397 | 0.404 | 0.413 | 0.384 | 0.402 | 0.386 | 0.400 | 0.376 | 0.419 |
| | 192 | **0.384** | **0.392** | 0.399 | 0.419 | 0.434 | 0.439 | 0.503 | 0.453 | **0.290** | **0.339** | 0.398 | 0.418 | 0.403 | 0.412 | 0.416 | 0.418 | 0.454 | 0.440 | 0.436 | 0.429 | 0.437 | 0.432 | 0.420 | 0.448 |
| | 336 | **0.394** | **0.409** | 0.412 | 0.429 | 0.468 | 0.457 | 0.540 | 0.477 | **0.383** | 0.424 | 0.430 | 0.427 | 0.420 | **0.422** | 0.442 | 0.433 | 0.497 | 0.462 | 0.491 | 0.469 | 0.481 | 0.459 | 0.459 | 0.465 |
| | 720 | **0.412** | **0.415** | 0.413 | 0.444 | 0.469 | 0.477 | 0.515 | 0.489 | **0.387** | **0.428** | 0.442 | 0.457 | 0.422 | 0.444 | 0.477 | 0.456 | 0.496 | 0.481 | 0.521 | 0.500 | 0.519 | 0.516 | 0.506 | 0.507 |
| | Avg | **0.388** | **0.405** | 0.399 | 0.424 | 0.442 | 0.448 | 0.502 | 0.461 | **0.334** | **0.383** | 0.408 | 0.423 | 0.404 | 0.418 | 0.428 | 0.426 | 0.463 | 0.449 | 0.458 | 0.450 | 0.456 | 0.452 | 0.440 | 0.460 |
| **ETTh2** | 96 | 0.321 | 0.362 | **0.281** | **0.334** | 0.296 | 0.345 | 0.303 | 0.349 | **0.241** | **0.301** | 0.268 | 0.328 | 0.262 | 0.332 | 0.285 | 0.342 | 0.312 | 0.358 | 0.340 | 0.374 | 0.333 | 0.387 | 0.358 | 0.397 |
| | 192 | 0.417 | 0.425 | **0.340** | **0.373** | 0.374 | 0.394 | 0.391 | 0.399 | **0.275** | **0.375** | 0.329 | 0.375 | 0.328 | 0.377 | 0.354 | 0.389 | 0.397 | 0.408 | 0.402 | 0.414 | 0.477 | 0.476 | 0.429 | 0.439 |
| | 336 | 0.426 | 0.437 | **0.362** | **0.393** | 0.415 | 0.427 | 0.422 | 0.428 | **0.302** | **0.372** | 0.368 | 0.409 | 0.353 | 0.396 | 0.373 | 0.407 | 0.435 | 0.440 | 0.452 | 0.452 | 0.594 | 0.541 | 0.496 | 0.487 |
| | 720 | 0.473 | 0.474 | **0.380** | **0.416** | 0.425 | 0.444 | 0.429 | 0.449 | **0.323** | **0.386** | 0.372 | 0.420 | 0.383 | 0.425 | 0.406 | 0.441 | 0.436 | 0.449 | 0.462 | 0.468 | 0.831 | 0.657 | 0.463 | 0.474 |
| | Avg | 0.409 | 0.425 | **0.341** | **0.379** | 0.378 | 0.403 | 0.386 | 0.406 | **0.285** | **0.358** | 0.334 | 0.383 | 0.331 | 0.383 | 0.355 | 0.395 | 0.395 | 0.414 | 0.414 | 0.427 | 0.559 | 0.515 | 0.437 | 0.449 |
| **Electricity** | 96 | **0.183** | 0.298 | 0.205 | 0.299 | 0.196 | **0.287** | 0.232 | 0.321 | **0.128** | **0.221** | 0.131 | 0.224 | 0.128 | 0.223 | 0.139 | 0.238 | 0.186 | 0.269 | 0.168 | 0.272 | 0.197 | 0.282 | 0.193 | 0.308 |
| | 192 | **0.192** | 0.302 | 0.220 | 0.310 | 0.199 | **0.291** | 0.234 | 0.325 | **0.147** | 0.240 | 0.152 | 0.241 | 0.146 | 0.240 | 0.153 | 0.251 | 0.190 | **0.273** | 0.184 | 0.289 | 0.196 | 0.285 | 0.201 | 0.315 |
| | 336 | **0.203** | **0.298** | 0.236 | 0.323 | 0.214 | 0.305 | 0.249 | 0.338 | **0.149** | **0.244** | 0.160 | 0.248 | 0.163 | 0.258 | 0.169 | 0.266 | 0.206 | 0.290 | 0.198 | 0.300 | 0.209 | 0.301 | 0.214 | 0.329 |
| | 720 | **0.230** | **0.333** | 0.270 | 0.347 | 0.254 | 0.335 | 0.289 | 0.366 | **0.172** | **0.272** | 0.192 | 0.298 | 0.200 | 0.292 | 0.206 | 0.297 | 0.247 | 0.322 | 0.220 | 0.320 | 0.245 | 0.333 | 0.246 | 0.355 |
| | Avg | **0.202** | 0.308 | 0.233 | 0.320 | 0.216 | **0.305** | 0.251 | 0.338 | **0.149** | **0.244** | 0.158 | 0.252 | 0.159 | 0.253 | 0.167 | 0.263 | 0.207 | 0.289 | 0.192 | 0.295 | 0.212 | 0.300 | 0.214 | 0.327 |
| **Traffic** | 96 | **0.309** | **0.214** | 0.343 | 0.263 | 0.328 | 0.252 | 0.388 | 0.282 | **0.284** | **0.203** | 0.362 | 0.248 | 0.372 | 0.259 | 0.388 | 0.282 | 0.360 | 0.249 | 0.593 | 0.321 | 0.420 | 0.282 | 0.587 | 0.366 |
| | 192 | **0.320** | **0.240** | 0.383 | 0.277 | 0.346 | 0.261 | 0.407 | 0.290 | **0.293** | **0.211** | 0.374 | 0.247 | 0.391 | 0.265 | 0.407 | 0.290 | 0.379 | 0.256 | 0.617 | 0.336 | 0.424 | 0.287 | 0.604 | 0.373 |
| | 336 | **0.328** | **0.241** | 0.390 | 0.281 | 0.354 | 0.265 | 0.412 | 0.294 | **0.308** | **0.215** | 0.385 | 0.271 | 0.405 | 0.275 | 0.412 | 0.294 | 0.392 | 0.264 | 0.629 | 0.336 | 0.436 | 0.296 | 0.621 | 0.383 |
| | 720 | **0.331** | **0.260** | 0.420 | 0.296 | 0.396 | 0.286 | 0.450 | 0.312 | **0.319** | **0.223** | 0.430 | 0.288 | 0.437 | 0.292 | 0.450 | 0.312 | 0.432 | 0.286 | 0.640 | 0.350 | 0.466 | 0.315 | 0.626 | 0.382 |
| | Avg | **0.322** | **0.239** | 0.384 | 0.279 | 0.356 | 0.266 | 0.414 | 0.295 | **0.301** | **0.213** | 0.388 | 0.264 | 0.401 | 0.273 | 0.414 | 0.295 | 0.391 | 0.264 | 0.620 | 0.336 | 0.437 | 0.264 | 0.611 | 0.376 |
| **Weather** | 96 | **0.157** | **0.206** | 0.173 | 0.212 | 0.171 | 0.214 | 0.212 | 0.251 | **0.133** | **0.195** | 0.147 | 0.201 | 0.147 | 0.196 | 0.162 | 0.212 | 0.177 | 0.218 | 0.172 | 0.220 | 0.196 | 0.255 | 0.217 | 0.296 |
| | 192 | **0.204** | **0.250** | 0.216 | 0.250 | 0.217 | 0.254 | 0.261 | 0.288 | **0.184** | **0.237** | 0.189 | **0.234** | 0.191 | 0.238 | 0.204 | 0.248 | 0.222 | 0.259 | 0.219 | 0.261 | 0.237 | 0.296 | 0.276 | 0.336 |
| | 336 | **0.251** | **0.279** | 0.260 | 0.282 | 0.274 | 0.293 | 0.313 | 0.324 | **0.207** | **0.258** | 0.262 | 0.279 | 0.241 | 0.277 | 0.254 | 0.286 | 0.277 | 0.297 | 0.280 | 0.306 | 0.283 | 0.335 | 0.339 | 0.380 |
| | 720 | **0.309** | **0.317** | 0.320 | 0.322 | 0.351 | 0.343 | 0.386 | 0.372 | **0.264** | **0.313** | 0.304 | 0.316 | 0.313 | 0.329 | 0.326 | 0.337 | 0.352 | 0.347 | 0.365 | 0.359 | 0.345 | 0.381 | 0.403 | 0.428 |
| | Avg | **0.230** | **0.263** | 0.242 | 0.267 | 0.253 | 0.276 | 0.293 | 0.309 | **0.202** | **0.246** | 0.225 | 0.257 | 0.223 | 0.260 | 0.237 | 0.271 | 0.257 | 0.280 | 0.259 | 0.287 | 0.265 | 0.317 | 0.309 | 0.360 |
| 1st Count | | **50** | | 11 | | 9 | | 0 | | **60** | | 7 | | 2 | | 0 | | 1 | | 0 | | 0 | | 0 | |

Table 3: Comparison of the performance on **Zero-shot Forecasting** task. We boldface the best performance in each metric. Where source→target indicates that the model is first pretrained on the source domain, subsequently, the model parameters are frozen and predicted on the target domain.

| Metric | UTSD | | TimeLLM | | LLMTime | | GPT4TS | | DLinear | | PatchTST | | TimesNet | | FEDformer | | Autoformer | | Informer | |
|---|---|---|---|---|---|---|---|---|---|---|---|---|---|---|---|---|---|---|---|---|---|
| | MSE | MAE | MSE | MAE | MSE | MAE | MSE | MAE | MSE | MAE | MSE | MAE | MSE | MAE | MSE | MAE | MSE | MAE | MSE | MAE |
| ETTm2→ETTm1 | **0.404** | **0.411** | 0.414 | 0.438 | 1.933 | 0.984 | 0.790 | 0.579 | 0.516 | 0.473 | 0.596 | 0.508 | 0.857 | 0.599 | 0.718 | 0.564 | 0.722 | 0.566 | 1.180 | 0.804 |
| ETTm1→ETTm2 | 0.283 | 0.349 | **0.268** | **0.320** | 1.867 | 0.869 | 0.342 | 0.369 | 0.360 | 0.410 | 0.325 | 0.361 | 0.357 | 0.384 | 0.321 | 0.360 | 0.325 | 0.365 | 0.513 | 0.518 |
| ETTh2→ETTh1 | **0.425** | **0.439** | 0.479 | 0.474 | 1.961 | 0.981 | 0.780 | 0.604 | 0.609 | 0.532 | 0.616 | 0.537 | 0.920 | 0.635 | 0.746 | 0.598 | 0.735 | 0.593 | 1.201 | 0.842 |
| ETTh1→ETTh2 | **0.337** | **0.375** | 0.353 | 0.387 | 0.992 | 0.708 | 0.420 | 0.430 | 0.478 | 0.483 | 0.416 | 0.444 | 0.443 | 0.442 | 0.444 | 0.463 | 0.445 | 0.459 | 0.729 | 0.652 |
| ETTm1→ETTh2 | **0.370** | **0.398** | 0.381 | 0.412 | 0.992 | 0.708 | 0.433 | 0.439 | 0.464 | 0.475 | 0.439 | 0.438 | 0.457 | 0.454 | 0.468 | 0.483 | 0.470 | 0.479 | 0.768 | 0.680 |
| ETTh1→ETTm2 | 0.301 | 0.366 | **0.273** | **0.340** | 1.867 | 0.869 | 0.325 | 0.363 | 0.415 | 0.452 | 0.314 | 0.360 | 0.327 | 0.361 | 0.455 | 0.487 | 0.457 | 0.483 | 0.747 | 0.686 |
| ETTh1→ETTh2 | **0.303** | **0.368** | 0.353 | 0.387 | 0.992 | 0.708 | 0.406 | 0.422 | 0.493 | 0.488 | 0.380 | 0.405 | 0.421 | 0.431 | 0.582 | 0.548 | 0.784 | 0.781 | 0.973 | 1.092 |

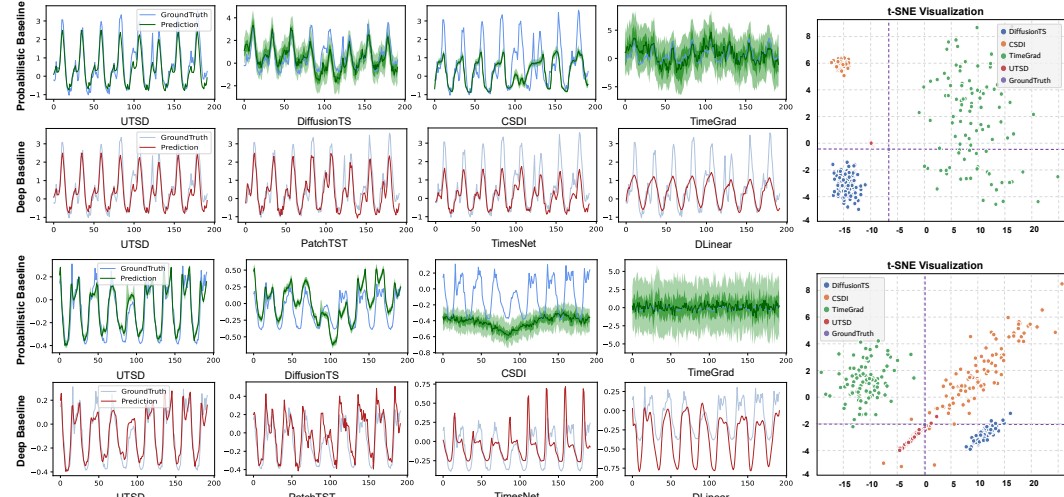

Figure 4: Visualization of comparisons between UTSD and exsting probabilistic and deep model baselines on the Electricity (upper) and Traffic (bottom) dataset.

Table 4: Comparison of the performance from diverse diffusion model on **Probabilistic Forecasting** task with $L = 336, H = 96$. We boldface the best performance in each metric. The unit of $STA$ is $10^{-2}$.

| Metric | | Ours | | | | | DiffusionTS | | | | | LDT | | | | | CSDI | | | | | TimeGrad | | | | |
|---|---|---|---|---|---|---|---|---|---|---|---|---|---|---|---|---|---|---|---|---|---|---|---|---|---|---|
| | | topQ | midQ | lastQ | AVG | STA | topQ | midQ | lastQ | AVG | STA | topQ | midQ | lastQ | AVG | STA | topQ | midQ | lastQ | AVG | STA | topQ | midQ | lastQ | AVG | STA |
| ETTh1 | MSE | 0.274 | 0.274 | 0.276 | **0.274** | **0.028** | 0.377 | 0.401 | 0.454 | 0.442 | 0.175 | 0.611 | 0.632 | 0.758 | 0.720 | 0.271 | 0.962 | 0.751 | 0.848 | 0.854 | 0.445 | 1.297 | 1.187 | 0.739 | 1.095 | 0.517 |
| | MAE | 0.301 | 0.301 | 0.303 | **0.302** | **0.016** | 0.411 | 0.439 | 0.499 | 0.480 | 0.073 | 0.668 | 0.685 | 0.832 | 0.786 | 0.142 | 0.704 | 0.578 | 0.615 | 0.632 | 0.155 | 1.118 | 1.009 | 0.811 | 1.039 | 0.323 |
| ETTh2 | MSE | 0.239 | 0.241 | 0.249 | **0.243** | **0.065** | 0.455 | 0.314 | 0.493 | 0.427 | 0.225 | 0.583 | 0.669 | 0.741 | 0.696 | 0.419 | 0.432 | 0.741 | 1.659 | 0.944 | 1.319 | 0.856 | 1.007 | 1.184 | 0.931 | 0.922 |
| | MAE | 0.301 | 0.300 | 0.308 | **0.303** | **0.030** | 0.556 | 0.416 | 0.581 | 0.530 | 0.136 | 0.708 | 0.745 | 0.825 | 0.750 | 0.209 | 0.458 | 0.579 | 0.892 | 0.643 | 0.557 | 0.957 | 1.232 | 1.324 | 1.135 | 0.390 |
| ETTm1 | MSE | 0.297 | 0.298 | 0.302 | **0.299** | **0.019** | 0.292 | 0.291 | 0.349 | 0.324 | 0.146 | 0.408 | 0.432 | 0.434 | 0.418 | 0.143 | 0.395 | 0.315 | 0.384 | 0.365 | 0.193 | 0.619 | 0.476 | 0.729 | 0.628 | 0.508 |
| | MAE | 0.331 | 0.332 | 0.336 | **0.333** | **0.008** | 0.338 | 0.343 | 0.384 | 0.363 | 0.087 | 0.465 | 0.499 | 0.512 | 0.486 | 0.106 | 0.407 | 0.352 | 0.401 | 0.387 | 0.180 | 0.631 | 0.547 | 0.740 | 0.708 | 0.377 |
| ETTm2 | MSE | 0.204 | 0.191 | 0.186 | **0.194** | **0.036** | 0.312 | 0.279 | 0.334 | 0.302 | 0.216 | 0.377 | 0.369 | 0.416 | 0.382 | 0.281 | 0.199 | 0.331 | 0.753 | 0.428 | 0.719 | 0.471 | 0.513 | 0.669 | 0.574 | 0.590 |
| | MAE | 0.300 | 0.285 | 0.278 | **0.288** | **0.019** | 0.412 | 0.357 | 0.461 | 0.395 | 0.153 | 0.471 | 0.439 | 0.565 | 0.490 | 0.221 | 0.271 | 0.366 | 0.581 | 0.406 | 0.328 | 0.518 | 0.573 | 0.757 | 0.662 | 0.389 |

## 4.3 PROBABILISTIC FORECASTING

Diffusion-based forecasting methdologies generally face the challenge of generation dithering, and existing models mitigate this difficulty through repeatedly sampling and subsequently taking the average or median as the final prediction result. In real-world scenarios, models are required to generate stable predictions for a fixed observation sequence. Besides, repeated sampling inevitably results in intolerable inference overhead since iterative denoising is required for each sample. Based on this, improved evaluation metrics are utilized to simultaneously measure prediction accuracy and generation stability. Table 4 demonstrates the mse and mae between the predicted and true results at different quantile points. All probabilistic models first repeat the sampling 100 times, and then take out the values at the 25, 50, and 75 percentile positions at each time point, sorted from smallest to largest, and calculate the mean square error (mae is the same) between this prediction and the true result, denoted as Top Quartile MSE (topQ), Middle Quartile MSE (midQ) and Last Quartile MSE (lastQ), respectively. In addition, the standard deviation of the distribution consisting of the mean square error of all the predictions is displayed as the stability of the prediction (STA) in Table 4. Specifically, the average MSE is reduced by **32.3%**, **54.3%** and **90.2%** compared to the existing DiffusionTS, LDT and CSDI.

## 4.4 VISUALIZATION

To visualize the performance of UTSD compared to the deep models and probabilistic models, three visualization tasks were devised. We plotted the prediction intervals for the probabilistic baseline as shown in the upper left of Figure 4, where the light and dark green colors indicate the prediction results for the 10-90% and 25-75% confidence intervals (with 50 repetitive samples for each model), and the blue and green curves indicate the ground truth and median prediction results, respectively. Specifically, compared to other probabilistic methods with large prediction intervals, it is proposed that the prediction results of UTSD in multiple benchmarks converge to a very small region.

In addition, on the right part of Figure 4, t-SNE is utilized to project the all prediction results onto a two-dimensional space, where the cross dashed lines mark the ground truth. Among them, UTSD 's predictions are more aggregated and closest to the actual sequence, which implies that UTSD has stronger generative robustness compared to other diffusion models and can be utilized in real-world highly accurate scenarios.

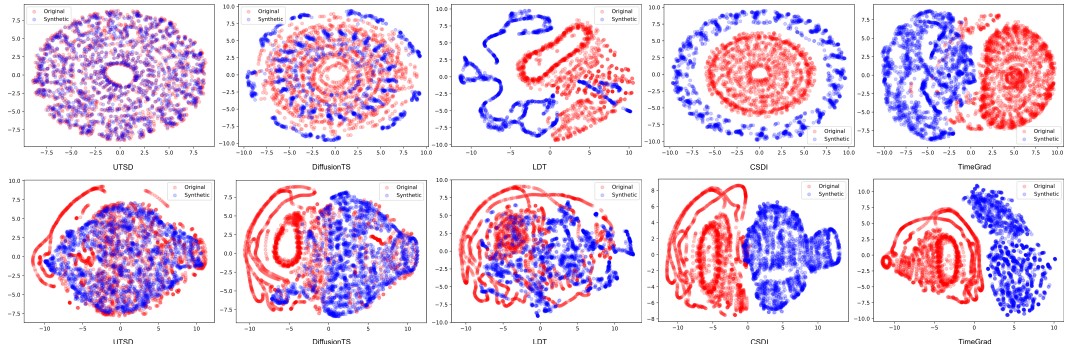

Figure 5: Visualization of comparisons between UTSD and exsting probabilistic and deep model baselines on the ETTh1 (Upper) and ETTh2 (Bottom) dataset.

The bottom left of Figure 4 demonstrates the prediction results of the deep model baseline, where the blue and red curves indicate the ground truth and prediction results, respectively. Note that UTSD demonstrates that the prediction results with the single sampling alone can match or even outperform the traditional regression model. The results of the visualization demonstrate that the proposed unified diffusion architecture to model the distribution of multiple domains are broadly effective, which ensures the superior performance of UTSD .

To further demonstrate the distributions of the generated and truth series, Figure 5 shows the results of comparing several probabilistic baselines with UTSD . The results show that UTSD learns comprehensive characterization information and generates sequences that are more consistent with the actual distribution. More visualization results of the baseline models on the multiple dataset are shown in the Appendix.

Table 5: Ablation study for model architecture.

| w/o Adapter | | | | w/o Classifier-free | | | | w/o ConditionNet | | | |
|---|---|---|---|---|---|---|---|---|---|---|---|
| ETTh1 | ETTm1 | ECL | Weather | ETTh1 | ETTm1 | ECL | Weather | ETTh1 | ETTm1 | ECL | Weather |
| MSE  MAE | MSE  MAE | MSE  MAE | MSE  MAE | MSE  MAE | MSE  MAE | MSE  MAE | MSE  MAE | MSE  MAE | MSE  MAE | MSE  MAE | MSE  MAE |
| 0.347  0.412 | 0.344  0.412 | 0.169  0.278 | 0.204  0.254 | 0.371  0.443 | 0.339  0.408 | 0.17  0.285 | 0.224  0.279 | 0.412  0.487 | 0.361  0.431 | 0.189  0.316 | 0.251  0.311 |
| ↓5.8% | ↓13.7% | ↓13.5% | ↓2.1% | ↓13.3% | ↓12.2% | ↓15.4% | ↓12.0% | ↓25.3% | ↓19.1% | ↓27.9% | ↓25.2% |

## 4.5 ABLATION STUDY

To elaborate on the property of our proposed UTSD , we conduct detailed ablations on model architecture. As shown in Table 5, we find that removing the *ConditionNet* module in UTSD will cause significant performance degradation. These results may come from that the proposed condition network will improve the the generalization capability of UTSD to learn multi-scale representations from complex sequences, and this temporal information is crucial for the reverse denoising process.

Specifically, for fairness purposes, the ablation model *w/o Adapter* is designed to feed the multilevel features captured by ConditionNet directly to DenoisingNet. In addition, the ablation model *w/o Classifier-free* follows the same design as the traditional condition diffusion model. In *w/o ConditionNet*, the observation sequences are directly input into the denoising model as prompt information.

From Table 5, we can find that the performance of *w/o ConditionNet* is degraded by **27.9%** on the ECL dataset, and by **25.2%** on the Weather dataset respectively, and degraded **25.3%** and **19.1%** on the ETT dataset, respectively. Similar results were obtained for ablation experiments on other components, demonstrating the superiority of the design.

## 5 CONCLUSION AND FUTURE WORK

In this paper, a unified time series diffusion (UTSD) model was established for the first time to model the joint probability distribution of multiple data domains by using the powerful probability distribution modeling ability of Diffusion. To ensure that the model has sufficient generalization ability for the generation task of multiple data domains, UTSD contains two pivotal modules: ConditionNet learns the general representation of fluctuation patterns from multiple domains in the pre-training phase, and DenoisingNet accepts multi-scale representations as conditional context in the reverse denoising process. In the fine-tuning stage, ConditionNet and DenoisingNet are frozen, and the Transfer-Adapter Module is used to transform the fluctuation patterns shared across domains into the corresponding latent space of the downstream data domain, so as to allow the model to generate time series samples that match the style of the specified data domain. Besides, this paper also designs the diffusion and denoising process on the actual sequence space, combined with the improved classifier-free guidance as condition generation strategy, which greatly improves the accuracy of the model in the forecasting task.

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

# A RELATED WORK

## A.1 LLM-BASED TS MODEL

The first attempt to establish a unified time series model is OneFitsAll (Zhou et al., 2023). OneFitsAll uses GPT2 (Radford et al., 2019) pretrained from billions of tokens as a backbone, where it freezes the self-attention and feedforward layers in the pretrained language model and evaluates by fine-tuning the output and normalization layers on the time series data domain. However, the semantic information in the pre-trained model is difficult to be directly used in temporal scene. TimeLLM (Jin et al., 2023) uses text prototypes to reprogram the input sequence data and then feed it into the frozen LLM (Touvron et al., 2023) to align the two modalities of time series and natural language. In addition, to fully activate the modeling ability of LLM for time-series data, TEST (Sun et al., 2023) builds an encoder to align the embedding Spaces of two modalities by comparing the alignment of instances, features and text prototypes. Although LLM-based models show good zero-shot inference ability, these models still face cross-modal challenges, and it is still urgent to establish a time series foundation model trained from scratch.

## A.2 UNIFIED TS MODEL

Different from NLP (Brown et al., 2020; Hu et al., 2022) and CV (Ho et al., 2020; Rombach et al., 2021), the background knowledge and statistical characteristics of time series data from different domains often vary greatly (Woo et al., 2024), so it is challenging to train a unified time series model by utilizing multiple data domains. The first attempt to cross-domain training is UniTime (Liu et al., 2024a), which uses domain instructions and a Language-TS transfer module to provide recognition information to distinguish time series data from different domains, and uses masking technology to alleviate the problem of unbalanced domain convergence speed. On this basis, many time series foundation models have emerged. The first is the MO-MENT (Goswami et al., 2024) of encoder-only attention architecture with input patching; Then, in order to overcome the differences between data domains, MOIRAI (Woo et al., 2024) based on mask encoder architecture includes multiple input-output projection layers to deal with different patterns of frequency-varying time series, and a spatio-temporal shared attention mechanism is designed. Different from the mainstream encoder-only architectures, decoder-only based timers show similar capabilities to large language models (Das et al., 2024). In addition to showing strong generalization in zero-shot inference, the Timer (Liu et al., 2024c) based on autoregressive generation strategy can capture the temporal representation from any length of context.

## A.3 PRE-TRAINED DIFFUSION ON VISION GENERATION

Due to its powerful distribution modeling and generation capabilities, Diffusion has quickly become a popular component in the field of high-quality image generation (Balaji & Nah, 2022; Huang et al., 2023; Nichol & Dhariwal, 2022). However, the image generation paradigm of diffusion through iterative denoising process leads to a large amount of time overhead. In order to ensure the quality of generated images while reducing the time overhead required for inference, LDM (Rombach et al., 2021) successfully compresses the forward diffusion process and the reverse noise reduction process from the real image space to the latent space through the pre-trained VAE (van den Oord et al., 2017), which greatly reduces the computational overhead and memory requirements for inference. Subsequently, researchers have focused on generating high-quality images that meet user expectations. Large pre-trained text-image diffusion models (Rombach et al., 2021) based on CLIP (Radford et al., 2021) allow users to input text as prompt to generate pictures with specified styles. Since it is often difficult to describe every image/video detail with text alone, there are many works (Hoe et al., 2024; Qi et al., 2024) by providing additional inputs as condition context. ControlNet (Zhang et al., 2023) in particular, by producing trainable copies of its encoder connected to zero convolutions, by reusing a powerful backbone derived from pre-training process, Additional images provided by the user (e.g., Canny Edge (Canny, 1986), Depth Map (Ranftl et al., 2019), and Normal Map (Vasiljevic et al., 2019), etc.) to enable more fine-grained spatial control.

## A.4 DIFFUSION MODEL ON TS

Since the success of diffusion models in vision, TimeGrad (Rasul et al., 2021) is the first time to use diffusion model to model the probability distribution of time series data, which chooses LSTM or GRU model as the architecture to predict future series in an autoregressive way. Subsequently, CSDI (Tashiro et al., 2021) and SSSD (Alcaraz & Strodthoff, 2022) designed a diffusion model with the observed data as the context condition, and filled the missing part by introducing the noise of the diffusion process into the missing part, and then gradually denoising at each step. To improve the learning ability to model long-term dependencies in time series data, TimeDiff (Shen & Kwok, 2023) introduces future-Mixup and autoregressive initialization mechanisms to predict all timepoints of future sequence. DiffusionTS (Yuan & Qiao, 2024) utilizes the transformer architecture to model the seasonal-trend components separately, and the Fourier-based losses are designed to reconstruct the

sequence sample directly rather than the noise at each diffusion step. Inspired by LDM, recent LDT (Li et al., 2024) utilizes a transformer-based autoencoder to learn latent representations from raw observation sequence and subsequently predicts future sequence in a non-autoregressive manner in the latent space.

Although there have been a lot of methodologies on the application of diffusion to time series, to the best of our knowledge, there are few researches about building a unified time series diffusion model. The establishment of UTSD faces the challenge of capturing distributions from multiple data domains, and only introducing inductive biases may not be sufficient for diffusion models to capture distribution characteristics of different domains (Shen et al., 2024). The Condition-Denoising architecture is designed where the independent ConditionNet ensures that the model can capture multi-scale domain specified pattern features. In addition, LDMs are widely used in Vision (Zhang et al., 2023; Esser & Kulal, 2024; Liu et al., 2024b) because high-resolution images are generally large in size (512*512), while the number of timepoints in time-series data is usually less than 1,000. By modeling in the actual sequence space instead of the latent space, the proposed UTSD avoids error accumulation in the latent space during multiple rounds of iterative noise reduction (e.g., the number of steps of DDPM is 200).

# B   DIFFUSION MODEL

The diffusion model is a popular generative model and has attracted significant attention in various domains, such as image, video, 3-D objective, etc. A well-known diffusion model is the denoising diffusion probabilistic model (DDPM). DDPM consists of a forward diffusion process and a backward denoising process. The diffusion process means gradually adding Gaussian noise to the real samples of the dataset, while the denoising process means gradually denoising the noisy data to restore the real data points.

Given a data point sampled from a real data distribution $\mathbf{x}_0 \sim q(\mathbf{x})$, the forward diffusion process gradually adds Gaussian noise $T$ steps, producing a series of noisy samples $\{\mathbf{x}_1, \cdots, \mathbf{x}_T\}$. The step size are controlled by a variance schedule $\{\beta_T \in (0,1)\}_{t=1}^{T}$. The diffusion process can be formulated as:

$$q(\mathbf{x}_t|\mathbf{x}_{t-1}) = \mathcal{N}(\mathbf{x}_t; \sqrt{1-\beta_t}\mathbf{x}_{t-1}, \beta_t \mathbf{I}). \tag{4}$$

Which means $\mathbf{x}_t$ is sampled from $q(\mathbf{x}_t|\mathbf{x}_{t-1})$, satisfied the Gaussian distribution $\mathcal{N}(\mathbf{x}_t; \sqrt{1-\beta_t}\mathbf{x}_{t-1}, \beta_t \mathbf{I})$. The diffusion process follows a Markov process:

$$q(\mathbf{x}_{1:T}|\mathbf{x}_0) = \prod_{t=1}^{T} q(\mathbf{x}_t|\mathbf{x}_{t-1}) \tag{5}$$

$\mathbf{x}_t$ is defined by $\mathbf{x}_{t-1}$ and $\beta_T$, and can be directly calculated with given $\mathbf{x}_0$ and $\{\beta_1, \cdots, \beta_T\}$ step values. Let $\alpha_t = 1 - \beta_t$, and $\bar{\alpha}_t = \prod_{t=1}^{T} \alpha_i$, with the re-parametric, we get:

$$\begin{aligned} \mathbf{x}_t &= \sqrt{\alpha_t}\mathbf{x}_{t-1} + \sqrt{1-\alpha_t}\mathbf{z}_{t-1} \\ &= \sqrt{\alpha_t \alpha_{t-1}}\mathbf{x}_{t-2} + \sqrt{1-\alpha_t \alpha_{t-1}}\mathbf{z}_{t-2} \\ &= \cdots \\ &= \sqrt{\bar{\alpha}_t}\mathbf{x}_0 + \sqrt{1-\bar{\alpha}_t}\mathbf{z}_0 \end{aligned} \tag{6}$$

where $\mathbf{z}_0, \cdots, \mathbf{z}_{t-1} \sim \mathcal{N}(\mathbf{0}, \mathbf{I})$, and

$$q(\mathbf{x}_t|\mathbf{x}_0) = \mathcal{N}(\mathbf{x}_t; \sqrt{\bar{\alpha}_t}\mathbf{x}_0, (1-\bar{\alpha}_t)\mathbf{I}). \tag{7}$$

This indicates that with $\mathbf{x}_0$ and a fixed-value sequence $\{\beta_T \in (0,1)\}_{t=1}^{T}$, and sample $\mathbf{z}$ from norm distribution $\mathcal{N}(\mathbf{0}, \mathbf{I})$, $\mathbf{x}_t$ is defined. In general, we can afford a larger update step when the sample gets noisier, so $\beta_1 < \beta_2 < \cdots < \beta_T$, and therefore $\bar{\alpha}_1 > \cdots > \bar{\alpha}_T$.

The reverse denoising process samples from $q(\mathbf{x}_{t-1}|\mathbf{x}_t)$, and we can reconstruct the real data point for a random Gaussian distribution. However, we need to find the data distribution from the whole dataset, and we cannot predict the conditional distribution $q(\mathbf{x}_{t-1}|\mathbf{x}_t)$ directly, so we need to learn a model $p_\theta$ to approximately simulate this conditional probability to run the inverse diffusion process.

$$p_\theta(\mathbf{x}_{0:T}) = p(\mathbf{x}_T) \prod_{t=1}^{T} p_\theta(\mathbf{x}_{t-1}|\mathbf{x}_t)$$

$$p_\theta(\mathbf{x}_{t-1}|\mathbf{x}_t) = \mathcal{N}(\mathbf{x}_{t-1}; \mu_\theta(\mathbf{x}_t, t), \Sigma_\theta(\mathbf{x}_t, t)). \tag{8}$$

Given $\mathbf{x}_t$ and $\mathbf{x}_0$ the posteriori diffusion conditional probability can be formulated as :

$$q(\mathbf{x}_{t-1}|\mathbf{x}_t, \mathbf{x}_0) = \mathcal{N}(\mathbf{x}_{t-1}; \tilde{\mu}(\mathbf{x}_t, \mathbf{x}_0), \tilde{\beta}_t \mathbf{I}). \tag{9}$$

Following the Bayes' rule:

$$q(\mathbf{x}_{t-1}|\mathbf{x}_t, \mathbf{x}_0) = q(\mathbf{x}_t|\mathbf{x}_{t-1}, \mathbf{x}_0)\frac{q(\mathbf{x}_{t-1}|\mathbf{x}_0)}{q(\mathbf{x}_t|\mathbf{x}_0)}$$

$$\propto \exp\left(-\frac{1}{2}\left(\frac{(\mathbf{x}_t - \sqrt{\alpha_t}\mathbf{x}_{t-1})^2}{\beta_t} + \frac{(\mathbf{x}_{t-1} - \sqrt{\bar{\alpha}_{t-1}}\mathbf{x}_0)^2}{1 - \bar{\alpha}_{t-1}} - \frac{(\mathbf{x}_t - \sqrt{\bar{\alpha}_t}\mathbf{x}_0)^2}{1 - \bar{\alpha}_t}\right)\right) \tag{10}$$

$$= \exp\left(-\frac{1}{2}\left(\left(\frac{\alpha_t}{\beta_t} + \frac{1}{1 - \bar{\alpha}_{t-1}}\right)\mathbf{x}_{t-1}^2 - \left(\frac{2\sqrt{\alpha_t}}{\beta_t}\mathbf{x}_t + \frac{2\sqrt{\bar{\alpha}_{t-1}}}{1 - \bar{\alpha}_{t-1}}\mathbf{x}_0\right)\mathbf{x}_{t-1} + C(\mathbf{x}_t, \mathbf{x}_0)\right)\right)$$

where $C(\mathbf{x}_t, \mathbf{x}_0)$ is a function contains $\mathbf{x}_t$ and $\mathbf{x}_0$, without $\mathbf{x}_{t-1}$. The mean and variance can be calculated by:

$$\tilde{\beta}_t = 1/\left(\frac{\alpha_t}{\beta_t} + \frac{1}{1 - \bar{\alpha}_{t-1}}\right) = \frac{1 - \bar{\alpha}_{t-1}}{1 - \bar{\alpha}_t} \cdot \beta_t$$

$$\tilde{\mu}_t(\mathbf{x}_t, \mathbf{x}_0) = \left(\frac{\sqrt{\alpha_t}}{\beta_t}\mathbf{x}_t + \frac{\sqrt{\bar{\alpha}_{t-1}}}{1 - \bar{\alpha}_{t-1}}\mathbf{x}_0\right)/\left(\frac{\alpha_t}{\beta_t} + \frac{1}{1 - \bar{\alpha}_{t-1}}\right) \tag{11}$$

$$= \frac{\sqrt{\alpha_t}(1 - \bar{\alpha}_{t-1})}{1 - \bar{\alpha}_t}\mathbf{x}_t + \frac{\sqrt{\bar{\alpha}_{t-1}}\beta_t}{1 - \bar{\alpha}_t}\mathbf{x}_0$$

In the forward process, we have $\mathbf{x}_0 = \frac{1}{\sqrt{\bar{\alpha}_t}}(\mathbf{x}_t = \sqrt{1 - \bar{\alpha}_t}\mathbf{z}_t)$. Taking into Eqn. 11, we have:

$$\tilde{\mu}_t = \frac{\sqrt{\alpha_t}(1 - \bar{\alpha}_{t-1})}{1 - \bar{\alpha}_t}\mathbf{x}_t + \frac{\sqrt{\bar{\alpha}_{t-1}}\beta_t}{1 - \bar{\alpha}_t}\frac{1}{\sqrt{\bar{\alpha}_t}}(\mathbf{x}_t - \sqrt{1 - \bar{\alpha}_t}z_t) \tag{12}$$

$$= \frac{1}{\sqrt{\alpha_t}}\left(\mathbf{x}_t - \frac{\beta_t}{\sqrt{1 - \bar{\alpha}_t}}z_t\right) \tag{13}$$

To train the diffusion model, one uniformly samples $t$ form $\{1, 2, \cdots, T\}$ and then minimizes the following KL-divergence:

$$\mathcal{L}_t = D_{KL}(q(\mathbf{x}_{t-1}|\mathbf{x}_t)||p_\theta(\mathbf{x}_{t-1}|\mathbf{x}_t)). \tag{14}$$

Connecting Eqn. 9, 11, and 13, the training objective is transformed into:

$$\mathcal{L}_t = \frac{1}{2\sigma_t^2}||\tilde{\mu}_t(\mathbf{x}_t, \mathbf{x}_0, t) - \mu_\theta(\mathbf{x}_t, t)||^2. \tag{15}$$

## C MORE DISCUSSION ABOUT ARCHITECTURE

The pre-training, fine-tuning, and inference paradigm for UTSD is shown in Figure 9. **(a)** In the pretraining stage, all modules of UTSD are trained end-to-end on the fusion dataset with the forecasting task as the metric. **(b)** In the finetuning stage, only the adapter module is allowed to continue training on a specific dataset. **(c)** Finally, all the weights were frozen, and the prediction sequence was generated after $T$ rounds of iterative denoising.

Table 6: We compare the performance difference between two versions: UTSD and Latent-UTSD on multiple datasets derived from different domain knowledge, sampling rates, scale features. Extensive experimental results reflect the performance of the two versions of the model in the real world, which serves as the experimental basis for our final establishment of the UTSD model architecture. Specifically, the input sequence length is fixed bits **336**, and we bold the best performance in each metric.

| Metric | | ETTh1 | | | | | ETTm1 | | | | | Traffic | | | | | Weather | | | | |
|---|---|---|---|---|---|---|---|---|---|---|---|---|---|---|---|---|---|---|---|---|---|
| | | 96 | 192 | 336 | 720 | avg | 96 | 192 | 336 | 720 | avg | 96 | 192 | 336 | 720 | avg | 96 | 192 | 336 | 720 | avg |
| UTSD | MSE | 0.274 | 0.290 | 0.383 | 0.387 | **0.334** | 0.299 | 0.304 | 0.312 | 0.317 | **0.308** | 0.284 | 0.293 | 0.308 | 0.319 | **0.301** | 0.133 | 0.184 | 0.207 | 0.264 | **0.202** |
| | MAE | 0.301 | 0.339 | 0.424 | 0.428 | **0.383** | 0.333 | 0.358 | 0.365 | 0.368 | **0.356** | 0.203 | 0.211 | 0.215 | 0.223 | **0.213** | 0.195 | 0.237 | 0.258 | 0.313 | **0.246** |
| Latent-UTSD | MSE | 0.323 | 0.342 | 0.450 | 0.454 | 0.392 | 0.352 | 0.365 | 0.365 | 0.370 | 0.363 | 0.314 | 0.316 | 0.341 | 0.343 | 0.329 | 0.154 | 0.211 | 0.238 | 0.313 | 0.229 |
| | MAE | 0.348 | 0.386 | 0.493 | 0.492 | 0.430 | 0.390 | 0.418 | 0.435 | 0.443 | 0.422 | 0.223 | 0.236 | 0.242 | 0.242 | 0.235 | 0.226 | 0.271 | 0.297 | 0.359 | 0.288 |

Figure 6 illustrates the two frameworks contained in UTSD, with the left side showing the model infrastructure described in the main text and the right side showing the model architecture we discuss in this section, hereafter referred to as Latent-UTSD. Table 6 demonstrates the overall performance of the proposed UTSD and Latent-UTSD. In the algorithmic framework of Latent-UTSD, a multi-domain mixed comprehensive dataset is utilized

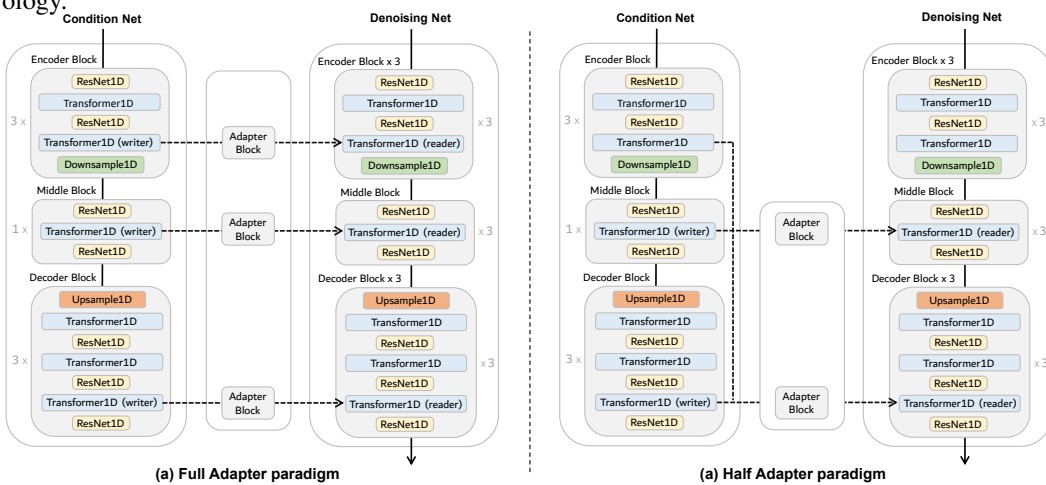

Figure 6: Illustration of Comparison about two computational frameworks proposed by our methodology.

Figure 7: Illustration of Comparison about two adapter paradigm proposed for our methodology.

to train the autoencoder with the end-to-end manner. Firstly, a pre-trained encoder is utilized to calculate the latent representation $Z_0$ of the original sequence $X^0_{-L+1:0}$, corresponding to which the initial random situation input in the denoising network has the same shape as the latent representation. Subsequently, the multi-scale fluctuation patterns embedded in the latent representation space are captured by the condition net, and the model reconstructs the latent representation of the fusion distribution from the gaussian distribution by iterative denoising process. Finally, the pre-trained decoder generates the forecast sequence $Y^0_{1:H}$ from the latent representation.

Table 7: Detailed experimental configuration of training preocess for each dataset.

| config | Multi-domain | ETTh1 | ETTh2 | ETTm1 | ETTm2 | Weather | ECL | Traffic |
|---|---|---|---|---|---|---|---|---|
| SquenceLen | 27,593,879 | 17,420 | 17,420 | 69,680 | 69,680 | 52,696 | 26,304 | 17,544 |
| Channel | 1 | 7 | 7 | 7 | 7 | 21 | 321 | 862 |
| BatchSize | 2,048 | 512 | 512 | 512 | 512 | 128 | 16 | 16 |
| TrainSteps | 1,000,000 | 10,000 | 10,000 | 10,000 | 10,000 | 20,000 | 100,000 | 100,000 |

Figure 7 demonstrates the two adapter application paradigms proposed in this paper, which are to establish connections between condition net and denoising net through the adapter block at the location of all Blocks, or only at the location of Middle Block as well as Decoder Block.

In addition to cross-domain conditional generation that includes forecasting and imputation, UTSD also supports the unconditional cross-domain generation task, the implementation of which is demonstrated in Figure 8. The unconditional generation paradigm of UTSD can be divided into two phases: 1) In the multi-domain pretraining phase, time series $Y^T_{1:H}$ from different domain backgrounds are successively obtained a set of noised

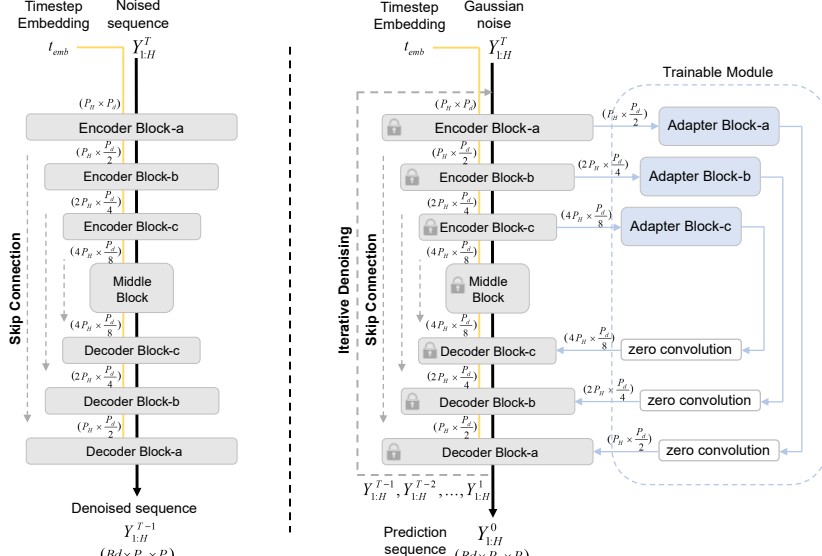

Figure 8: In the cross-domain time series generation paradigm, the UTSD architecture is connected by a bridge module consisting of an encoder block and an adapter block, and the locked grey block shows the backbone of the pre-trained obtained UTSD. based on this, several trainable consecutive adapter blocks (blue blocks) are added with a set of zero convolution layer (white) to construct the fine-tuned network. Which is used to align the multi-domain uniform representation space with the specified domain representation space.

sequence $\left\{Y_{1:H}^1, ..., Y_{1:H}^{T-1}, Y_{1:H}^T\right\}$. In each iterative step, the noised sequences $Y_{1:H}^T$ as the input of encoder block-a,b,c and a set of multi-scale condition variables is captured from it, and the rich fluctuation pattern information embedded is used to reconstruct the denoised result $Y_{1:H}^{T-1}$ in Decoder; 2) In the downstream data domain fine-tuning phase, the pre-training weights in Encoder and Decoder are frozen. An additional set of adapter block and zero convolution is introduced to align the multi-domain uniform representation space with the specified domain representation space.

In UTSD-generation, the trainable Adapter is connected to the locked pre-trained weights. The zero convolution with weights initialised to zero are designed to ensure that they grow progressively during training. This architecture ensures that harmful noise is not added to the deep features of large diffusion models at the start of training, and protects the large-scale pre-trained backbone in the trainable replicas from such noise.

# D    EXPERIMENTAL DETAILS

## D.1    MODEL AND DIFFUSION PROCESS HYPERPARAMETERS

The proposed model includes three components, ConditionNet, DenoisingNet and Adapter. ConditionNet and DenoisingNet are composed of the same Unet structure, which contains 3 encoderblocks, 3 decoderblocks and 1 MiddleBlock. In the UTSD architecture, all components are composed four minimal blocks (encoder block, decoder block, middle block and adapter block). The basic construction of each block is illustrated in Figure 2, however, in the concrete implementation we allow consecutive $L$-layer block residues to be stacked together to form stacked-block and used to form the Condition Net, Denoising Net and Adapter. Furthermore, the number of input and output channels of the middle block is denoted as $D$, and accordingly the number of output channels in the three pairs of encoder-decoder blocks are $D/4$, $D/2$, and $D$, respectively.

In order to verify the sensitivity of UTSD to hyperparameter selection, the Table 8 shows the performance of UTSD with different parameter scales on multiple benchmarks. Where the model parameter combinations include $(L, D) = (2, 128), (2, 256), (3, 128), (3, 256), (4, 128), (4, 256)$, with a fixed forecasting window of 96, and a performance metric of MSE.

By default, the hyperparameters of UTSD are fixed to L equal to 3 and D equal to 256, and all experimental results presented follow this setting. In the scratch forecasting task, the number of output channels in the three pairs of Encoder-Decoder blocks is 64, 128, 256, respectively, and the number of input and output channels of the Middle Block is fixed to 256. Correspondingly, the number of output channels of the three groups of blocks

Table 8: To verify the sensitivity of UTSD to hyperparameter selection, the table 1 shows the performance of UTSD with different parameter scales on multiple benchmarks, with a fixed forecasting window of 96, and a performance metric of MSE.

| Hyperparameter | ETTh1 | ETTh2 | ETTm1 | ETTm2 | Weather | ECL | Traffic |
|---|---|---|---|---|---|---|---|
| (L=2,D=128) | 0.352 | 0.307 | 0.335 | 0.244 | 0.215 | 0.153 | 0.311 |
| (L=2,D=256) | 0.344 | 0.302 | 0.319 | 0.254 | 0.211 | 0.158 | 0.325 |
| (L=3,D=128) | 0.328 | **0.274** | 0.321 | 0.239 | 0.203 | 0.154 | **0.296** |
| (L=3,D=256) | 0.334 | 0.285 | 0.308 | **0.233** | 0.202 | 0.149 | 0.301 |
| (L=4,D=128) | **0.322** | 0.280 | **0.302** | 0.235 | **0.198** | **0.145** | 0.298 |
| (L=4,D=256) | 0.340 | 0.313 | 0.319 | 0.234 | 0.214 | 0.160 | 0.306 |

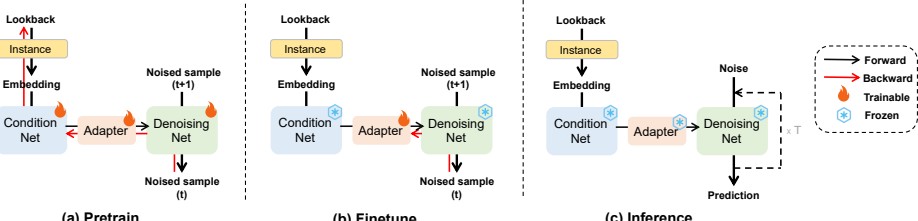

Figure 9: Illustration of the paradigm in the pre-training, fine-tuning, inference stages of the proposed UTSD architecture.

in the across-domain pretraining forecasting task is $128, 256, 512$ respectively. The patch size of lookback window is fixed to $48$ in the full prediction task. We adopted the setting of DDPM, where the number of backward iteration steps is set to 200 for the inference process and 1000 for the training process.

# E  DIFFUSION MODEL FOR TIME SERIES PREDICTION

The pre-training, fine-tuning, and inference paradigm for UTSD is shown in Figure 9. **(a)** In the pretraining stage, all modules of UTSD are trained end-to-end on the fusion dataset with the forecasting task as the metric. **(b)** In the finetuning stage, only the adapter module is allowed to continue training on a specific dataset. **(c)** Finally, all the weights were frozen, and the prediction sequence was generated after $T$ rounds of iterative denoising.

## E.1  EXPERIMENTAL CONFIGURATION

All experiments are repeated three times, implemented in PyTorch and conducted on a single Tesla V100 SXM2 32GB GPU. Our method is trained with the L2 Loss, using the ADAM optimizer with an initial learning rate of $10^{-4}$, and Batch size is set in $16 \rightarrow 256$. The mean square error (MSE) and mean absolute error (MAE) are used as metrics in all forecasting tasks. By default, the proposed **UTSD** contains 3 pairs of Encoder-Decoder Blocks. All the baselines that we reproduced are implemented based on configurations of the original paper or their official code. For a fair comparison, we design the same input embedding and final prediction layer for all base models. We provide the detailed experimental configuration about the batch size and number of training steps for several benchamrk in Table 7.

## E.2  BENCHMARKS

To evaluate the performance of the proposed method, we extensively experiment with the mainstream time series analysis tasks including long-term forecasting and imputation (i.e., predicting the missing data in a time series). The long-term forecasting and imputation are evaluated with several popular real-world datasets, including: **ETT (ETTh1, ETTh2, ETTm1, and ETTm2)**[1] (Zhou et al., 2021) contains six power load features and oil temperature used for monitoring electricity transformers. ETT involves four subsets. ETTm1 and ETTm2 are recorded at 15-minute intervals, while ETTh1 and ETTh2 are recorded hourly. **Weather**[2] contains 21 meteorological indicators, such as temperature, humidity, and precipitation, which are recorded every 10

---

[1]https://github.com/zhouhaoyi/Informer2020

[2]https://www.bgc-jena.mpg.de/wetter/

minutes in the year 2020. **Electricity**[3] comprises hourly power consumption of 321 clients from 2012 to 2014. **Traffic**[4] reports the number of vehicles loaded on all 862 roads at each moment in time.

Table 9: Comparison of the performance on **Zero-shot Forecasting** task. We boldface the best performance in each metric. Where source→target indicates that the model is first rained on the source domain, subsequently, the model parameters are frozen and predicted on the target domain.

| Metric | | UTSD | | TimeLLM | | LLMTime | | GPT4TS | | DLinear | | PatchTST | | TimesNet | | Autoformer | |
|---|---|---|---|---|---|---|---|---|---|---|---|---|---|---|---|---|---|
| | | MSE | MAE | MSE | MAE | MSE | MAE | MSE | MAE | MSE | MAE | MSE | MAE | MSE | MAE | MSE | MAE |
| ETTm2 → ETTm1 | 96 | 0.365 | 0.387 | 0.359 | 0.397 | 1.179 | 0.781 | 0.747 | 0.558 | 0.570 | 0.490 | 0.491 | 0.437 | 0.747 | 0.558 | 0.735 | 0.576 |
| | 192 | 0.384 | 0.400 | 0.390 | 0.420 | 1.327 | 0.846 | 0.781 | 0.560 | 0.590 | 0.506 | 0.530 | 0.470 | 0.781 | 0.560 | 0.753 | 0.586 |
| | 336 | 0.410 | 0.414 | 0.421 | 0.445 | 1.478 | 0.902 | 0.778 | 0.578 | 0.706 | 0.567 | 0.565 | 0.497 | 0.778 | 0.578 | 0.750 | 0.593 |
| | 720 | 0.457 | 0.441 | 0.487 | 0.488 | 3.749 | 1.408 | 0.769 | 0.573 | 0.731 | 0.584 | 0.686 | 0.565 | 0.769 | 0.573 | 0.782 | 0.609 |
| | Avg | 0.404 | 0.411 | 0.414 | 0.438 | 1.933 | 0.984 | 0.769 | 0.567 | 0.649 | 0.537 | 0.568 | 0.492 | 0.769 | 0.567 | 0.755 | 0.591 |
| ETTm1 → ETTm2 | 96 | 0.196 | 0.292 | 0.169 | 0.257 | 0.646 | 0.563 | 0.217 | 0.294 | 0.221 | 0.314 | 0.195 | 0.271 | 0.222 | 0.295 | 0.385 | 0.457 |
| | 192 | 0.248 | 0.325 | 0.227 | 0.318 | 0.934 | 0.654 | 0.277 | 0.327 | 0.286 | 0.359 | 0.258 | 0.311 | 0.288 | 0.337 | 0.433 | 0.469 |
| | 336 | 0.305 | 0.364 | 0.290 | 0.338 | 1.157 | 0.728 | 0.331 | 0.360 | 0.357 | 0.406 | 0.317 | 0.348 | 0.341 | 0.367 | 0.476 | 0.477 |
| | 720 | 0.384 | 0.413 | 0.375 | 0.367 | 4.730 | 1.531 | 0.429 | 0.413 | 0.476 | 0.476 | 0.416 | 0.404 | 0.436 | 0.418 | 0.582 | 0.535 |
| | Avg | 0.283 | 0.349 | 0.268 | 0.320 | 1.867 | 0.869 | 0.313 | 0.348 | 0.335 | 0.389 | 0.296 | 0.334 | 0.322 | 0.354 | 0.469 | 0.484 |
| ETTh2 → ETTh1 | 96 | 0.373 | 0.404 | 0.450 | 0.452 | 1.130 | 0.777 | 0.732 | 0.577 | 0.689 | 0.555 | 0.485 | 0.465 | 0.848 | 0.601 | 0.693 | 0.569 |
| | 192 | 0.405 | 0.422 | 0.465 | 0.461 | 1.242 | 0.820 | 0.758 | 0.559 | 0.707 | 0.568 | 0.565 | 0.509 | 0.860 | 0.610 | 0.760 | 0.601 |
| | 336 | 0.434 | 0.442 | 0.501 | 0.482 | 1.328 | 0.864 | 0.759 | 0.578 | 0.710 | 0.577 | 0.581 | 0.515 | 0.867 | 0.626 | 0.781 | 0.619 |
| | 720 | 0.489 | 0.488 | 0.501 | 0.502 | 4.145 | 1.461 | 0.781 | 0.597 | 0.704 | 0.596 | 0.628 | 0.561 | 0.887 | 0.648 | 0.796 | 0.644 |
| | Avg | 0.425 | 0.439 | 0.479 | 0.474 | 1.961 | 0.981 | 0.757 | 0.578 | 0.703 | 0.574 | 0.565 | 0.513 | 0.865 | 0.621 | 0.757 | 0.608 |
| ETTh1 → ETTh2 | 96 | 0.273 | 0.328 | 0.279 | 0.337 | 0.510 | 0.576 | 0.335 | 0.374 | 0.347 | 0.400 | 0.304 | 0.350 | 0.358 | 0.387 | 0.469 | 0.486 |
| | 192 | 0.309 | 0.356 | 0.351 | 0.374 | 0.523 | 0.586 | 0.412 | 0.417 | 0.447 | 0.460 | 0.386 | 0.400 | 0.427 | 0.429 | 0.634 | 0.567 |
| | 336 | 0.335 | 0.378 | 0.388 | 0.415 | 0.640 | 0.637 | 0.441 | 0.444 | 0.515 | 0.505 | 0.414 | 0.428 | 0.449 | 0.451 | 0.655 | 0.588 |
| | 720 | 0.430 | 0.439 | 0.391 | 0.420 | 2.296 | 1.034 | 0.438 | 0.452 | 0.665 | 0.589 | 0.419 | 0.443 | 0.448 | 0.458 | 0.570 | 0.549 |
| | Avg | 0.337 | 0.375 | 0.353 | 0.387 | 0.992 | 0.708 | 0.406 | 0.422 | 0.493 | 0.488 | 0.380 | 0.405 | 0.421 | 0.431 | 0.582 | 0.548 |
| ETTm1 → ETTh2 | 96 | 0.317 | 0.357 | 0.321 | 0.369 | 0.510 | 0.576 | 0.353 | 0.392 | 0.365 | 0.415 | 0.354 | 0.385 | 0.377 | 0.407 | 0.435 | 0.470 |
| | 192 | 0.351 | 0.384 | 0.389 | 0.410 | 0.523 | 0.586 | 0.443 | 0.437 | 0.454 | 0.462 | 0.447 | 0.434 | 0.471 | 0.453 | 0.495 | 0.489 |
| | 336 | 0.369 | 0.402 | 0.408 | 0.433 | 0.640 | 0.637 | 0.469 | 0.461 | 0.496 | 0.494 | 0.481 | 0.463 | 0.472 | 0.484 | 0.470 | 0.472 |
| | 720 | 0.441 | 0.448 | 0.406 | 0.436 | 2.296 | 1.034 | 0.466 | 0.468 | 0.541 | 0.529 | 0.474 | 0.471 | 0.495 | 0.482 | 0.480 | 0.485 |
| | Avg | 0.370 | 0.398 | 0.381 | 0.412 | 0.992 | 0.708 | 0.433 | 0.439 | 0.464 | 0.475 | 0.439 | 0.438 | 0.457 | 0.454 | 0.470 | 0.479 |
| ETTh1 → ETTm2 | 96 | 0.214 | 0.315 | 0.189 | 0.293 | 0.646 | 0.563 | 0.236 | 0.315 | 0.255 | 0.357 | 0.215 | 0.304 | 0.239 | 0.313 | 0.352 | 0.432 |
| | 192 | 0.269 | 0.348 | 0.237 | 0.312 | 0.934 | 0.654 | 0.287 | 0.342 | 0.338 | 0.413 | 0.275 | 0.339 | 0.291 | 0.342 | 0.413 | 0.460 |
| | 336 | 0.319 | 0.376 | 0.291 | 0.365 | 1.157 | 0.728 | 0.341 | 0.374 | 0.425 | 0.465 | 0.334 | 0.373 | 0.342 | 0.371 | 0.465 | 0.489 |
| | 720 | 0.403 | 0.425 | 0.372 | 0.390 | 4.730 | 1.531 | 0.435 | 0.422 | 0.640 | 0.573 | 0.431 | 0.424 | 0.434 | 0.419 | 0.599 | 0.551 |
| | Avg | 0.301 | 0.366 | 0.273 | 0.340 | 1.867 | 0.869 | 0.325 | 0.363 | 0.415 | 0.452 | 0.314 | 0.360 | 0.327 | 0.361 | 0.457 | 0.483 |
| ETTh1 → ETTh2 | 96 | 0.259 | 0.321 | 0.279 | 0.337 | 0.510 | 0.576 | 0.335 | 0.374 | 0.347 | 0.400 | 0.304 | 0.350 | 0.358 | 0.387 | 0.469 | 0.486 |
| | 192 | 0.295 | 0.349 | 0.351 | 0.374 | 0.523 | 0.586 | 0.412 | 0.417 | 0.447 | 0.460 | 0.386 | 0.400 | 0.427 | 0.429 | 0.634 | 0.567 |
| | 336 | 0.321 | 0.371 | 0.388 | 0.415 | 0.640 | 0.637 | 0.441 | 0.444 | 0.515 | 0.505 | 0.414 | 0.428 | 0.449 | 0.451 | 0.655 | 0.588 |
| | 720 | 0.336 | 0.432 | 0.391 | 0.420 | 2.296 | 1.034 | 0.438 | 0.452 | 0.665 | 0.589 | 0.419 | 0.443 | 0.448 | 0.458 | 0.570 | 0.549 |
| | Avg | 0.303 | 0.368 | 0.353 | 0.387 | 0.992 | 0.708 | 0.406 | 0.422 | 0.493 | 0.488 | 0.380 | 0.405 | 0.421 | 0.431 | 0.582 | 0.548 |

# F    MORE EXPERIMENTAL RESULTS

Table 9 shows the complete experimental results of zero-shot forecasting task. In zero-shot inference, the model is first trained on the dataset A, and subsequently the prediction performance is tested on dataset B. Table 9 demonstrates the experimental results under the full prediction length 96, 192, 336, 720 settings, which show that the proposed UTSD has excellent generalization ability and robustness compared to existing methods.

 Table 10 demonstrates, the results of the multi-domain pretrained model after finetuning, where the model achieves a performance that exceeds the performance of the trained from scratch model on both ETTh2 and ETTm2 datasets. The experimental results illustrate that the adapter-based finetuning strategy fully utilise the potential representations learned from multiple data domains during the pretraining phase. Our finetuning strategy effectively activates the inference performance of pretrained models in downstream tasks, which provides implications for future work.

The Table 11 demonstrates the imputation results of training from scratch on each particular dataset, the average MSE is reduced by **17.0%**, **20.1%** and **38.7%** compared to the existing GPT4TS, TimesNet and PatchTST. Excitingly, the proposed UTSD achieves better overall results on the imputation task than existing multitasking

---

[3]https://archive.ics.uci.edu/dataset/321/electricity

[4]http://pems.dot.ca.gov

Table 10: Comparison of the performance of the proposed methods under training from scratch and fine-tuning settings.

| Metric | ETTh2 | | | | | | | | | | ETTm2 | | | | | | | | | |
|---|---|---|---|---|---|---|---|---|---|---|---|---|---|---|---|---|---|---|---|---|
| | 96 | | 192 | | 336 | | 720 | | avg | | 96 | | 192 | | 336 | | 720 | | avg | |
| | MSE | MAE | MSE | MAE | MSE | MAE | MSE | MAE | MSE | MAE | MSE | MAE | MSE | MAE | MSE | MAE | MSE | MAE | MSE | MAE |
| Scratch | 0.241 | 0.301 | 0.275 | 0.375 | 0.302 | 0.372 | 0.323 | 0.386 | 0.285 | 0.358 | 0.191 | 0.284 | 0.221 | 0.306 | 0.235 | 0.314 | 0.283 | 0.353 | 0.233 | 0.314 |
| Finetune | 0.212 | 0.261 | 0.250 | 0.304 | 0.274 | 0.323 | 0.295 | 0.347 | 0.258 | 0.309 | 0.187 | 0.244 | 0.213 | 0.290 | 0.222 | 0.268 | 0.263 | 0.297 | 0.221 | 0.275 |

Table 11: Comparison of the complete performance with diverse mask ratios on **full-data imputation** task.

| Method | | UTSD | | TimeLLM | | GPT4TS | | TimesNet | | LLMTime | | PatchTST | | DLinear | | FEDformer | | Stationary | | Autoformer | |
|---|---|---|---|---|---|---|---|---|---|---|---|---|---|---|---|---|---|---|---|---|---|
| | MaskRatio | MSE | MAE | MSE | MAE | MSE | MAE | MSE | MAE | MSE | MAE | MSE | MAE | MSE | MAE | MSE | MAE | MSE | MAE | MSE | MAE |
| ETTm1 | 12.5% | 0.019 | **0.077** | **0.017** | 0.085 | 0.023 | 0.101 | 0.041 | 0.130 | 0.096 | 0.229 | 0.093 | 0.206 | 0.080 | 0.193 | 0.052 | 0.166 | 0.032 | 0.119 | 0.046 | 0.144 |
| | 25% | 0.022 | **0.092** | **0.022** | 0.096 | 0.025 | 0.104 | 0.047 | 0.139 | 0.100 | 0.234 | 0.098 | 0.212 | 0.086 | 0.201 | 0.059 | 0.174 | 0.040 | 0.128 | 0.056 | 0.156 |
| | 37.5% | **0.028** | **0.110** | 0.029 | 0.111 | 0.029 | 0.111 | 0.049 | 0.143 | 0.133 | 0.271 | 0.113 | 0.231 | 0.103 | 0.219 | 0.069 | 0.191 | 0.039 | 0.131 | 0.057 | 0.161 |
| | 50% | **0.035** | **0.117** | 0.040 | 0.128 | 0.036 | 0.124 | 0.055 | 0.151 | 0.186 | 0.323 | 0.134 | 0.255 | 0.132 | 0.248 | 0.089 | 0.218 | 0.047 | 0.145 | 0.067 | 0.174 |
| | Avg | **0.026** | **0.099** | 0.028 | 0.105 | 0.027 | 0.107 | 0.047 | 0.140 | 0.120 | 0.253 | 0.104 | 0.218 | 0.093 | 0.206 | 0.062 | 0.177 | 0.036 | 0.126 | 0.051 | 0.150 |
| ETTm2 | 12.5% | 0.018 | 0.079 | **0.017** | **0.076** | 0.018 | 0.080 | 0.026 | 0.094 | 0.108 | 0.239 | 0.034 | 0.127 | 0.062 | 0.166 | 0.056 | 0.159 | 0.021 | 0.088 | 0.023 | 0.092 |
| | 25% | **0.019** | 0.082 | 0.020 | **0.080** | 0.020 | 0.085 | 0.028 | 0.099 | 0.164 | 0.294 | 0.042 | 0.143 | 0.085 | 0.196 | 0.080 | 0.195 | 0.024 | 0.096 | 0.026 | 0.101 |
| | 37.5% | **0.021** | **0.085** | 0.022 | 0.087 | 0.023 | 0.091 | 0.030 | 0.104 | 0.237 | 0.356 | 0.051 | 0.159 | 0.106 | 0.222 | 0.110 | 0.231 | 0.027 | 0.103 | 0.030 | 0.108 |
| | 50% | **0.024** | **0.094** | 0.025 | 0.095 | 0.026 | 0.098 | 0.034 | 0.110 | 0.323 | 0.421 | 0.059 | 0.174 | 0.131 | 0.247 | 0.156 | 0.276 | 0.030 | 0.108 | 0.035 | 0.119 |
| | Avg | **0.020** | 0.085 | 0.021 | **0.084** | 0.022 | 0.088 | 0.029 | 0.102 | 0.208 | 0.327 | 0.046 | 0.151 | 0.096 | 0.208 | 0.101 | 0.215 | 0.026 | 0.099 | 0.029 | 0.105 |
| ETTh1 | 12.5% | **0.040** | **0.137** | 0.043 | 0.140 | 0.057 | 0.159 | 0.093 | 0.201 | 0.126 | 0.263 | 0.240 | 0.345 | 0.151 | 0.267 | 0.070 | 0.190 | 0.060 | 0.165 | 0.074 | 0.182 |
| | 25% | **0.053** | **0.155** | 0.054 | 0.156 | 0.069 | 0.178 | 0.107 | 0.217 | 0.169 | 0.304 | 0.265 | 0.364 | 0.180 | 0.292 | 0.106 | 0.236 | 0.080 | 0.189 | 0.090 | 0.203 |
| | 37.5% | **0.070** | **0.175** | 0.072 | 0.180 | 0.084 | 0.196 | 0.120 | 0.230 | 0.220 | 0.347 | 0.296 | 0.382 | 0.215 | 0.318 | 0.124 | 0.258 | 0.102 | 0.212 | 0.109 | 0.222 |
| | 50% | **0.093** | **0.202** | 0.107 | 0.216 | 0.102 | 0.215 | 0.141 | 0.248 | 0.293 | 0.402 | 0.334 | 0.404 | 0.257 | 0.347 | 0.165 | 0.299 | 0.133 | 0.240 | 0.137 | 0.248 |
| | Avg | **0.064** | **0.167** | 0.069 | 0.173 | 0.078 | 0.187 | 0.115 | 0.224 | 0.202 | 0.329 | 0.284 | 0.373 | 0.201 | 0.306 | 0.117 | 0.246 | 0.094 | 0.201 | 0.103 | 0.214 |
| ETTh2 | 12.5% | 0.040 | **0.124** | **0.039** | 0.125 | 0.040 | 0.130 | 0.057 | 0.152 | 0.187 | 0.319 | 0.101 | 0.231 | 0.100 | 0.216 | 0.095 | 0.212 | 0.042 | 0.133 | 0.044 | 0.138 |
| | 25% | **0.043** | **0.131** | 0.044 | 0.135 | 0.046 | 0.141 | 0.061 | 0.158 | 0.279 | 0.390 | 0.115 | 0.246 | 0.127 | 0.247 | 0.137 | 0.258 | 0.049 | 0.147 | 0.050 | 0.149 |
| | 37.5% | **0.049** | **0.143** | 0.051 | 0.147 | 0.052 | 0.151 | 0.067 | 0.166 | 0.400 | 0.465 | 0.126 | 0.257 | 0.158 | 0.276 | 0.187 | 0.304 | 0.056 | 0.158 | 0.060 | 0.163 |
| | 50% | **0.053** | **0.155** | 0.059 | 0.158 | 0.060 | 0.162 | 0.073 | 0.174 | 0.602 | 0.572 | 0.136 | 0.268 | 0.183 | 0.299 | 0.232 | 0.341 | 0.065 | 0.170 | 0.068 | 0.173 |
| | Avg | **0.047** | **0.138** | 0.048 | 0.141 | 0.049 | 0.146 | 0.065 | 0.163 | 0.367 | 0.436 | 0.119 | 0.250 | 0.142 | 0.259 | 0.163 | 0.279 | 0.053 | 0.152 | 0.055 | 0.156 |
| Electricity | 12.5% | **0.043** | **0.129** | 0.080 | 0.194 | 0.085 | 0.202 | 0.055 | 0.160 | 0.196 | 0.321 | 0.102 | 0.229 | 0.092 | 0.214 | 0.107 | 0.237 | 0.093 | 0.210 | 0.089 | 0.210 |
| | 25% | **0.049** | **0.142** | 0.087 | 0.203 | 0.089 | 0.206 | 0.065 | 0.175 | 0.207 | 0.332 | 0.121 | 0.252 | 0.118 | 0.247 | 0.120 | 0.251 | 0.097 | 0.214 | 0.096 | 0.220 |
| | 37.5% | **0.056** | **0.151** | 0.094 | 0.211 | 0.094 | 0.213 | 0.076 | 0.189 | 0.219 | 0.344 | 0.141 | 0.273 | 0.144 | 0.276 | 0.136 | 0.266 | 0.102 | 0.220 | 0.104 | 0.229 |
| | 50% | **0.065** | **0.165** | 0.101 | 0.220 | 0.100 | 0.221 | 0.091 | 0.208 | 0.235 | 0.357 | 0.160 | 0.293 | 0.175 | 0.305 | 0.158 | 0.284 | 0.108 | 0.228 | 0.113 | 0.239 |
| | Avg | **0.053** | **0.147** | 0.090 | 0.207 | 0.092 | 0.210 | 0.072 | 0.183 | 0.214 | 0.339 | 0.131 | 0.262 | 0.132 | 0.260 | 0.130 | 0.259 | 0.100 | 0.218 | 0.101 | 0.225 |
| Weather | 12.5% | **0.024** | **0.040** | 0.026 | 0.049 | 0.025 | 0.045 | 0.029 | 0.049 | 0.057 | 0.141 | 0.047 | 0.101 | 0.039 | 0.084 | 0.041 | 0.107 | 0.027 | 0.051 | 0.026 | 0.047 |
| | 25% | **0.026** | **0.043** | 0.028 | 0.052 | 0.029 | 0.052 | 0.031 | 0.053 | 0.065 | 0.155 | 0.052 | 0.111 | 0.048 | 0.103 | 0.064 | 0.163 | 0.029 | 0.056 | 0.030 | 0.054 |
| | 37.5% | **0.030** | **0.047** | 0.033 | 0.060 | 0.031 | 0.057 | 0.035 | 0.058 | 0.081 | 0.180 | 0.058 | 0.121 | 0.057 | 0.117 | 0.107 | 0.229 | 0.033 | 0.062 | 0.032 | 0.060 |
| | 50% | **0.033** | **0.052** | 0.037 | 0.065 | 0.034 | 0.062 | 0.038 | 0.063 | 0.102 | 0.207 | 0.065 | 0.133 | 0.066 | 0.134 | 0.183 | 0.312 | 0.037 | 0.068 | 0.037 | 0.067 |
| | Avg | **0.028** | **0.046** | 0.031 | 0.056 | 0.030 | 0.054 | 0.060 | 0.144 | 0.076 | 0.171 | 0.055 | 0.117 | 0.052 | 0.110 | 0.099 | 0.203 | 0.032 | 0.059 | 0.031 | 0.057 |
| 1st Count | | **53** | | 7 | | 0 | | 0 | | 0 | | 0 | | 0 | | 0 | | 0 | | 0 | |

foundation and specific models. Surprisingly, the proposed UTSD shows better results on the dataset characterized by multi-periodic patterns, which conforms to the multi-scale representation mechanism designed in our Condition-Denoising component. Specifically,the average MSE is reduced by **41.1%**, **42.3%** and **26.3%** compared to the existing GPT4TS, TimesNet and PatchTST on the ECL dataset.

Table 12: To further verify the comprehensive performance of the proposed UTSD in Long-term Time-series Generation, we introduce additional evaluation metrics: Context-FID Score, Correlational Score, Discriminative Score, Predictive Score. We boldface the best performance on all metrics and datasets, respectively.

| | Dataset | Length | UTSD | Diffusion-TS | TimeGAN | TimeVAE | Diffwave | DiffTime | Cot-GAN | Improve(%) |
|---|---|---|---|---|---|---|---|---|---|---|
| ETTh | Context-FID | 64 | **0.522±.031** | 0.631±.058 | 1.130±.102 | 0.827±.146 | 1.543±.153 | 1.279±.083 | 3.008±.277 | |
| | Score | 128 | **0.633±.029** | 0.787±.062 | 1.553±.169 | 1.062±.134 | 2.354±.170 | 2.554±.318 | 2.639±.427 | 18.2 |
| | (Lower the Better) | 256 | **0.347±.010** | 0.423±.038 | 5.872±.208 | 0.826±.093 | 2.899±.289 | 3.524±.830 | 4.075±.894 | |
| | Correlational | 64 | **0.070±.002** | 0.082±.005 | 0.483±.019 | 0.067±.006 | 0.186±.008 | 0.094±.010 | 0.271±.007 | |
| | Score | 128 | **0.072±.002** | 0.088±.005 | 0.188±.006 | 0.054±.007 | 0.203±.006 | 0.113±.012 | 0.176±.006 | 16.2 |
| | (Lower the Better) | 256 | **0.054±.003** | 0.064±.007 | 0.522±.013 | 0.046±.007 | 0.199±.003 | 0.135±.006 | 0.222±.010 | |
| | Discriminative | 64 | **0.087±.017** | 0.106±.048 | 0.227±.078 | 0.171±.142 | 0.254±.074 | 0.150±.003 | 0.296±.348 | |
| | Score | 128 | **0.120±.023** | 0.144±.060 | 0.188±.074 | 0.154±.087 | 0.274±.047 | 0.176±.015 | 0.451±.080 | 18.5 |
| | (Lower the Better) | 256 | **0.048±.011** | 0.060±.030 | 0.442±.056 | 0.178±.076 | 0.304±.068 | 0.243±.005 | 0.461±.010 | |
| | Predictive | 64 | **0.098±.003** | 0.116±.000 | 0.132±.008 | 0.118±.004 | 0.133±.008 | 0.118±.004 | 0.135±.003 | |
| | Score | 128 | **0.087±.003** | 0.110±.003 | 0.153±.014 | 0.113±.005 | 0.129±.003 | 0.120±.008 | 0.126±.001 | 17.8 |
| | (Lower the Better) | 256 | **0.090±.006** | 0.109±.013 | 0.220±.008 | 0.110±.027 | 0.132±.001 | 0.118±.003 | 0.129±.000 | |
| Energy | Context-FID | 64 | 0.136±.014 | **0.135±.017** | 1.230±.070 | 2.662±.087 | 2.697±.418 | 0.762±.157 | 1.824±.144 | |
| | Score | 128 | **0.084±.015** | 0.087±.019 | 2.535±.372 | 3.125±.106 | 5.552±.528 | 1.344±.131 | 1.822±.271 | 2.2 |
| | (Lower the Better) | 256 | **0.122±.019** | 0.126±.024 | 5.032±.831 | 3.768±.998 | 5.572±.584 | 4.735±.729 | 2.533±.467 | |
| | Correlational | 64 | **0.618±.012** | 0.672±.035 | 3.668±.106 | 1.653±.208 | 6.847±.083 | 1.281±.218 | 3.319±.062 | |
| | Score | 128 | **0.426±.031** | 0.451±.079 | 4.790±.116 | 1.820±.329 | 6.663±.112 | 1.376±.201 | 3.713±.055 | 6.4 |
| | (Lower the Better) | 256 | **0.341±.039** | 0.361±.092 | 4.487±.214 | 1.279±.114 | 5.690±.102 | 1.800±.138 | 3.739±.089 | |
| | Discriminative | 64 | **0.066±.005** | 0.078±.021 | 0.498±.001 | 0.499±.000 | 0.497±.004 | 0.328±.031 | 0.499±.001 | |
| | Score | 128 | **0.127±.038** | 0.143±.075 | 0.499±.001 | 0.499±.000 | 0.499±.001 | 0.396±.024 | 0.499±.001 | 13.2 |
| | (Lower the Better) | 256 | **0.252±.047** | 0.290±.123 | 0.499±.000 | 0.499±.000 | 0.499±.000 | 0.437±.095 | 0.498±.004 | |
| | Predictive | 64 | **0.225±.001** | 0.249±.000 | 0.291±.003 | 0.302±.001 | 0.252±.001 | 0.252±.000 | 0.262±.002 | |
| | Score | 128 | **0.221±.001** | 0.247±.001 | 0.303±.002 | 0.318±.000 | 0.252±.000 | 0.251±.000 | 0.269±.002 | 11.0 |
| | (Lower the Better) | 256 | **0.214±.001** | 0.245±.001 | 0.351±.004 | 0.353±.003 | 0.251±.000 | 0.251±.000 | 0.275±.004 | |

The Table 12 demonstrates the generation results of training from scratch on each particular dataset. The generation metrics Context-FID Score, Correlational Score, Discriminative Score and Predictive Score is reduced by **18.2%**, **16.2%**, **18.5%** and **17.8%** on ETTh dataset compared to the existing DiffusionTS. Besides, compared with DiffusionTS, the average improvement of all indicators of UTSD on ETTh and Energy datasets is **17.6%** and **8.2%**, respectively.

## G  MORE VISUALIZATIONS

In Figure 12, to validate the generative power of the diffusion-based probabilistic model, we visualise the generation results of UTSD and the pre-existing diffusion method on the same dataset using the t-SNE method. The red colour represents the real sequence samples and the blue colour represents the generated dataset, where the degree of aggregation of these two samples in two-dimensional space reflects the generative ability of the model. Specifically, when the projections of the two samples in 2D space are fully aggregated, the model exhibits excellent generative performance.

The Figure 13 illustrates the dithering issue of popular diffusion model on the temporal generation task, as one of the challenges in building a unified times series generation model. The existing CSDI (Tashiro et al., 2021) and TimeGrad (Rasul et al., 2021) try to improve the prediction accuracy by averaging the results of multiple samples, however, this leads to a huge time overhead. In contrast, the proposed UTSD can generate high-quality prediction sequences with only one sampling. Visualization of the prediction results obtained by repeated sampling of the four probabilistic models. The same observation sequence is fixed, and each model repeats the inference 50 times. tSNE is utilized to visualize all the predicted sequences. UTSD has the highest stability and accuracy.

To validate the effectiveness of the proposed model architecture, we visualised experiments against condition net ablation, as shown in Figure 14. For each dataset, the generative capacity of the full UTSD model is

shown on the left, and the generative capacity of the model when the condition net structure is excluded is shown on the right, where the true values are in blue and the generated samples are in yellow. For each experiment, the model demonstrated the best generative performance when the generated samples and the true values completely overlapped together.

To visualise the performance of the model on the Scratch prediction task, Figures 15—20 show the results on several datasets. Specifically, the upper half shows the prediction results obtained from the probabilistic model with 50 repetitive samples, where the light and dark green colours denote the prediction results for the $10-90\%$ and $25-75\%$ confidence intervals, respectively (50 repetitive samples for each model), and the blue and green curves denote the true value and median prediction results, respectively. The lower half of the display shows shows the prediction results of the deep regression model (UTSD samples only once), where the blue and red curves indicate the ground truth and prediction results, respectively.

Regarding long-term multi-periodic series and short-term non-periodic series have been a great challenge for time series forecasting. Therefore it is considered important to introduce more visualisation results as shown in Figures 10 and 11. Figure 10 illustrates the real long-periodic series **(timesteps equals to 720)** sampled from ETT, ECL and Traffic datasets and Figure 11 illustrates the real short-term non-periodic series **(timesteps equals to 96)** sampled from ETT, ECL and Traffic datasets. Specifically, UTSD demonstrates excellent performance in long-term series forecasting, which verifies that UTSD has the ability to capture long-term dependencies, which is crucial for practical applications. In addition, UTSD likewise exhibits satisfactory prediction results in short-term sequences, which demonstrates the ability of the diffusion-based forecasting model to generate high-quality time-series samples.

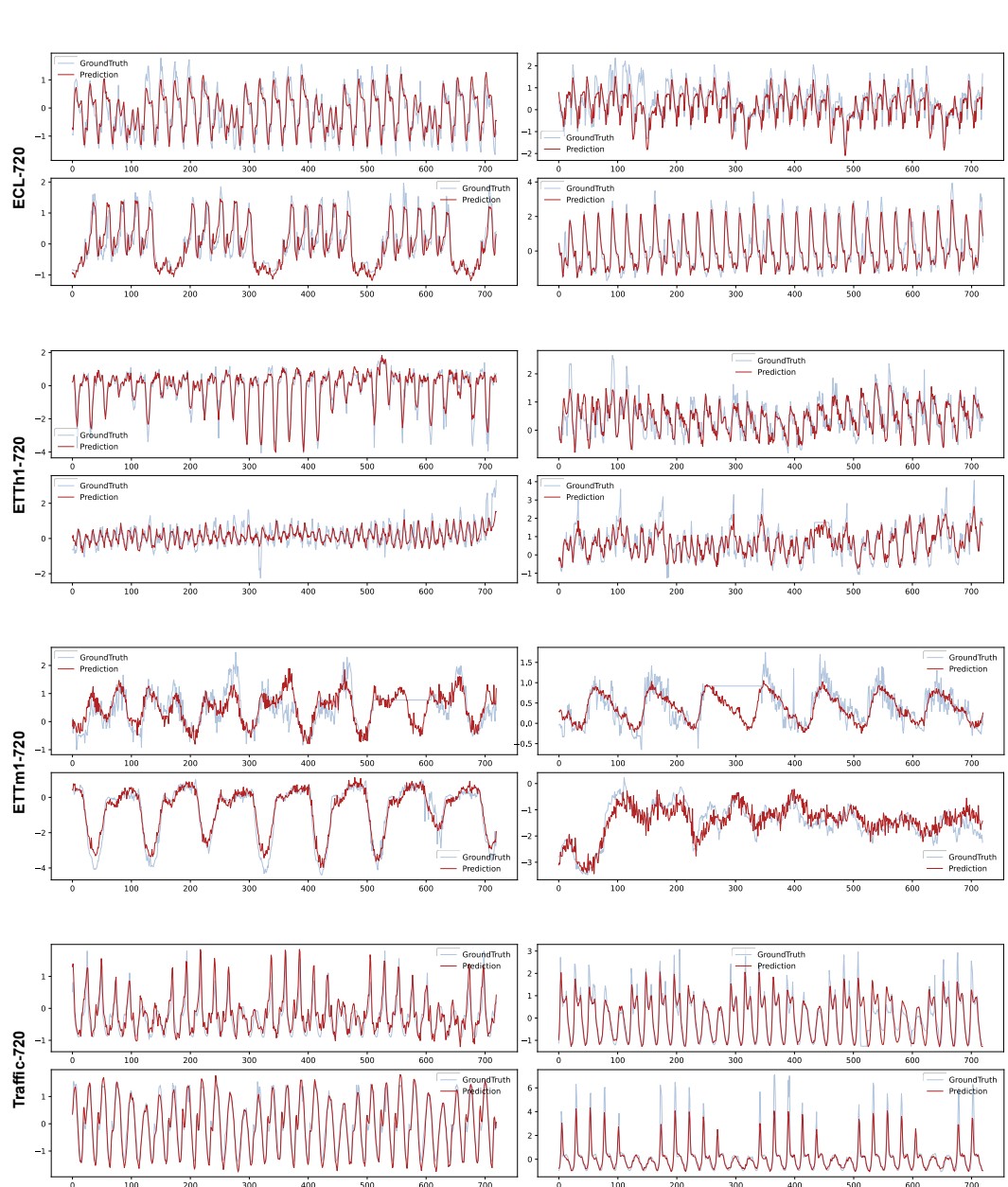

Figure 10: Demonstration of UTSD prediction results on real **long-term multi-periodic** sequences sampled from ETT, ECL and Traffic datasets.

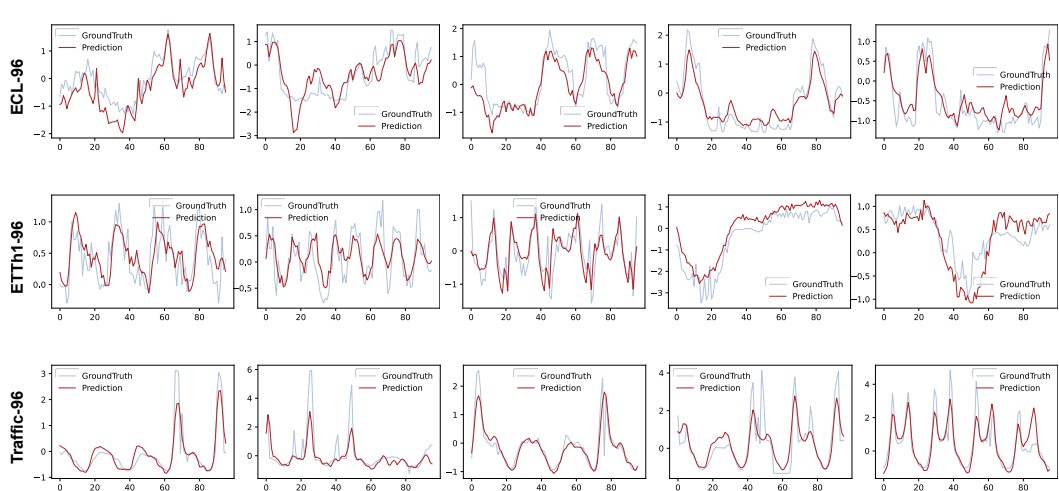

Figure 11: Demonstration of UTSD prediction results on real **short-term non-periodic** sequences sampled from ETT, ECL and Traffic datasets.

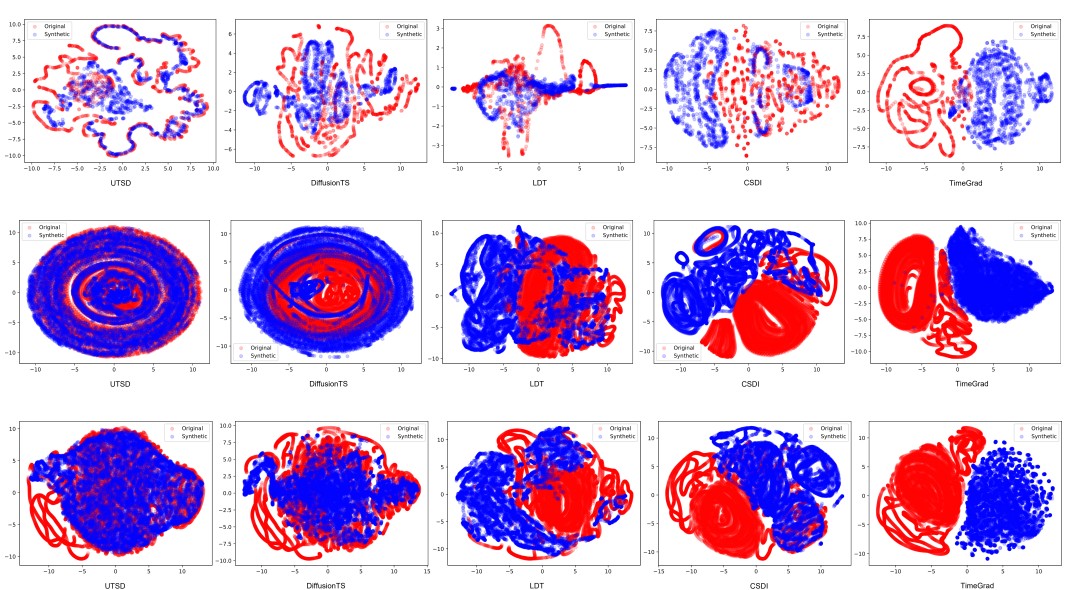

Figure 12: Visualization of comparisons between UTSD and exsting probabilistic and deep model baselines on the ETTh (Upper) and ETTm (Bottom) dataset.

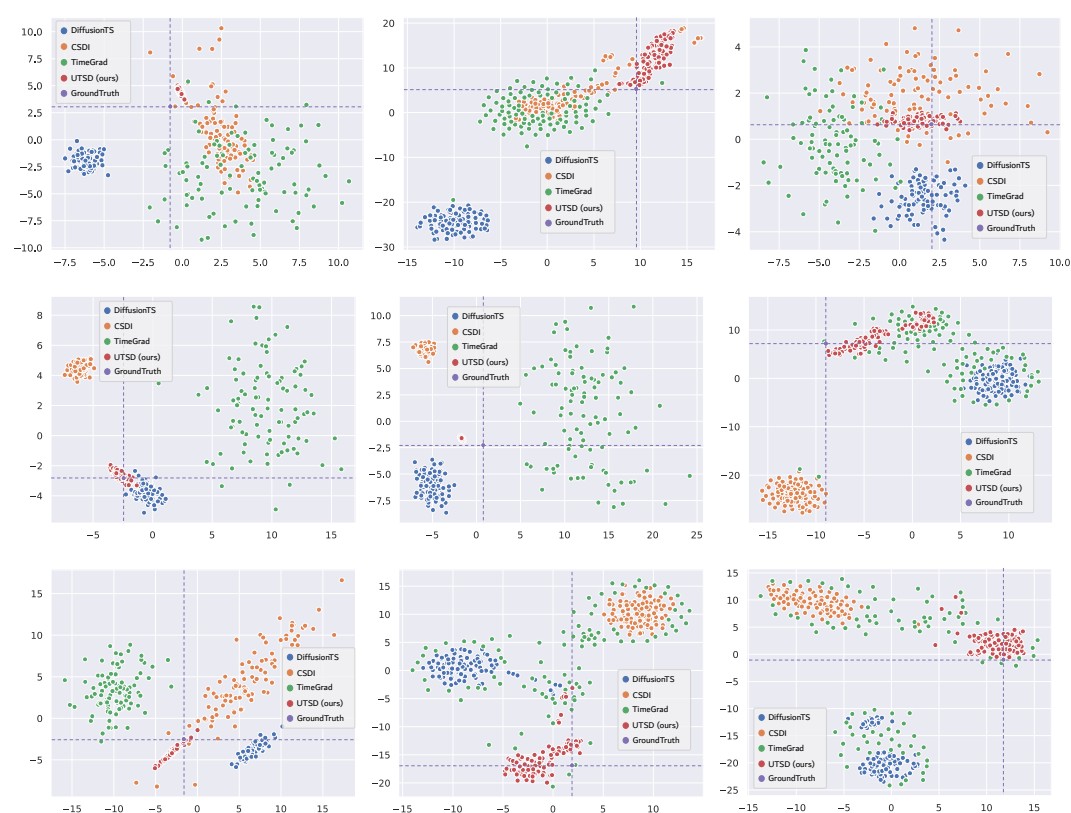

Figure 13: Illustration of the dithering issue with popular diffusion models for time series generation tasks, which is one of the challenges in building a unified time series generation model. With the fixed observation sequence and groundtruth, each fully trained diffusion model is repeatedly sampled 50 times. All generated sequences and the groundtruth are projected into a two-dimensional space, by the t-SNE approach. The visualisation results demonstrate the excellent stability and accuracy of the proposed method.

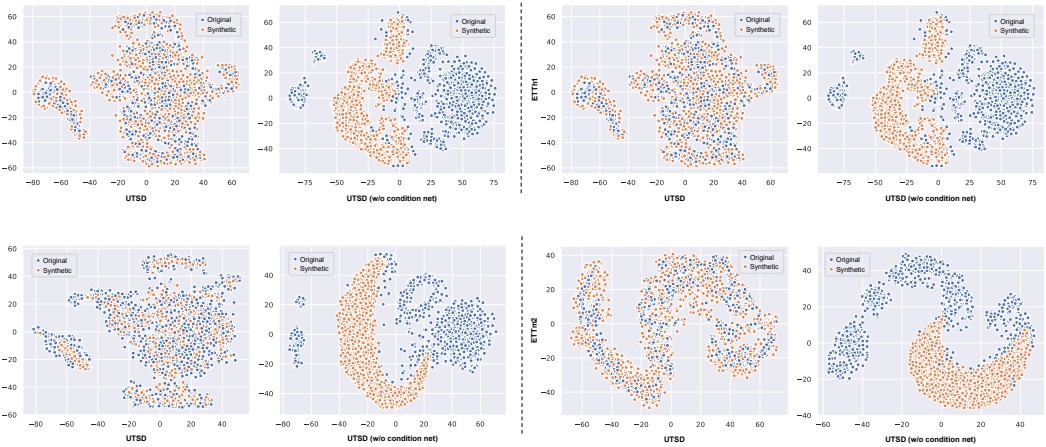

Figure 14: Visualisation on the validity of the proposed model architecture condition net. Where the upper part shows the visualisation results on the ETTh1 and ETTh2 datasets, and the bottom part shows the visualisation results on the ETTm1 and ETTm2 datasets.

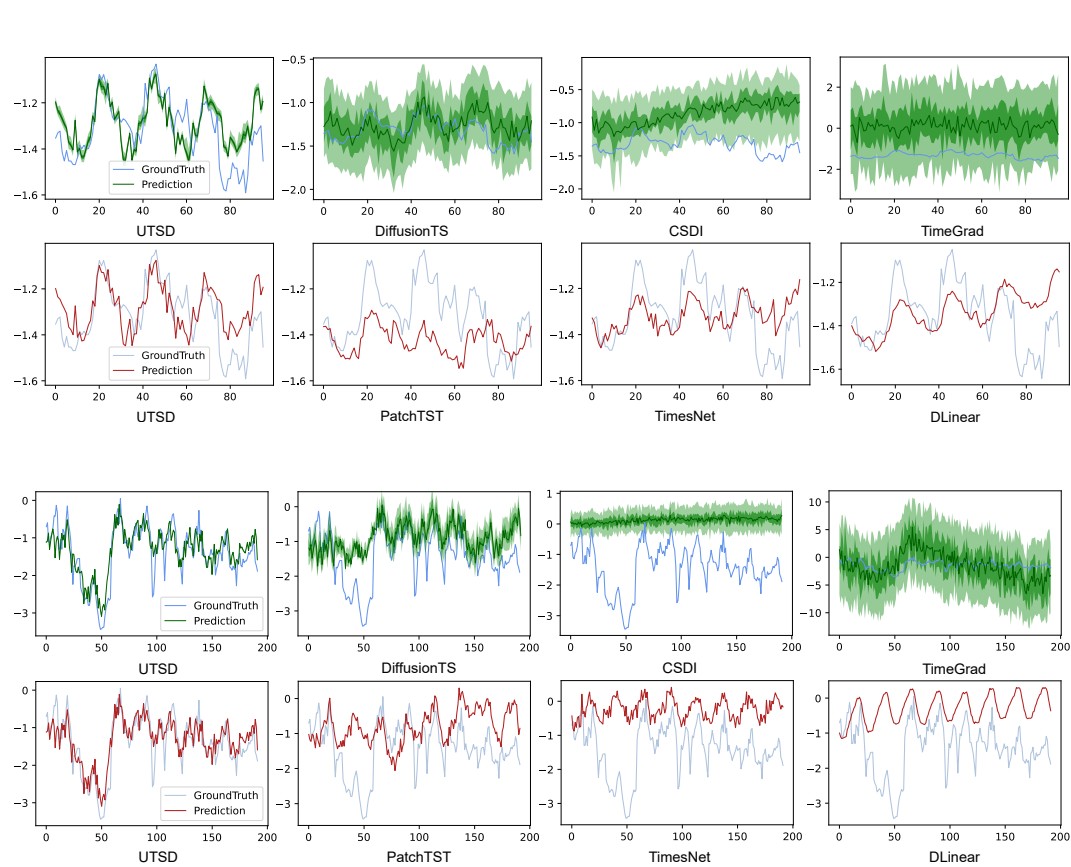

Figure 15: Visualization of comparisons between UTSD and exsting probabilistic (upper) and deep model (bottom) baselines on the **ETTh1** dataset. Where the light blue curve represents the groundtruth, and the green and red curves represent the prediction results of several baselines. The light and dark green staining show the predictions of the probabilistic model at the $10 - 90\%$ and $25 - 75\%$ confidence intervals, respectively.

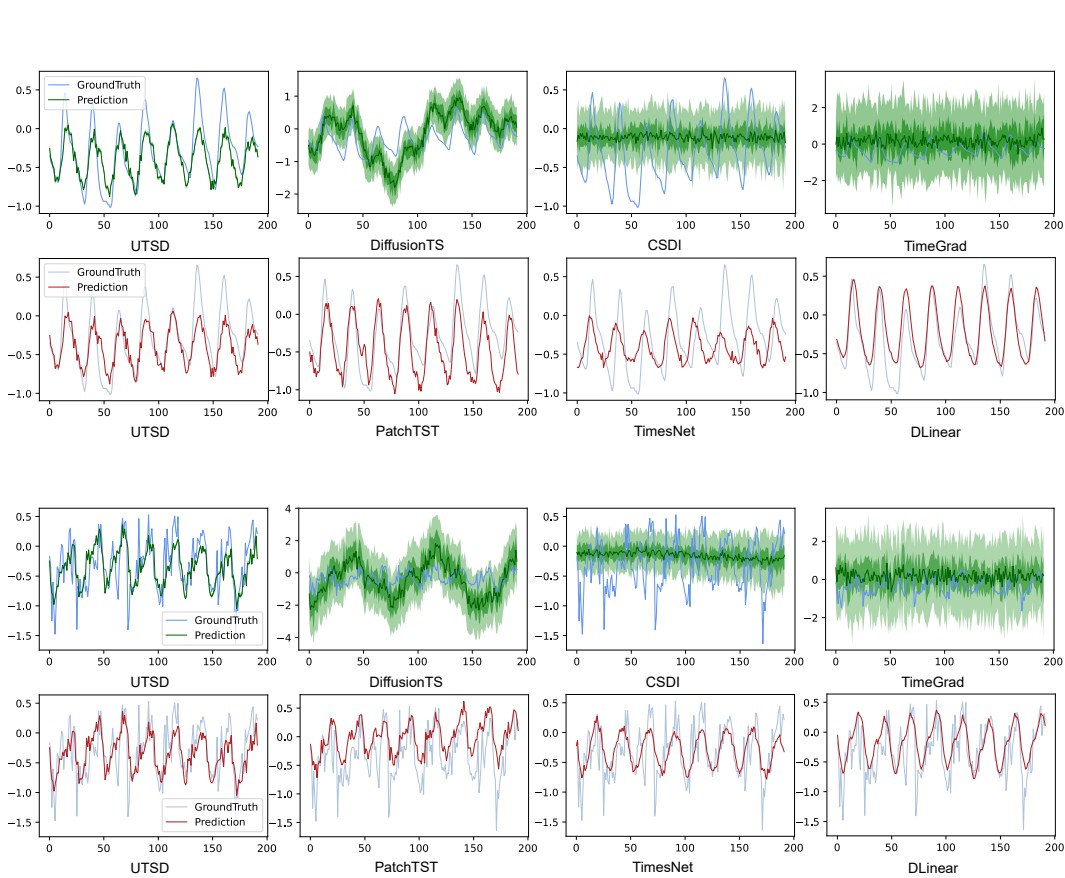

Figure 16: Visualization of comparisons between UTSD and exsting probabilistic (upper) and deep model (bottom) baselines on the **ETTh2** dataset. Where the light blue curve represents the groundtruth, and the green and red curves represent the prediction results of several baselines. The light and dark green staining show the predictions of the probabilistic model at the $10 - 90\%$ and $25 - 75\%$ confidence intervals, respectively.

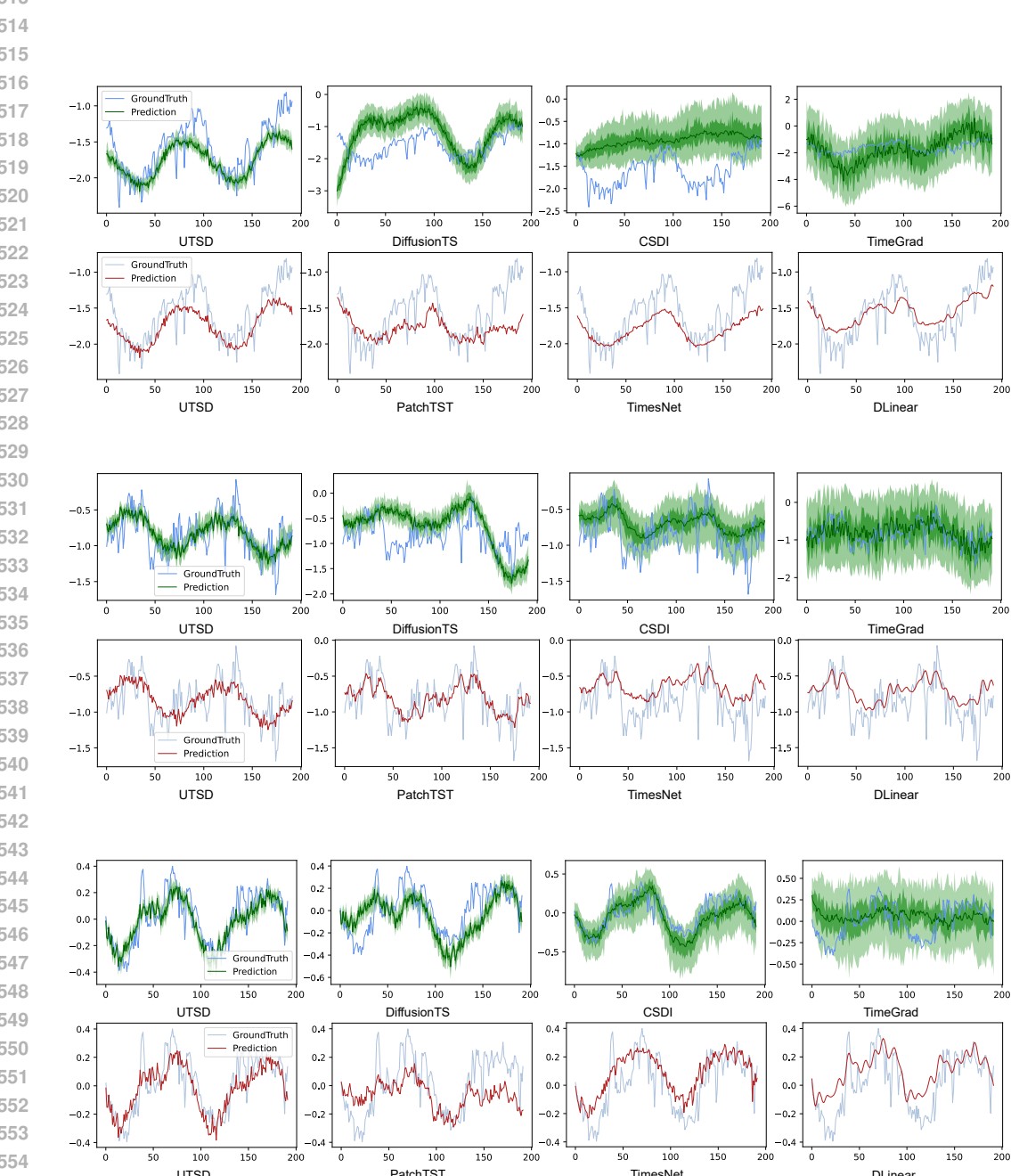

Figure 17: Visualization of comparisons between UTSD and exsting probabilistic (upper) and deep model (bottom) baselines on the **ETTm1** and **ETTm2** dataset. Where the light blue curve represents the groundtruth, and the green and red curves represent the prediction results of several baselines. The light and dark green staining show the predictions of the probabilistic model at the $10 - 90\%$ and $25 - 75\%$ confidence intervals, respectively.

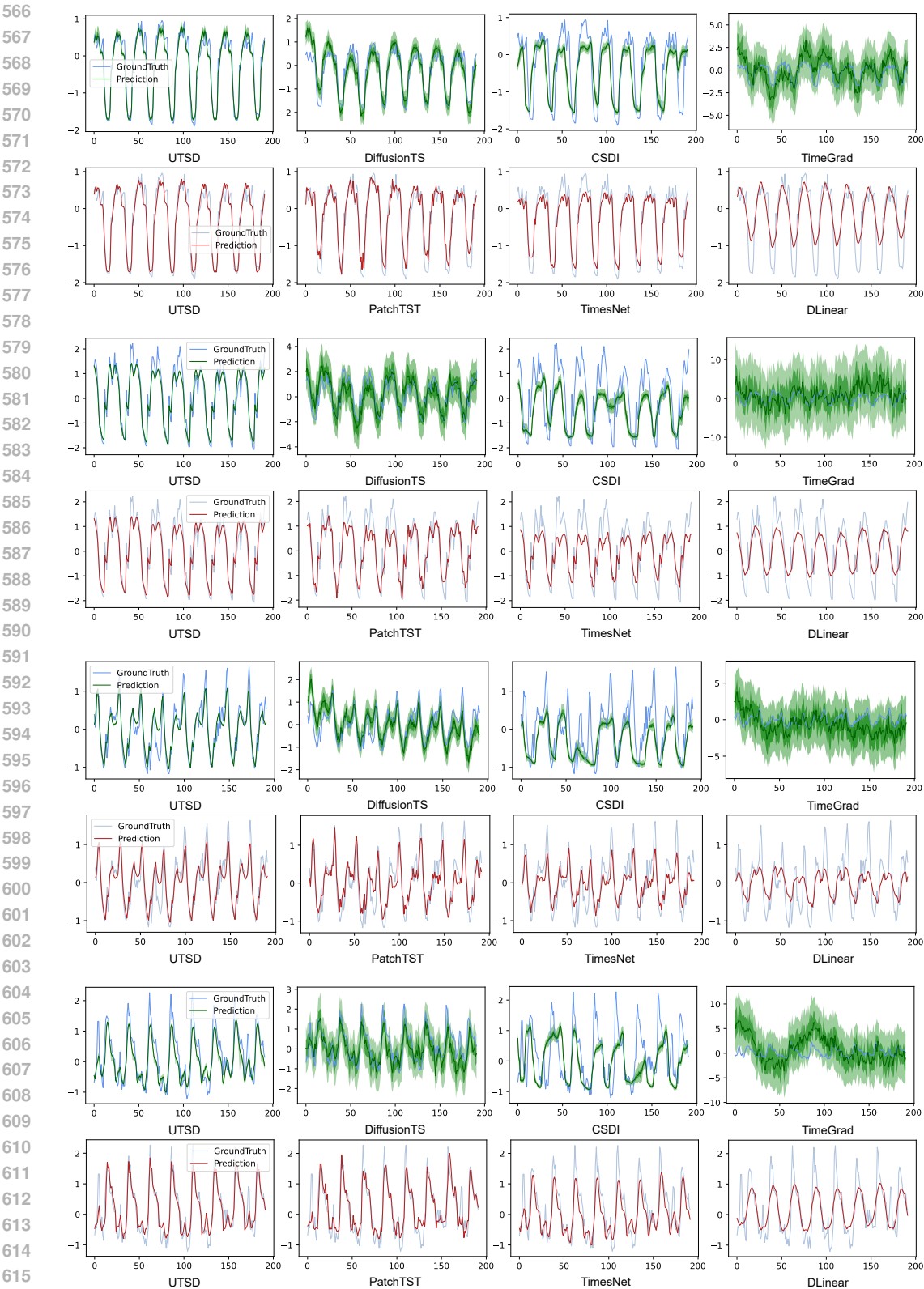

Figure 18: Visualization of comparisons between UTSD and exsting probabilistic (upper) and deep model (bottom) baselines on the **ECL** dataset.

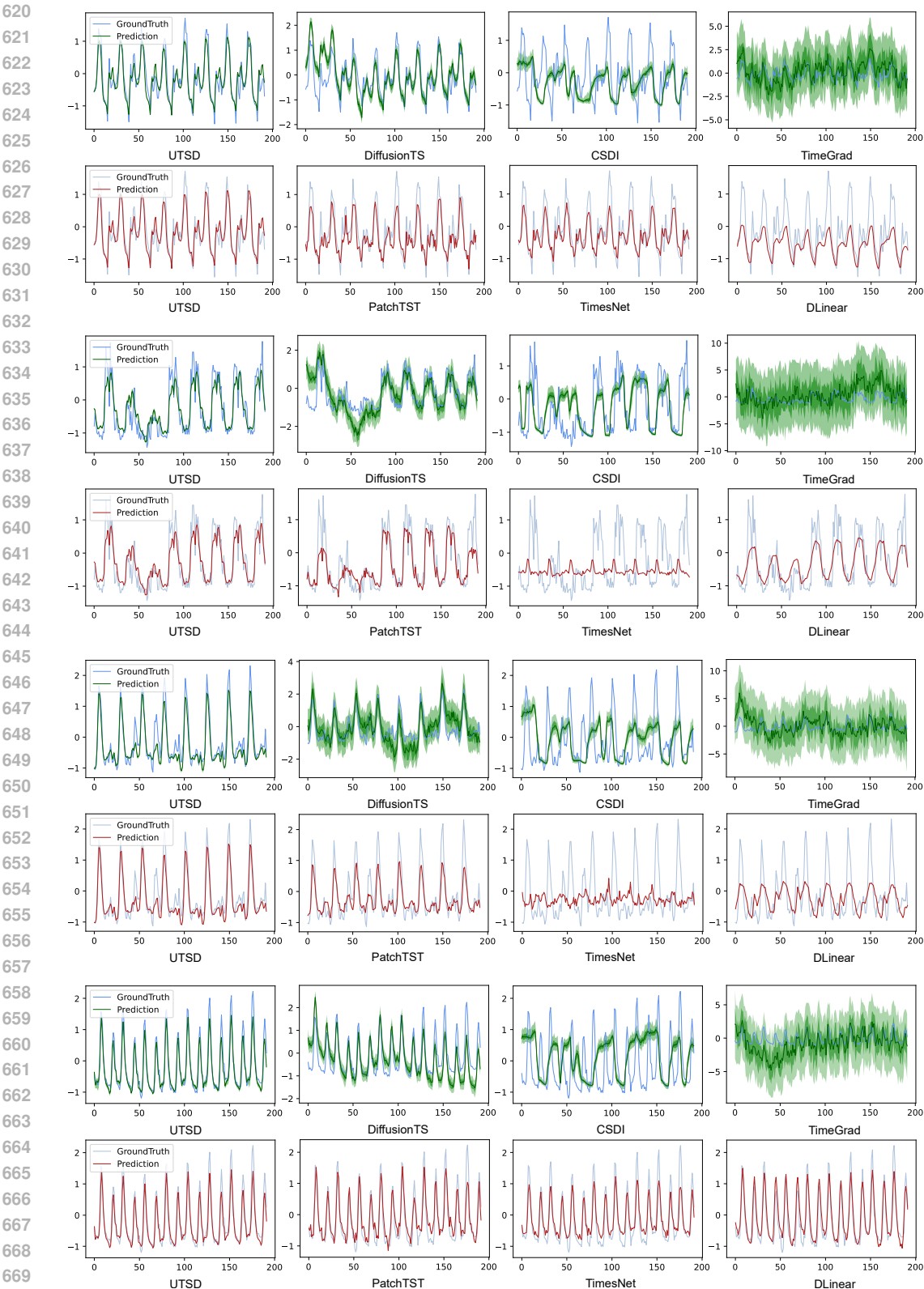

Figure 19: Visualization of comparisons between UTSD and exsting probabilistic (upper) and deep model (bottom) baselines on the **ECL** dataset.

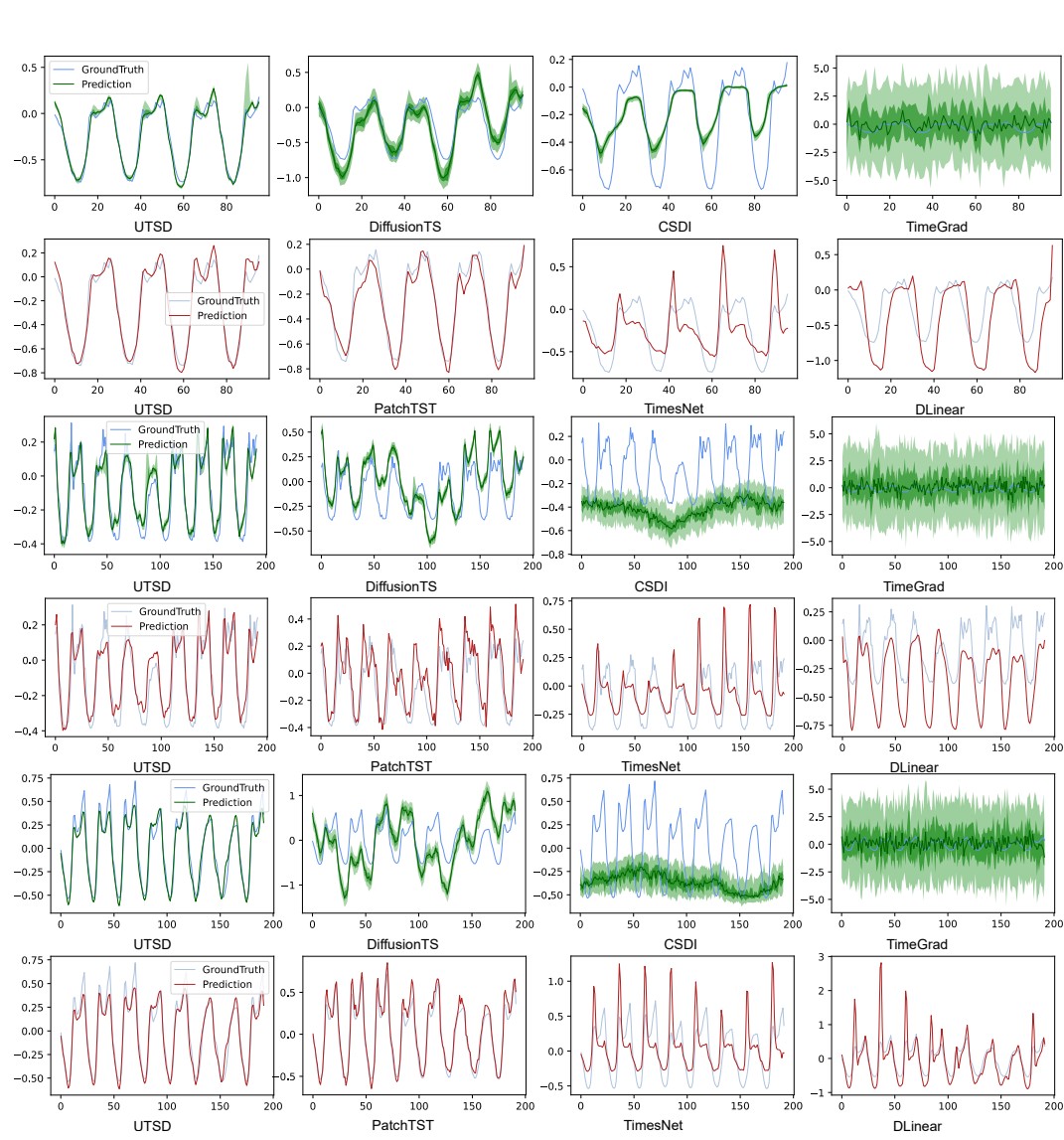

Figure 20: Visualization of comparisons between UTSD and exsting probabilistic (upper) and deep model (bottom) baselines on the **ECL** dataset.

