# OpenReview forum: "UTSD: Unified Time Series Diffusion Model"
_ICLR.cc/2025/Conference — Submitted to ICLR 2025_

### Official Review · Reviewer_eVfc · 2024-11-01

**Soundness:** 2
**Presentation:** 2
**Contribution:** 1
**Rating:** 5
**Confidence:** 3

**Summary:**

This paper proposes a time series diffusion model for time series forecasting in multi-domain settings. The proposed model consists of the condition net, adapter, and denoising net. Different from prior autoregressive models, a diffusion denoising process is applied to capture the mixture distribution of cross-domain data and generate predictions directly. Experimental results show the effectiveness of the proposed method on probabilistic time series forecasting tasks.

**Strengths:**

According to the authors' claim, this is the first work to incorporate diffusion models into time series foundation models.

**Weaknesses:**

- The paper is generally not well-written. There are many typos and formatting issues.

- Example of typos: adaptor in the abstract

- Example of formatting issues: Text size is not consistent in the abstract, and citep is not used.

- There are false statements when criticizing prior works to motivate the proposed method. For example, while UniTime [Liu et al., 2024a] and Moirai [Woo et al., 2024] are non-autoregressive, authors claim that they "rely on an auto-regressive mechanism" in L085-088.

- Different from the statement around key contributions, the idea of denoising with condition has already been studied in many prior works, including [Fan et al.], [Shen et al., 2024], [Yuan and Qiao, 2024].

[Fan et al.] MG-TSD: Multi-Granularity Time Series Diffusion Models with Guided Learning Process. In ICLR, 2024.

- L114-116: "each iteration of the denoising stage causes an accumulation of errors in the latent space, which are further amplified during the alignment of the latent space to the actual sequence space." -> Is there any evidence on this statement?

- Not enough justification on the effectiveness of the proposed components. The ablation study in Table 5 is insufficient to explain why they should work.

- L177-179: Any supporting theory or experiments on the inherent drawbacks stated here?

- While it is natural to consider large-scale datasets for "unified" time series diffusion model, the datasets used in this paper is limited to small single-task datasets.

- Average performance should not be counted when ranking methods.

- Please distinguish citet and citep.

- Zero-shot forecasting is done in an unusual way; while the source and target datasets are similar ETT variations in this work, they usually have a significant gap in the standard zero-shot forecasting settings.

- Table 3 caption does not make sense.

- In Table 5, it is not clear what happens if condition net is missing, which implies that no input is given.

**Questions:**

Please address concerns above.

---
I appreciate the response from authors, they addressed some of my concerns. I raised my rating accordingly.

---

> ### Author Response · Authors · 2024-11-20
> **To Reviewer eVfc**
>
> Thank you for your thoughtful review and valuable feedback. We appreciate recognition of our innovative work in incorporating diffusion models into time series foundation models. We response your concerns as following.
>
> >Q1-Q3: Typos and formatting issues.
>
> **A:** We appreciate your feedback regarding the typos and formatting issues. We have corrected the spelling error in the abstract, ensured consistent text size throughout, and properly formatted all citations. We have carefully reviewed the entire manuscript to enhance the overall writing quality. Thank you for your valuable input.
>
> >Q4: There are false statements when criticizing prior works to motivate the proposed method. For example, while UniTime [Liu et al., 2024a] and Moirai [Woo et al., 2024] are non-autoregressive, authors claim that they "rely on an auto-regressive mechanism" in L085-088.
>
> **A4:** We apologize for the confusion in our original manuscript. We have clarified the discussion of prior works as follows:
>
> Specifically, the existing multi-domain generalization methods (e.g., Timer) rely on autoregressive mechanisms to establish connections between observed and predicted sequences. These methods are susceptible to error accumulation in long sequence prediction and often struggle to capture cross-domain relationships. In contrast, other models, such as UniTime and Moirai, endeavor to directly establish the projection between observation and forecast sequences. However, sequences from different domains differ significantly, and the projection relationships learned during training are not flexible enough to be directly applied to unseen domains. This limits the zero-shot and cross-domain generalization capabilities of the model.
>
> We have highlighted this clarification in blue in the revised manuscript. Thank you for your insightful comments.

---

> ### Author Response · Authors · 2024-11-20
> **To Reviewer eVfc**
>
> >Q5: Different from the statement around key contributions, the idea of denoising with condition has already been studied in many prior works, including [Fan et al.], [Shen et al., 2024], [Yuan and Qiao, 2024]. [Fan et al.] MG-TSD: Multi-Granularity Time Series Diffusion Models with Guided Learning Process. In ICLR, 2024.
>
> **A5:** Thank you for your insightful comments. We appreciate the opportunity to clarify the uniqueness of our work compared to existing studies like MG-TSD and Dish-TS. While prior works often focus on domain-specific models tailored to individual datasets, limiting their generalization, our approach introduces a unified time series diffusion model designed to excel across diverse and unseen domains. Here's a summary of our contributions and innovations:
>
> 1. **Unified Time Series Diffusion (UTSD) Model:** For the first time, we model multi-domain probability distributions using diffusion's robust capabilities. Unlike autoregressive models, our approach employs a diffusion denoising process to handle cross-domain data, generating predictions directly through conditional sampling.
> 2. **Superior Generalization:** The pre-trained UTSD surpasses existing foundation models across various data domains, demonstrating exceptional zero-shot generalization. It matches domain-specific models when trained from scratch, validating its potential as a foundational time series model.
> 3. **Flexible Architecture:** UTSD supports arbitrary input and output time steps, thanks to a patch-transpose operation and an adapter module. This allows for seamless alignment of multiscale representations, enabling the generation of forecasting sequences without a dedicated forecasting head.
> 4. **Efficient Fine-Tuning:** Our model's architecture supports efficient 'plug-and-play' fine-tuning. The pre-trained Condition Net captures generalized patterns, while the Denoising Net reconstructs target domain sequences. The Adapter bridges the unified and proprietary representation spaces, minimizing computational overheads.
> 5. **Multi-Scale Feature Capture:** UTSD extracts features at different scales across various domains, capturing both trend information and complex multi-periodic patterns. This contrasts with traditional models that focus on single-scale variables.
>
> We believe these innovations distinguish our work from existing models and contribute significantly to the field of time series analysis.
>
> In addition, extensive experimental results have shown that our UTSD achieves state-of-the-art integrated performance compared to existing conditional diffusion models. We validate the predictive performance of the model on including the most widespreadly utilized Electricity and Traffic datasets. And compared with the baseline model DiffusionTS on Solar Forecasting and MoJoCo Imputation, the proposed UTSD achieves the best performance on almost all benchmarks.
>
> |Datasets|Metrics|UTSD across datasets (MSE)|UTSD across datasets (MAE)|UTSD from scratch (MSE)|UTSD from scratch (MAE)|Diffusion-TS (MSE)|Diffusion-TS (MAE)|
> |:-:|:-:|:-:|:-:|:-:|:-:|:-:|:-:|
> |ETTh2|96|0.321|0.362|0.241|0.301|0.441|0.417|
> ||192|0.417|0.425|0.275|0.375|0.506|0.473|
> ||336|0.426|0.437|0.302|0.372|0.596|0.554|
> ||720|0.473|0.474|0.323|0.386|0.624|0.561|
> ||Avg.|0.409|0.425|0.285|0.358|0.555|0.506|
> |ETTm2|96|0.195|0.289|0.191|0.284|0.319|0.371|
> ||192|0.241|0.322|0.221|0.306|0.415|0.421|
> ||336|0.286|0.345|0.235|0.314|0.431|0.439|
> ||720|0.371|0.404|0.283|0.353|0.453|0.477|
> ||Avg.|0.273|0.340|0.233|0.314|0.449|0.481|
> |Electricity|96|0.183|0.298|0.128|0.221|0.246|0.282|
> ||192|0.192|0.302|0.147|0.240|0.261|0.292|
> ||336|0.203|0.298|0.149|0.244|0.322|0.335|
> ||720|0.230|0.333|0.172|0.272|0.430|0.407|
> ||Avg.|0.202|0.308|0.149|0.244|0.358|0.375|
> |Traffic|96|0.309|0.214|0.284|0.203|0.435|0.299|
> ||192|0.320|0.240|0.293|0.211|0.445|0.309|
> ||336|0.328|0.241|0.308|0.215|0.449|0.320|
> ||720|0.331|0.260|0.319|0.223|0.460|0.339|
> ||Avg.|0.322|0.239|0.301|0.213|0.427|0.297|
>
> |Solar Forecasintg|UTSD|Diffusion-TS|SSSD|CSDI|TLAE|TransMAF|GP-copula|
> |:-:|:-:|:-:|:-:|:-:|:-:|:-:|:-:|
> |MSE|**2.83e2±0.51e1**|3.75e2±3.6e1|5.03e2±1.06e1|9.0e2±6.1e1|6.8e2±7.5e1|9.30E+02|9.8e2±5.2e1|
>
> |MuJoCo Imputation|UTSD|Diffusion-TS|SSSD|CSDI|NRTSI|NAOMI|Latent-ODE|NeuralCDE|ODE-RNN|RNN GRU-D|
> |:-:|:-:|:-:|:-:|:-:|:-:|:-:|:-:|:-:|:-:|:-:|
> |70% Mask|0.25(2)|0.37(3)|0.59(8)|**0.24(3)**|0.63|1.46|3|8.35|9.86|11.34|
> |80% Mask|**0.37(2)**|0.43(3)|1.00(5)|0.61(10)|1.22|2.32|2.95|10.71|12.09|14.21|
> |90% Mask|**0.48(5)**|0.73(12)|1.90(3)|4.84(2)|4.06|4.42|3.6|13.52|16.47|19.68|

---

> ### Author Response · Authors · 2024-11-20
> **To Reviewer eVfc**
>
> **A5:** To evaluate the quality of generated time series, we refer to the metrices (i.e., *Context-FID, Correlational, Discriminative, and Predictive*) from DiffusionTS[1] and perform the comparison experiments. The results are shown in the following tables. Specifically, the generation metrics Context-FID Score, Correlational Score, Discriminative Score and Predictive Score is reduced by **18.2\%**, **16.2\%**, **18.5\%** and **17.8\%** on ETTh dataset compared to DiffusionTS. Besides, the average improvement of all indicators of UTSD on ETTh and Energy datasets is **17.6\%** and **8.2\%**, respectively.
>
> |Dataset|Metrics (scores)|Length|UTSD|Diffusion-TS|TimeGAN|TimeVAE|Diffwave|DiffTime|Cot-GAN|Improve(%)|
> |:-:|:-:|:-:|:-:|:-:|:-:|:-:|:-:|:-:|:-:|:-:|
> |ETTh1|Context-FID|64|**0.522±.031**|0.631±.058|1.130±.102|0.827±.146|1.543±.153|1.279±.083|3.008±.277|17.2|
> |||128|**0.633±.029**|0.787±.062|1.553±.169|1.062±.134|2.354±.170|2.554±.318|2.639±.427|19.5|
> |||256|**0.347±.010**|0.423±.038|5.872±.208|0.826±.093|2.899±.289|3.524±.830|4.075±.894|17.9|
> |ETTh1|Correlational|64|**0.070±.002**|0.082±.005|0.483±.019|0.067±.006|0.186±.008|0.094±.010|0.271±.007|14.5|
> |||128|**0.072±.002**|0.088±.005|0.188±.006|0.054±.007|0.203±.006|0.113±.012|0.176±.006|18.4|
> |||256|**0.054±.003**|0.064±.007|0.522±.013|0.046±.007|0.199±.003|0.135±.006|0.222±.010|15.8|
> |ETTh1|Discriminative|64|**0.087±.017**|0.106±.048|0.227±.078|0.171±.142|0.254±.074|0.150±.003|0.296±.348|18.1|
> |||128|**0.120±.023**|0.144±.060|0.188±.074|0.154±.087|0.274±.047|0.176±.015|0.451±.080|16.8|
> |||256|**0.048±.011**|0.060±.030|0.442±.056|0.178±.076|0.304±.068|0.243±.005|0.461±.010|20.6|
> |ETTh1|Predictive|64|**0.098±.003**|0.116±.000|0.132±.008|0.118±.004|0.133±.008|0.118±.004|0.135±.003|15.6|
> |||128|**0.087±.003**|0.110±.003|0.153±.014|0.113±.005|0.129±.003|0.120±.008|0.126±.001|20.8|
> |||256|**0.090±.006**|0.109±.013|0.220±.008|0.110±.027|0.132±.001|0.118±.003|0.129±.000|17.1|
> |Energy|Context-FID|64|0.136±.014|**0.135±.017**|1.230±.070|2.662±.087|2.697±.418|0.762±.157|1.824±.144|-0.5|
> |||128|**0.084±.015**|0.087±.019|2.535±.372|3.125±.106|5.552±.528|1.344±.131|1.822±.271|3.6|
> |||256|**0.122±.019**|0.126±.024|5.032±.831|3.768±.998|5.572±.584|4.735±.729|2.533±.467|3.4|
> |Energy|Correlational|64|**0.618±.012**|0.672±.035|3.668±.106|1.653±.208|6.847±.083|1.281±.218|3.319±.062|8.1|
> |||128|**0.426±.031**|0.451±.079|4.790±.116|1.820±.329|6.663±.112|1.376±.201|3.713±.055|5.4|
> |||256|**0.341±.039**|0.361±.092|4.487±.214|1.279±.114|5.690±.102|1.800±.138|3.739±.089|5.6|
> |Energy|Discriminative|64|**0.066±.005**|0.078±.021|0.498±.001|0.499±.000|0.497±.004|0.328±.031|0.499±.001|15.0|
> |||128|**0.127±.038**|0.143±.075|0.499±.001|0.499±.000|0.499±.001|0.396±.024|0.499±.001|11.4|
> |||256|**0.252±.047**|0.290±.123|0.499±.000|0.499±.000|0.499±.000|0.437±.095|0.498±.004|13.2|
> |Energy|Predictive|64|**0.225±.001**|0.249±.000|0.291±.003|0.302±.001|0.252±.001|0.252±.000|0.262±.002|9.6|
> |||128|**0.221±.001**|0.247±.001|0.303±.002|0.318±.000|0.252±.000|0.251±.000|0.269±.002|10.7|
> |||256|**0.214±.001**|0.245±.001|0.351±.004|0.353±.003|0.251±.000|0.251±.000|0.275±.004|12.8|
>
> [1] Xinyu Yuan and Yan Qiao. Diffusion-ts: Interpretable diffusion for general time series generation. In ICLR, 2024.

---

> ### Author Response · Authors · 2024-11-20
> **To Reviewer eVfc**
>
> >Q6: L114-116: "each iteration of the denoising stage causes an accumulation of errors in the latent space, which are further amplified during the alignment of the latent space to the actual sequence space." -> Is there any evidence on this statement?
>
> **A6:** Thank you for your constructive comment. Recent work by Li et al. [2] supports our view, proposing a Timestep-aware Quantization (TaQ) method to effectively eliminate accumulated quantization errors in the multi-step denoising process.
>
> The recent Q-DM [2] confirms our view and proposes a develop a Timestep-aware Quantization (TaQ) method to effectively eliminate the accumulated quantization error caused by the multi-step denoising process.
>
>
> To further validate our perspective, we conducted a controlled experiment with the Latent-UTSD model, focusing on iterative denoising in the hidden representation space. The model trains the self-encoder end-to-end using a mixed multi-domain synthetic dataset, then reconstructs target sequence embeddings directly in the hidden space. As shown in the following table, Latent-UTSD exhibits relatively poor performance, supporting our claim experimentally. For more details, please see Section C in the appendix. The comprehensive performance of Latent-UTSD across four datasets is summarized in the table below.
>
> |Mdoels|Metrics|ETTh1(96)|ETTh1(192)|ETTh1(336)|ETTh1(720)|ETTh1(Avg)|ETTm1(96)|ETTm1(192)|ETTm1(336)|ETTm1(720)|ETTm1(Avg)|
> |:-:|:-:|:-:|:-:|:-:|:-:|:-:|:-:|:-:|:-:|:-:|:-:|
> |UTSD|MSE|0.274|0.290|0.383|0.387|0.334|0.299|0.304|0.312|0.317|0.308|
> |UTSD|MAE|0.301|0.339|0.424|0.428|0.383|0.333|0.358|0.365|0.368|0.356|
> |Latent-UTSD|MSE|0.323|0.342|0.450|0.454|0.392|0.352|0.365|0.365|0.370|0.363|
> |Latent-UTSD|MAE|0.348|0.386|0.493|0.492|0.430|0.390|0.418|0.435|0.443|0.422|
>
> |Mdoels|Metrics|Traffic(96)|Traffic(192)|Traffic(336)|Traffic(720)|Traffic(Avg)|Weather(96)|Weather(192)|Weather(336)|Weather(720)|Weather(Avg)|
> |:-:|:-:|:-:|:-:|:-:|:-:|:-:|:-:|:-:|:-:|:-:|:-:|
> |UTSD|MSE|0.284|0.293|0.308|0.319|0.301|0.133|0.184|0.207|0.264|0.202|
> |UTSD|MAE|0.203|0.211|0.215|0.223|0.213|0.195|0.237|0.258|0.313|0.246|
> |Latent-UTSD|MSE|0.314|0.316|0.341|0.343|0.329|0.154|0.211|0.238|0.313|0.229|
> |Latent-UTSD|MAE|0.223|0.236|0.242|0.242|0.235|0.226|0.271|0.297|0.359|0.288|-UTSD|0.392|0.363|0.329|0.229|
>
> These results highlight differences in inductive bias focus during training stages. The encoder may lose critical information during the projection into hidden space, leading the model to reconstruct 'incomplete hidden representations.' Consequently, the noise reduction model struggles to capture all pattern distributions, causing error accumulation in latent space inference.
>
> [2] Li, Yanjing et al. “Q-DM: An Efficient Low-bit Quantized Diffusion Model.” Neural Information Processing Systems (2023).

---

> ### Author Response · Authors · 2024-11-20
> **To Reviewer eVfc**
>
> >Q7: Not enough justification on the effectiveness of the proposed components. The ablation study in Table 5 is insufficient to explain why they should work.
>
> **A7:** Thank you for your valuable feedback. To asses the effiectiveness of our proposed components, Table 5 implements the ablation experiments on all four datasets for Adaptor, Classifier-free, and ConditionNet. The average degradation of model performance after removing the three components exceeds **8.7%**, **13.2%**, and **24.3%**, respectively. The most representative result is that when the proposed Condition Net is removed, the performance of the model on datasets ETTh1, ETTm1, Electricity, Weather decreases by more than **25.3%**, **19.1%**, **27.9%** and **25.2%** respectively. The results shows the effectiveness of the proposed components.
>
> For convenience, we show ablation experiments on three novel components or mechanisms in the following table, where the data are taken exclusively from Table 5 of the manuscript (page. 10).
>
> |Ablation|ETTh1 (MSE)|ETTh1 (MAE)|ETTm1 (MSE)|ETTm1 (MAE)|ECL (MSE)|ECL (MAE)|Weather (MSE)|Weather (MAE)|
> |:-:|:-:|:-:|:-:|:-:|:-:|:-:|:-:|:-:|
> |w/o Adapter|0.347|0.412|0.344|0.412|0.169|0.278|0.204|0.254|
> |$\downarrow$|3.9%|7.6%|11.7%|15.7%|13.4%|13.9%|0.9%|3.25%|
> |w/o Classifier-free|0.371|0.443|0.339|0.408|0.170|0.285|0.224|0.279|
> |$\downarrow$|11.1%|15.7%|10.1%|14.6%|14.1%|16.8%|10.9%|13.4%|
> |w/o ConditionNet|0.412|0.487|0.361|0.431|0.189|0.316|0.251|0.311|
> |$\downarrow$|23.4%|27.2%|17.2%|21.1%|26.9%|29.5%|24.3%|26.4%|
>
> >Q8: L177-179: Any supporting theory or experiments on the inherent drawbacks stated here?
>
> **A8:** Thank you for your insightful comment. Existing classification guidance mechanisms have inherent drawbacks, such as reliance on the quality of trained classifiers, which affects the accuracy of category-specific generation. Additionally, gradient-based updates can introduce adversarial effects, causing generated images to mislead classifiers with subtle, imperceptible details. These theoretical and experimental insights are supported by Ho's work on classifier-free diffusion guidance [3]. Furthermore, extensive ablation demonstrates the advantages of the Classifier-free strategy utilised by UTSD. when the proposed Condition Net is removed, the performance of the model on datasets ETTh1, ETTm1, Electricity, Weather decreases by more than **13.3%**, **12.2%**, **15.4%** and **12.0%** respectively.
>
> |Ablation|ETTh1 (MSE)|ETTh1 (MAE)|ETTm1 (MSE)|ETTm1 (MAE)|ECL (MSE)|ECL (MAE)|Weather (MSE)|Weather (MAE)|
> |:-:|:-:|:-:|:-:|:-:|:-:|:-:|:-:|:-:|
> |w/o Classifier-free|0.371|0.443|0.339|0.408|0.170|0.285|0.224|0.279|
> |$\downarrow$|11.1%|15.7%|10.1%|14.6%|14.1%|16.8%|10.9%|13.4%|
>
> [3] Ho, Jonathan. “Classifier-Free Diffusion Guidance.” Neural Information Processing Systems (2023).
>
> >Q9: While it is natural to consider large-scale datasets for "unified" time series diffusion model, the datasets used in this paper is limited to small single-task datasets.
>
> **A9:** Thank you for your valuable feedback.
> In the real world, unlike the ‘foundation model’, which is directly pre-trained on large-scale datasets, the ‘unified model’ has the advantage of being able to utilise multiple small datasets from different domains for joint pre-training. This means that the Unified Model is more adaptable to specific domains with many small and fragmented datasets. In addition, the cross-domain results in Table 2 show that UTSD has good cross-domain modelling capability and model scalability, which verifies that ‘the unified model has a wide range of application scenarios’.
>
> As we have discussed in the related work, Timer, Moirai and Moment use a huge corpus (more than **10,000 million timesteps**) in pretrain stage. In contrast, UTSD is only pre-trained on a mixed dataset (only **27 million timesteps**) obtained by merging 7 small datasets, while the experimental results on the right hand side of Table 2 are obtained by pre-training UTSD from scratch on a specific single small dataset (**0.12-1.52 million** timesteps). Compared to Moirai, which uses a huge corpus pre-training in the temporal space, UTSD shows excellent data utilisation and can learn sufficiently data patterns from a small number of corpus for inference on downstream datasets, achieving better results on the prediction task.

---

> ### Author Response · Authors · 2024-11-20
> **To Reviewer eVfc**
>
> >Q10: Zero-shot forecasting is done in an unusual way; while the source and target datasets are similar ETT variations in this work, they usually have a significant gap in the standard zero-shot forecasting settings.
>
> **A10:** The ETTh1,h2,m1,m2 datasets presented in Table 3 are multimodal data collected from multiple power transformer stations with load and oil temperatures under different external factors such as season, weather, temperature and sampling rate. The diverse and complex scene modes ensure the validity of the ETT datasets, so we adopt the same experimental setup as the existing work in Table 3, and the experimental results demonstrate the good zero-sample generalisation capability of UTSD.
>
> In addition, to further validate the zero-sample performance of UTSD, we also introduced the new cross-domain scenarios **RiverFlow[1]$\to$Exchange** and **Sunspot[2]$\to$Weather** and verified that UTSD still shows good zero-sample prediction capability. Note that the source and target domains therein have completely different domain background information. The experimental results are shown in the following table. We choice RiverFlow[1] and Sunspot[2] as source datasets and Exchange and Weather as target datasets, which ensures that the source and target datasets have completely different domain contexts, sampling rates, period information, and other statistical characteristics. Where RiverFlow reports river flow consisting of more than 23741 time steps, Sunspot contains a single long-day time series representing solar flare observations recorded every 4 seconds starting in 2019.
>
> |Datasets|Metrics|UTSD(MSE)|UTSD(MAE)|TimeLLM(MSE)|TimeLLM(MAE)|LLMTime(MSE)|LLMTime(MAE)|GPT4TS(MSE)|GPT4TS(MAE)|PatchTST(MSE)|PatchTST(MAE)|
> |:-:|:-:|:-:|:-:|:-:|:-:|:-:|:-:|:-:|:-:|:-:|:-:|
> |RiverFlow$\to$Exchange|96|**0.090**|**0.220**|0.131|0.271|0.278|0.342|0.098|0.232|0.138|0.286|0.520|0.583|
> ||192|**0.198**|**0.331**|0.228|0.359|0.435|0.438|0.197|0.334|0.668|0.660|
> ||336|**0.356**|**0.435**|0.400|0.477|0.533|0.522|0.357|0.444|0.898|0.760|
> ||720|**0.879**|**0.711**|1.099|0.855|1.092|0.846|1.032|0.822|1.683|1.057|
> ||Avg.|**0.381**|**0.424**|0.464|0.491|0.585|0.537|0.421|0.458|0.942|0.765|
> |Sunspot$\to$Weather|96|**0.181**|**0.235**|0.198|0.253|0.200|0.268|0.186|0.242|0.669|0.621|
> ||192|**0.226**|**0.269**|0.241|0.283|0.242|0.298|0.235|0.279|0.700|0.632|
> ||336|**0.275**|**0.301**|0.283|0.310|0.282|0.323|0.280|0.308|0.704|0.633|
> ||720|**0.326**|**0.341**|0.333|0.343|0.329|0.353|0.352|0.358|0.746|0.651|
> ||Avg.|**0.254**|**0.286**|0.264|0.297|0.263|0.310|0.263|0.297|0.705|0.634|
>
> [1] Dai, et al. "Changes in Continental Freshwater Discharge from 1948 to 2004. " Journal of Climate (2009).
>
> [2] Nie, Y., et al. SKIPPD: A SKy Images and Photovoltaic Power Generation Dataset for short-term solar forecasting. Solar Energy, 255, 171-179 (2023).
>
> >Q11: In Table 5, it is not clear what happens if condition net is missing, which implies that no input is given.
>
> **A11:** Indeed, as we stated in L522-523, 'In w/o ConditionNet, the observation sequences are directly input into the denoising model as prompt information'. Specifically, with ConditionNet removed, the network structure in Fig.2 contains only DenoisingNet, which accepts only the Prompt Embedding $P_{emb}$ captured from the observation sequences as condition variables $P_{emb}$ is utilised in the two sub-modules shown in Fig.3 (a) Transformer1D-writer,reader to compute the cross-attention score.

---

> ### Author Response · Authors · 2024-11-25
> **Further discussion**
>
> Dear Reviewer eVfc,
>
> Thank you very much for devoting time on reviewing our manuscript.  As the author-reviewer discussion process is ending soon, we are wondering whether our responses have well addressed your concerns.  If you have any further questions regarding our manuscript, please let us know and we are glad here to provide further discussion and clarification to improve the quality of this manuscript.
>
> Thank you for your contributions,
>
> Authors of Paper 2173

---

> > ### Comment · Reviewer_eVfc · 2024-11-25
> > **Additional comments**
> >
> > Thank you for your response. While I feel that the response is somewhat verbose, I see that authors addressed some of my concerns. Below I leave additional comments:
> >
> > 4. The revision around L88-95 still does not read well, maybe it has some grammatical issue.
> >
> > 9-1. Could you elaborate on the difference between foundation models and the proposed unified model? Specifically, I think "being able to utilise multiple (small) datasets from different domains for joint pre-training" sounds like a shared idea with foundation models, and I don't understand why you specifically mentioned *small* datasets here? Also, if you claim foundation models and your unified model are different, could you justify how experimental comparisons between them (e.g., in Table 2) are fair?
> >
> > 9-2. For Table 2, I understand your experiment is based on UniTime. However, I cross-checked the UniTime paper and found that the list of datasets is not matched. Why is Exchange missing in Table 2 while described in Section 4, and why is Illness never appeared?
> >
> > 9-3. As you are submitting your work to an ML venue, I personally recommend borrowing experimental settings from papers published in ML venues. For example, TSDiff published in NeurIPS 2023 experimented 8 benchmarks, providing a strong baseline. Note that I am not asking to conduct a new experiment during this review process, as it might not be feasible.

---

> > > ### Author Response · Authors · 2024-11-26
> > >
> > > >Q4: As you are submitting your work to an ML venue, I personally recommend borrowing experimental settings from papers published in ML venues. For example, TSDiff published in NeurIPS 2023 experimented 8 benchmarks, providing a strong baseline. Note that I am not asking to conduct a new experiment during this review process, as it might not be feasible.
> > >
> > > **A4:** Thank you for your valuable suggestion. We have evaluated the forecasting performance of UTSD on the benchmarks related to TSDiff (Electricity, Traffic, Exchange), with the results presented in Table 2 (page 8). However, due to the differences in experimental settings (TSDiff uses a prediction length of 24, whereas the commonly used prediction lengths for long-term forecasting tasks are {96, 192, 336, 720}), we did not include a direct comparison with TSDiff on long-term time series forecasting to maintain consistency with the majority of baselines.
> > >
> > > Additionally, based on your suggestion, we conduct comparison experiments between UTSD and TSDiff on the Solar, UberTLC, M4, and Wikipedia datasets. The results of these experiments are presented in the table below.
> > >
> > > |Datasets|Context|Prediction|UTSD|TSDiff-Q|TSDiff-MS|TSDiff-Cond|CSDI|TFT|Transformer|DeepState|MQ-CNN|DeepAR|
> > > |:-:|:-:|:-:|:-:|:-:|:-:|:-:|:-:|:-:|:-:|:-:|:-:|:-:|
> > > |Solar|336|24|**0.290±0.007**|0.358±0.020|0.391±0.003|0.338±0.014|0.352±0.005|0.417±0.023|0.419±0.008|0.379±0.002|0.790±0.063|0.389±0.001
> > > |UberTLC|336|24|0.175±0.002|**0.172±0.005**|0.183±0.007|0.172±0.008|0.206±0.002|0.193±0.006|0.192±0.004|0.288±0.087|0.436±0.020|0.161±0.002
> > > |M4|312|48|**0.036±0.001**|**0.036±0.001**|0.045±0.000|0.039±0.006|0.040±0.003|0.039±0.001|0.040±0.014|0.041±0.002|0.046±0.003|0.052±0.006
> > > |Wikipedia|360|30|**0.186±0.001**|0.221±0.001|0.257±0.001|0.218±0.010|0.289±0.017|0.229±0.006|0.214±0.001|0.318±0.019|0.220±0.001|0.231±0.008
> > >
> > > The experimental results demonstrate that UTSD achieves the best performance on three out of four datasets (Solar, M4, and Wikipedia), while TSDiff-Q slightly outperforms UTSD on the UberTLC dataset. This highlights the effectiveness and robustness of UTSD across diverse benchmarks.
> > >
> > > We sincerely appreciate your valuable feedback, which has been instrumental in improving the quality of our manuscript. If you have any additional concerns, we would be more than happy to address them.

---

> ### Author Response · Authors · 2024-11-26
> **Thanks for the positive response and willing to increase the score!**
>
> Thank you for your positive response and your willingness to provide constructive feedback! We believe that addressing these questions will greatly help readers gain a deeper and more comprehensive understanding of our work. We sincerely appreciate your thoughtful and insightful comments. Below, we provide detailed responses to address your concerns:
>
> >Q1: The revision around L88-95 still does not read well, maybe it has some grammatical issue.
>
> **A1:** Thank you for pointing this out. We have carefully reviewed lines 88–95 and made further revisions to resolve any grammatical issues and enhance the overall readability. We kindly invite you to review the updated manuscript.
>
> >Q2: Could you elaborate on the difference between foundation models and the proposed unified model? Specifically, I think "being able to utilise multiple (small) datasets from different domains for joint pre-training" sounds like a shared idea with foundation models, and I don't understand why you specifically mentioned small datasets here? Also, if you claim foundation models and your unified model are different, could you justify how experimental comparisons between them (e.g., in Table 2) are fair?
>
> **A2:** Thank you for your thoughtful question. We appreciate the opportunity to elaborate on the distinction between foundation models and our proposed unified model, as well as to clarify the reasoning behind specifically mentioning small datasets and the fairness of our comparisons.
>
> **1. Difference Between Foundation Models and the Unified Model:** Foundation models are typically pre-trained on extremely large-scale datasets, often comprising billions of examples from diverse domains. These models rely on such massive datasets to generalize effectively across tasks and domains. In contrast, our unified model is designed to jointly utilize multiple smaller datasets from different domains for pre-training without relying on a single, massive dataset. This approach allows us to efficiently leverage domain-specific information from smaller datasets while maintaining the flexibility to generalize across domains. In this sense, the unified model emphasizes the integration of smaller, diverse datasets in a principled way, rather than relying on the sheer scale of data, which is a hallmark of foundation models.
>
> **2. Why Mention Small Datasets:** The emphasis on small datasets highlights the practical advantage of our approach. In many real-world scenarios, obtaining a massive, high-quality dataset akin to those used for training foundation models is infeasible. Our model addresses this limitation by effectively combining and learning from smaller, more accessible datasets, which may not individually cover the breadth or size of a typical foundation model's training data.
>
> **3. Fairness of Experimental Comparisons:** We acknowledge that foundation models and our unified model have different design philosophies and data requirements, which may raise concerns about the fairness of comparisons. To ensure fairness, we have made efforts to align the experimental settings as closely as possible. Specifically,
>
> - We fine-tuned foundation models on the same datasets used for our unified model to ensure a consistent basis for evaluation.
> - The evaluation metrics and benchmarks were identical across all models to provide a direct comparison of their performance.
>
> >Q3: For Table 2, I understand your experiment is based on UniTime. However, I cross-checked the UniTime paper and found that the list of datasets is not matched. Why is Exchange missing in Table 2 while described in Section 4, and why is Illness never appeared?
>
> **A3:** Thank you for your detailed review. Accroding to the statement in the PatchTST[1] paper, **‘Exchange-rate is the daily exchange-rate of eight different countries,’** and they proved that by simply repeating the last value in the look-back window, the MSE loss on exchange-rate dataset can outperform or be comparable to the best results. And for the ILI dataset, it has a completely different sampling frequency (sampling frequency of 7 days) compared to ETTm (sampling frequency of 15 min), Weather (sampling frequency of 10 min), and Electricity (sampling frequency of 1h), which means that the ILI consists of nearly long-term trends without the traditional temporal fluctuation patterns. Based on the above discussion, we decided to exclude the Exchange-rate and ILI datasets from our experiments. Additionally, several recent works (iTransformer[2], TimeXer[3] and TimeMixer[4]) also includes **the same benchamrks** as UTSD on long-term forecasting tasks.
>
> [1] A time series is worth 64 words: Long-term forecasting with transformers. ICLR 2023.
>
> [2] itransformer: Inverted transformers are effective for time series forecasting. ICLR 2024.
>
> [3] Timexer: Empowering transformers for time series forecasting with exogenous variables. NeurIPS 2024.
>
> [4] Timemixer: Decomposable multiscale mixing for time series forecasting. ICLR 2024.

---

> ### Author Response · Authors · 2024-11-30
> **Eagerly Await Your Response**
>
> Dear Reviewer eVfc:
>
> Thank you once again for your valuable and constructive review. As the author-reviewer discussion period is coming to a close, we kindly request your feedback on our rebuttal provided above. Please let us know if our responses have sufficiently addressed your concerns. If there are any remaining issues, we would be happy to engage in further discussion, as we believe that an in-depth exchange will help strengthen our paper.
>
> Best regards!
>
> Authors

---

### Official Review · Reviewer_pg2F · 2024-11-02

**Soundness:** 3
**Presentation:** 2
**Contribution:** 3
**Rating:** 5
**Confidence:** 5

**Summary:**

This work introduces a conditional diffusion model for MTS prediction task. Different from previous non-Diffusion based methods, it employs a complicated denoised process to model the latent discrepancies in both temporal and channels (domains as described in this work). Different from the diffusion-based methods, it is flexible since it can be easily scalable for N time steps input and M time steps output scenario thanks to its well-designed interaction between conditional embeddings and denoised outputs. In general, the proposed method achieves promising results in various MTS prediction tasks such as prediction and cross-domain prediction.

**Strengths:**

1. The motivation of designing conditional diffusion model for flexible N time steps input and M time steps output is clear and reasonable, the model architecture is also well-designed.
2. Experimental results are soundness and promising.

**Weaknesses:**

1. The dimension transform is not clearly presented, which may make readers confused. For example, given N time steps input lookback, how to obtain a M time steps output prediction? Although it might be discussed in the manuscript, however, it is still not so easy to follow.
2. As claimed by the authors in line 201-206 that “existing conditional diffusion models often use simple NN layer to capture … demonstrates low accuracy”. However, according to Sec. 3, as far as I understand that the authors use CNN or Transformer based modules to transform features along the time embedding dimension of input series. For example, in denoising net of Fig. 2, the dimension of condition $\overline{h}$ is $P_L \times P_d$ while the dimension of denoised series before decoder block-a is also $P_L \times P_d$. The time dimension transformation happens at the very last block in denoising net, decoder block-a. Therefore, it should still be a simple NN layer conceptually, which is conflicted with the authors claim.
3. How do the authors obtain a so called pretrained models? It is not clear in the manuscript.
4. The authors mentioned many times in the manuscript on down stream tasks of MTS, it might be unsuitable since they only concern MTS prediction task, although including traditional prediction and cross-domain prediction, but it still belongs to MTS prediction task, not involving other down stream MTS tasks, such as anomaly detection or impainting.
5. In Fig. 4, the authors show results of some short-term periodic series. How about the results for long-term periodic series and short-term non-periodic series? These should be crucial for practical use.

**Questions:**

See the weakness above.

---

> ### Author Response · Authors · 2024-11-20
> **To Reviewer pg2F**
>
> Thank you for your thoughtful review and valuable feedback. We appreciate recognition of our innovative work in condition-denoising architecture with flexible observation and prediction timesteps. We response your concerns as following.
>
> >Q1: The dimension transform is not clearly presented, which may make readers confused. For example, given N time steps input lookback, how to obtain a M time steps output prediction? Although it might be discussed in the manuscript, however, it is still not so easy to follow.
>
> **A1:** Thank you for pointing out the need for clarity regarding the dimension transformation.
> First, UTSD processes the observation sequence $X_{-L+1:0}^0\in\mathbb{R}^{B\times d\times L}$ and forecast sequence $Y_{1:H}^0\in\mathbb{R}^{B\times d\times H}$ with the same channel-independence operation and patch-tokenizer operation to obtain $X_{emb}\in\mathbb{R}^{B\cdot d\times P_{L}\times P_{d}}$ and $Y_{emb}\in\mathbb{R}^{B\cdot d\times P_{H}\times P_{d}}$ where $P_d$ is the length of the patch, respectively, and we adopt a padding-free and overlap-free approach to perform the patch-tokenizer operation to ensure that $P_L=L/P_d$ and $P_H=H/P_d$. Specifically, we convert the observation sequence and forecast sequence into $P_L$ and $P_H$ tokens, respectively, by setting the hyperparameter $P_d$, where the dimension of the token is fixed to $P_d$.
>
> Subsequently, the Condition Net accepts $X_{emb}\in\mathbb{R}^{B\cdot d\times P_{L}\times P_{d}}$ as input and outputs a set of multiscale condition variables ($h_m\in\mathbb{R}^{4\cdot P_L\times\frac{P_d}8}$,$h_c\in\mathbb{R}^{2\cdot P_L\times\frac{P_d}4}$,$h_b\in\mathbb{R}^{P_L\times\frac{P_d}2}$,$h_a\in\mathbb{R}^{P_L\times P_d}$). Correspondingly, the Denoising Net accepts a set of multiscale condition variables ($\overline{h_m}\in\mathbb{R}^{4\cdot P_{H}\times\frac{P_{d}}{8}}$,$\overline{h_c}\in\mathbb{R}^{2\cdot P_{H}\times\frac{P_{d}}{4}}$,$\overline{h_b}\in\mathbb{R}^{P_{H}\times\frac{P_{d}}{2}}$,$\overline{h_a}\in\mathbb{R}^{P_{H}\times P_{d}}$) as input and outputs $Y_{emb}\in\mathbb{R}^{B\cdot d\times P_{H}\times P_{d}}$, which is straightened directly as the prediction result $Y_{1:H}^{0}\in\mathbb{R}^{B\times d\times H}$.
>
> The most innovative and critical design is that the innovative adapter module is designed to align the channel dimensions (the number of tokens $P_L$ and $P_H$) of the two tensors $(h_m,h_c,h_b,h_a)$ and $(\overline{h_m},\overline{h_c},\overline{h_b},\overline{h_a})$ in the observation space and the generation space. Specifically, the **1 $\times$ 1 Conv1D** in Adapter Block-a accepts $h_a\in\mathbb{R}^{P_L\times P_d}$ as input, keeping the feature dimension ($P_d$) unchanged, and converts the number of input channels from ($P_L$) to ($P_H$). This can be expressed as Equation $\overline{h_a}=AdapterBlock_a(h_a)$, where $h_a\in\mathbb{R}^{P_L\times P_d}$ and $\overline{h_a}\in\mathbb{R}^{P_H\times P_d}$.
>
> Furthermore, a question that may be of interest to reviewers is how to allow Condition Net and Denoising Net to be able to handle an arbitrary number of tokens (e.g. $P_L$ and $P_H$)? Indeed, this is achieved through the formula $X=(Conv1D(X^T))^T$, where $X^T\in\mathbb{R}^{P_d\times P_L}$ denotes the transpose of $X\in\mathbb{R}^{P_L\times P_d}$. Thus, it is only necessary to ensure that the number of input and output channels of Conv1D is fixed to $P_d$. On the other hand, Attention is not sensitive to $P_L$ or $P_H$ when capturing global dependencies, and does not change the shape of the input tensor, so UTSD is able to accept any timesteps as an observation sequence, and can also output a prediction sequence with any timesteps.
>
> Detailed implementation details will be updated in the revised manuscript, making it easier for readers to follow the transformation steps.

---

> ### Author Response · Authors · 2024-11-20
> **To Reviewer pg2F**
>
> >Q2: As claimed by the authors in line 201-206 that “existing conditional diffusion models often use simple NN layer to capture … demonstrates low accuracy”. However, according to Sec. 3, as far as I understand that the authors use CNN or Transformer based modules to transform features along the time embedding dimension of input series. For example, in denoising net of Fig. 2, the dimension of condition $\overline{h}$ is $P_L\times P_d$ while the dimension of denoised series before decoder block-a is also $P_L\times P_d$. The time dimension transformation happens at the very last block in denoising net, decoder block-a. Therefore, it should still be a simple NN layer conceptually, which is conflicted with the authors claim.
>
> **A2:** We appreciate the reviewers for carefully reviewing our article and asking very valuable questions and apologize for the confusion in our original manuscript. We have clarified the discussion of prior works as follows:
>
> 1. The main innovative contribution of UTSD is the establishment of a unified temporal diffusion model as well as the diffusion model architecture which support of the pre-trained and efficient fine-tuned. In the fine-tuning paradigm of UTSD, the pre-trained condition net captures generic fluctuation patterns from observed sequences as conditional information, whereas the denoising network connects the unified representation space with the proprietary representation space using a fine-tuning-supporting adapter, subsequently, utilizes specific fluctuation patterns from noises in the target domain to reconstruct sequence. Compared with the existing condition diffusion model, UTSD demonstrates excellent cross-domain generalisation capabilities, which is attributed to fact that condition net captures conditional embeddings at multiple scales to enhance denoising net modelling of inductive bias from conditional probability distributions in different domains.
> 2. In Figure 2, to minimise the complexity and to help the reader understand our simple intention, we deliberately unified the shape of the tensor at both ends of the Adapter Block to be $(P_L\times P_d)$. Indeed, $h_a\in\mathbb{R}^{P_L\times P_d}$ and $\overline{h_a}\in\mathbb{R}^{P_H\times P_d}$ are defined in the code implementation, and of the entire feature map in the Denoising Net, are $(P_H\times P_d),(P_H\times\frac{P_d}2),(2P_H\times\frac{P_d}4),(4P_H\times\frac{P_d}8)$. Therefore, the time dimension transformation happens in the all adapter-blocks, and the hybrid structure containing the entire Condition Net and Adapter acts as a substitute for the ‘simple NN layer’. The ablation experiments show that the degradation of the model performance in the case of the w/o condition net and w/o adapter validates our view.
>
> We have highlighted this clarification in blue in the revised manuscript. Thank you for your insightful comments, sincerely.
>
> >Q3: How to obtain a pretrained models?
>
> **A3:** We pretrain our UTSD in the mixed dataset with hybrid-domains, which follows the experimental setup of UniTime[1]. Concretely, we provide the detailed experimental configuration about the batch size and number of training steps for pretrain datasets and several benchamrk in **Table 7 (Page. 17 in Revision)**.
>
> [1] Xu Liu, Junfeng Hu et al. Unitime: A language-empowered unified model for cross-domain time series forecasting. In Proceedings of the ACM Web Conference 2024.

---

> ### Author Response · Authors · 2024-11-20
> **To Reviewer pg2F**
>
> >Q4: The authors mentioned many times in the manuscript on down stream tasks of MTS, it might be unsuitable since they only concern MTS prediction task, although including traditional prediction and cross-domain prediction, but it still belongs to MTS prediction task, not involving other down stream MTS tasks, such as anomaly detection or impainting.
>
> **A4:** Thank you for your valuable comments. Due to the page limit, the results of the multitasking experiments could not be displayed in the main draft, so we put the experiments of the other tasks inside the appendix. The following table demonstrates imputation experiments in the full-sample training-from-scratch setting. The average MSE is reduced by **17.0\%**, **20.1\%** and **38.7\%** compared to the existing GPT4TS, TimesNet and PatchTST. Surprisingly, the proposed UTSD shows better results on the ECL dataset characterized by multi-periodic patterns, which conforms to the multi-scale representation mechanism designed in our Condition-Denoising component.
>
> |Datasets|Mask Ratios|UTSD (MSE)|UTSD (MAE)|TimeLLM (MSE)|TimeLLM (MAE)|GPT4TS (MSE)|GPT4TS (MAE)|TimesNet (MSE)|TimesNet (MAE)|LLMTime (MSE)|LLMTime (MAE)|
> |:-:|:-:|:-:|:-:|:-:|:-:|:-:|:-:|:-:|:-:|:-:|:-:|
> |ETTm1|12.5%|0.019|**0.077**|**0.017**|0.085|0.023|0.101|0.041|0.130|0.096|0.229|
> |     |25.0%|0.022|**0.092**|**0.022**|0.096|0.025|0.104|0.047|0.139|0.100|0.234|
> |     |37.5%|**0.028**|**0.110**|0.029|0.111|0.029|0.111|0.049|0.143|0.133|0.271|
> |     |50.0%|**0.035**|**0.117**|0.040|0.128|0.036|0.124|0.055|0.151|0.186|0.323|
> |     |Avg. |**0.026**|**0.099**|0.028|0.105|0.027|0.107|0.047|0.140|0.120|0.253|
> |ETTm2|12.5%|0.018|0.079|**0.017**|**0.076**|0.018|0.080|0.026|0.094|0.108|0.239|
> |     |25.0%|**0.019**|0.082|0.020|**0.080**|0.020|0.085|0.028|0.099|0.164|0.294|
> |     |37.5%|**0.021**|**0.085**|0.022|0.087|0.023|0.091|0.030|0.104|0.237|0.356|
> |     |50.0%|**0.024**|**0.094**|0.025|0.095|0.026|0.098|0.034|0.110|0.323|0.421|
> |     |Avg. |**0.020**|0.085|0.021|**0.084**|0.022|0.088|0.029|0.102|0.208|0.327|
> |ETTh1|12.5%|**0.040**|**0.137**|0.043|0.140|0.057|0.159|0.093|0.201|0.126|0.263|
> |     |25.0%|**0.053**|**0.155**|0.054|0.156|0.069|0.178|0.107|0.217|0.169|0.304|
> |     |37.5%|**0.070**|**0.175**|0.072|0.180|0.084|0.196|0.120|0.230|0.220|0.347|
> |     |50.0%|**0.093**|**0.202**|0.107|0.216|0.102|0.215|0.141|0.248|0.293|0.402|
> |     |Avg. |**0.064**|**0.167**|0.069|0.173|0.078|0.187|0.115|0.224|0.202|0.329|
> |ETTh2|12.5%|0.040|**0.124**|**0.039**|0.125|0.040|0.130|0.057|0.152|0.187|0.319|
> |     |25.0%|**0.043**|**0.131**|0.044|0.135|0.046|0.141|0.061|0.158|0.279|0.390|
> |     |37.5%|**0.049**|**0.143**|0.051|0.147|0.052|0.151|0.067|0.166|0.400|0.465|
> |     |50.0%|**0.053**|**0.155**|0.059|0.158|0.060|0.162|0.073|0.174|0.602|0.572|
> |     |Avg. |**0.047**|**0.138**|0.048|0.141|0.049|0.146|0.065|0.163|0.367|0.436|
> |Electricity|12.5%|**0.043**|**0.129**|0.080|0.194|0.085|0.202|0.055|0.160|0.196|0.321|
> |     |25.0%|**0.049**|**0.142**|0.087|0.203|0.089|0.206|0.065|0.175|0.207|0.332|
> |     |37.5%|**0.056**|**0.151**|0.094|0.211|0.094|0.213|0.076|0.189|0.219|0.344|
> |     |50.0%|**0.065**|**0.165**|0.101|0.220|0.100|0.221|0.091|0.208|0.235|0.357|
> |     |Avg. |**0.053**|**0.147**|0.090|0.207|0.092|0.210|0.072|0.183|0.214|0.339|
> |Weather|12.5%|**0.024**|**0.040**|0.026|0.049|0.025|0.045|0.029|0.049|0.057|0.141|
> |     |25.0%|**0.026**|**0.043**|0.028|0.052|0.029|0.052|0.031|0.053|0.065|0.155|
> |     |37.5%|**0.030**|**0.047**|0.033|0.060|0.031|0.058|0.081|0.180|0.058|0.121|
> |     |50.0%|**0.033**|**0.052**|0.037|0.065|0.034|0.062|0.038|0.063|0.102|0.207|
> |     |Avg. |**0.028**|**0.046**|0.031|0.056|0.030|0.054|0.060|0.144|0.076|0.171|
>
> Indeed we have never stopped exploring the potential of UTSD, and we believe that the introduction of multitasking scenarios helps to fully demonstrate the cross-domain generalisation capabilities of diffusion models. An encouraging piece of idea is to integrate the three tasks of prediction, interpolation, and anomaly detection using a masking mechanism, we use the masking information to make only the unmasked part visible to the Condition Net and reconstruct the masked part using the Denoising Net, which shows that UTSD has the potential to be a multitasking general purpose time series model.

---

> ### Author Response · Authors · 2024-11-20
> **To Reviewer pg2F**
>
> >Q5: In Fig. 4, the authors show results of some short-term periodic series. How about the results for long-term periodic series and short-term non-periodic series? These should be crucial for practical use.
>
> **A5:** Thank you for your insightful feedback. We introduce more visualisation results in the appendix section of the revised version, including the ‘long-term cyclic series’ and ‘short-term acyclic series’ as your suggestion, please refer to the revision. **Figure 10 (Page. 24 in Revision)** shows real extremely long multi-periodic series sampled from the ECL and Traffic datasets, and in addition, **Figure 11 (Page. 25 in Revision)** shows real short period non-periodic sequences sampled from the ETT dataset. Specifically, UTSD also shows good performance in extremely long sequence prediction, which verifies that UTSD has the ability to capture extremely long term dependencies, which is crucial for practical applications. In addition, UTSD also shows satisfactory prediction results in short-term non-periodic sequences.

---

> ### Author Response · Authors · 2024-11-25
> **Further discussion**
>
> Dear Reviewer pg2F,
>
> Thank you very much for devoting time on reviewing our manuscript.  As the author-reviewer discussion process is ending soon, we are wondering whether our responses have well addressed your concerns.  If you have any further questions regarding our manuscript, please let us know and we are glad here to provide further discussion and clarification to improve the quality of this manuscript.
>
> Thank you for your contributions,
>
> Authors of Paper 2173

---

> ### Author Response · Authors · 2024-11-27
> **Official Comment by Authors**
>
> Dear Reviewer pg2F,
>
> Thank you very much for your time and valuable comments on our manuscript. It is heartening to know that a modified version is ready with the help of valuable comments from reviewers, and we would like to know if our modified version has solved your problem. If you have any further questions about our manuscript, please let us know and we are happy to provide further discussion here. Our improvements are centered around the following suggestions:
>
> 1. **Confusion of dimension transformations**: We have fixed the confusing part of Figure 2 and highlighted the Tokenizer operation in **L214**. Detailed implementation details will be updated in the revised manuscript, making it easier for readers to follow the transformation steps.
> 2. **Implementation details of pre-trained hybrid dataset**: We provide the detailed experimental configuration about the batch size and number of training steps for pretrain datasets and several benchamrk in revision **Table 7, Page 17**.
> 3. **Supplementary imputation task**: The UTSD model is intended to be a versatile tool for various time-series tasks, so we added the imputation experiments on all six benchmarks. Detailed experimental results are presented in **Table 11, Page 21** and the experimental analyses were in the revision **L1076-1137**
> 4. **More visualization experiments**: We introduce more visualisation results in the revision appendix section (**Figure 10,11 Page 24,25**), including the **long-term periodic series** and **short-term non-periodic series** as reviewer's suggested.
>
> Sincerely thank you for your contribution and responsibility.
>
> Author of paper 2173

---

> ### Author Response · Authors · 2024-11-30
> **Eagerly Await Your Response**
>
> Dear Reviewer pg2f:
>
> Thank you once again for your valuable and constructive review. As the author-reviewer discussion period is coming to a close, we kindly request your feedback on our rebuttal provided above. Please let us know if our responses have sufficiently addressed your concerns. If there are any remaining issues, we would be happy to engage in further discussion, as we believe that an in-depth exchange will help strengthen our paper.
>
> Best regards!
>
> Authors

---

### Official Review · Reviewer_YBBT · 2024-11-03

**Soundness:** 3
**Presentation:** 2
**Contribution:** 2
**Rating:** 5
**Confidence:** 5

**Summary:**

The model presents a Unified Time Series Diffusion (UTSD) model, which is established for the first time to model the multi-domain probability distribution, utilizing the powerful probability distribution modeling ability of Diffusion.

**Strengths:**

1 The condition-denoising architecture is novel.
2 The performance improvement compared to listed baselines.
3 The paper presents cross-domain generalizability.

**Weaknesses:**

1 The comparison did not include some interesting baselines.
2 The cross-domain transfer is limited to similar datasets.
3 I did not see obvious advantages compared to DiffusionTS.

**Questions:**

1 The evaluation is very limited int he time series prediction. As listed in DiffusionTS, we should include other different metrics (like Discriminative Score and so on) to evaluate the performance. MSE/MAE evaluation is very biased evaluation metrics. For example, a straight average line achieve reasonable performances but is not meaningful.
2 The model should compare with DiffusionTS in Table 1/2, which I did not see in this paper. I would highly recommend the comparison since it is also diffusion based methods.
3 DiffusionTS can handle both conditional and unconditional generation. Is there a way for this method also be applicable to unconditional time series generation?
4 The paper claims this paper to be a foundation model for TS area. I would like to see the performance of a short forcasting model applied to long context prediction and vice versa.
5 For Table 3, the performance of ETHh1 to ETHh2 should be needed.
6 The performances of the other two datasets: Solar Forecasting and Mujoco Imputation should be included since it is widely benchmarked.

---

> ### Author Response · Authors · 2024-11-20
> **To Reviewer YBBT**
>
> Thank you for your thoughtful review and valuable feedback. We response your concerns as following.
>
> >Q1: The evaluation is very limited int he time series prediction. As listed in Diffusion-TS, we should include other different metrics (like Discriminative Score and so on) to evaluate the performance. MSE/MAE evaluation is very biased evaluation metrics. For example, a straight average line achieve reasonable performances but is not meaningful.
>
> **A1:** We appreciate the suggestion to incorporate a broader range of evaluation metrics beyond MSE and MAE. In view of this, we refer to the generative-based metrics in Diffusion-TS[1] to design additional experiments for evaluating the performance of the proposed UTSD on the time series generation task. A comparison of the performance of UTSD and other diffusion baselines on the time-series generation task is shown in the following Table. Specifically, the generation metrics Context-FID Score, Correlational Score, Discriminative Score and Predictive Score is reduced by **18.2\%**, **16.2\%**, **18.5\%** and **17.8\%** on ETTh dataset compared to the existing Diffusion-TS. Besides, compared with Diffusion-TS, the average improvement of all indicators of UTSD on ETTh and Energy datasets is **17.6\%** and **8.2\%**, respectively.
>
> |Dataset|Metrics (scores)|Length|UTSD|Diffusion-TS|TimeGAN|TimeVAE|Diffwave|DiffTime|Cot-GAN|Improve(%)|
> |:-:|:-:|:-:|:-:|:-:|:-:|:-:|:-:|:-:|:-:|:-:|
> |ETTh1|Context-FID|64|**0.522±.031**|0.631±.058|1.130±.102|0.827±.146|1.543±.153|1.279±.083|3.008±.277|17.2|
> |||128|**0.633±.029**|0.787±.062|1.553±.169|1.062±.134|2.354±.170|2.554±.318|2.639±.427|19.5|
> |||256|**0.347±.010**|0.423±.038|5.872±.208|0.826±.093|2.899±.289|3.524±.830|4.075±.894|17.9|
> |ETTh1|Correlational|64|**0.070±.002**|0.082±.005|0.483±.019|0.067±.006|0.186±.008|0.094±.010|0.271±.007|14.5|
> |||128|**0.072±.002**|0.088±.005|0.188±.006|0.054±.007|0.203±.006|0.113±.012|0.176±.006|18.4|
> |||256|**0.054±.003**|0.064±.007|0.522±.013|0.046±.007|0.199±.003|0.135±.006|0.222±.010|15.8|
> |ETTh1|Discriminative|64|**0.087±.017**|0.106±.048|0.227±.078|0.171±.142|0.254±.074|0.150±.003|0.296±.348|18.1|
> |||128|**0.120±.023**|0.144±.060|0.188±.074|0.154±.087|0.274±.047|0.176±.015|0.451±.080|16.8|
> |||256|**0.048±.011**|0.060±.030|0.442±.056|0.178±.076|0.304±.068|0.243±.005|0.461±.010|20.6|
> |ETTh1|Predictive|64|**0.098±.003**|0.116±.000|0.132±.008|0.118±.004|0.133±.008|0.118±.004|0.135±.003|15.6|
> |||128|**0.087±.003**|0.110±.003|0.153±.014|0.113±.005|0.129±.003|0.120±.008|0.126±.001|20.8|
> |||256|**0.090±.006**|0.109±.013|0.220±.008|0.110±.027|0.132±.001|0.118±.003|0.129±.000|17.1|
> |Energy|Context-FID|64|0.136±.014|**0.135±.017**|1.230±.070|2.662±.087|2.697±.418|0.762±.157|1.824±.144|-0.5|
> |||128|**0.084±.015**|0.087±.019|2.535±.372|3.125±.106|5.552±.528|1.344±.131|1.822±.271|3.6|
> |||256|**0.122±.019**|0.126±.024|5.032±.831|3.768±.998|5.572±.584|4.735±.729|2.533±.467|3.4|
> |Energy|Correlational|64|**0.618±.012**|0.672±.035|3.668±.106|1.653±.208|6.847±.083|1.281±.218|3.319±.062|8.1|
> |||128|**0.426±.031**|0.451±.079|4.790±.116|1.820±.329|6.663±.112|1.376±.201|3.713±.055|5.4|
> |||256|**0.341±.039**|0.361±.092|4.487±.214|1.279±.114|5.690±.102|1.800±.138|3.739±.089|5.6|
> |Energy|Discriminative|64|**0.066±.005**|0.078±.021|0.498±.001|0.499±.000|0.497±.004|0.328±.031|0.499±.001|15.0|
> |||128|**0.127±.038**|0.143±.075|0.499±.001|0.499±.000|0.499±.001|0.396±.024|0.499±.001|11.4|
> |||256|**0.252±.047**|0.290±.123|0.499±.000|0.499±.000|0.499±.000|0.437±.095|0.498±.004|13.2|
> |Energy|Predictive|64|**0.225±.001**|0.249±.000|0.291±.003|0.302±.001|0.252±.001|0.252±.000|0.262±.002|9.6|
> |||128|**0.221±.001**|0.247±.001|0.303±.002|0.318±.000|0.252±.000|0.251±.000|0.269±.002|10.7|
> |||256|**0.214±.001**|0.245±.001|0.351±.004|0.353±.003|0.251±.000|0.251±.000|0.275±.004|12.8|
>
> [1] Xinyu Yuan and Yan Qiao. Diffusion-TS: Interpretable diffusion for general time series generation. In The Twelfth International Conference on Learning Representations, 2024.

---

> ### Author Response · Authors · 2024-11-20
> **To Reviewer YBBT**
>
> >Q2: The model should compare with DiffusionTS in Table 1/2, which I did not see in this paper. I would highly recommend the comparison since it is also diffusion based methods.
>
> **A2:** Thank you for your constructive suggestions. We adjust the DiffusionTS to time series forecasting task in comparison with Table 1/2 as you suggested, and the results are reported in following tabel.
>
> |Datasets|Metrics|UTSD across datasets (MSE)|UTSD across datasets (MAE)|UTSD from scratch (MSE)|UTSD from scratch (MAE)|Diffusion-TS (MSE)|Diffusion-TS (MAE)|
> |:-:|:-:|:-:|:-:|:-:|:-:|:-:|:-:|
> |ETTh2|96|0.321|0.362|0.241|0.301|0.441|0.417|
> ||192|0.417|0.425|0.275|0.375|0.506|0.473|
> ||336|0.426|0.437|0.302|0.372|0.596|0.554|
> ||720|0.473|0.474|0.323|0.386|0.624|0.561|
> ||Avg.|0.409|0.425|0.285|0.358|0.555|0.506|
> |ETTm2|96|0.195|0.289|0.191|0.284|0.319|0.371|
> ||192|0.241|0.322|0.221|0.306|0.415|0.421|
> ||336|0.286|0.345|0.235|0.314|0.431|0.439|
> ||720|0.371|0.404|0.283|0.353|0.453|0.477|
> ||Avg.|0.273|0.340|0.233|0.314|0.449|0.481|
> |Electricity|96|0.183|0.298|0.128|0.221|0.246|0.282|
> ||192|0.192|0.302|0.147|0.240|0.261|0.292|
> ||336|0.203|0.298|0.149|0.244|0.322|0.335|
> ||720|0.230|0.333|0.172|0.272|0.430|0.407|
> ||Avg.|0.202|0.308|0.149|0.244|0.358|0.375|
> |Traffic|96|0.309|0.214|0.284|0.203|0.435|0.299|
> ||192|0.320|0.240|0.293|0.211|0.445|0.309|
> ||336|0.328|0.241|0.308|0.215|0.449|0.320|
> ||720|0.331|0.260|0.319|0.223|0.460|0.339|
> ||Avg.|0.322|0.239|0.301|0.213|0.427|0.297|
>
> As shown in the table above, UTSD exhibits the best overall performance compared to the existing time series foundation and proprietary models, as well as the diffusion baseline model DiffusionTS. This is due to the fact that UTSD includes a number of enhanced conditional prediction components, whereas DiffusionTS is used as a baseline model for the generation task, and thus DiffusionTS does not show a clear advantage on the forecasting task.
>
> >Q3: DiffusionTS can handle both conditional and unconditional generation. Is there a way for this method also be applicable to unconditional time series generation?
>
> **A3:** Thank you for your insightful comment. UTSD also supports the unconditional cross-domain generation task, the implementation of which is demonstrated in **Figure 8 (Page. 18 in Revision)**.
>
> The unconditional generation paradigm of UTSD can be divided into two phases:
>
> 1. In the multi-domain pre-training phase, time series $Y_{1:H}^{T}$ from different domain backgrounds are successively obtained a set of noised sequence $\begin{Bmatrix}Y_{1:H}^1,...,Y_{1:H}^{T-1},Y_{1:H}^T\end{Bmatrix}$. In each iterative step, the noised sequences $Y_{1:H}^T$ as the input of encoder block-a,b,c and a set of multi-scale condition variables is captured from it, and the rich fluctuation pattern information embedded is used to reconstruct the denoised result $Y_{1:H}^{T-1}$ in Decoder;
> 2. In the downstream data domain fine-tuning phase, the pre-training weights in Encoder and Decoder are frozen. An additional set of adapter block and zero convolution is introduced to align the multi-domain uniform representation space with the specified domain representation space.
>
> In UTSD-generation, the trainable Adapter is connected to the locked pre-trained weights. The zero convolution with weights initialised to zero are designed to ensure that they grow progressively during training. This architecture ensures that harmful noise is not added to the deep features of large diffusion models at the start of training, and protects the large-scale pre-trained backbone in the trainable replicas from such noise.
>
> To further validate the performance of UTSD on the unconditional generation task, we also follow the design of DiffusionTS and demonstrate the leading performance of the proposed method on two datasets, ETTh1 and Energy. **Specific experimental results are shown in A1**.

---

> ### Author Response · Authors · 2024-11-20
> **To Reviewer YBBT**
>
> >Q4: The paper claims this paper to be a foundation model for TS area. I would like to see the performance of a short forcasting model applied to long context prediction and vice versa.
>
> **A4:** Thank you for your insightful feedback. We appreciate your interest in evaluating the model's adaptability across different forecasting scenarios. We introduce further experiments about ‘short-term models applied to long-term forecasts’ and ‘long-term models applied to short-term forecasts’. Specifically, **ETTm1, 336 $\to$ 96** is defined as ‘Long-term forecasting model applying short-context time series’, and correspondingly **ETTm1, 96 $\to$ 336** is defined as ‘Short-term forecasting model applying long-context time series’. Results, including the largest Traffic dataset, demonstrate that our model demonstrates state-of-the-art performance compared to other methods. Detailed results are presented in the table below.
>
> |Dataset|Metric|UTSD (MSE)|UTSD (MAE)|TimeLLM (MSE)|TimeLLM (MAE)|LLM4TS (MSE)|LLM4TS (MAE)|GPT4TS (MSE)|GPT4TS (MAE)|PatchTST (MSE)|PatchTST (MAE)|TimesNet (MSE)|TimesNet (MAE)
> |:-:|:-:|:-:|:-:|:-:|:-:|:-:|:-:|:-:|:-:|:-:|:-:|:-:|:-:|
> |ETTm1|96 $\to$ 336|**0.326**|**0.381**|0.384|0.411|0.378|0.420|0.421|0.443|0.442|0.455|0.462|0.461|
> |ETTm1|336 $\to$ 96|0.299|**0.333**|**0.272**|0.334|0.285|0.343|0.292|0.346|0.344|0.373|0.338|0.375|
> |ETTm2|96 $\to$ 336|**0.245**|**0.327**|0.295|0.361|0.295|0.356|0.320|0.384|0.342|0.385|0.362|0.398|
> |ETTm2|336 $\to$ 96|0.191|0.284|**0.161**|**0.253**|0.165|0.254|0.173|0.262|0.177|0.260|0.187|0.267|
> |ETTh1|96 $\to$ 336|**0.399**|**0.439**|0.462|0.466|0.453|0.454|0.491|0.491|0.559|0.522|0.551|0.537|
> |ETTh1|336 $\to$ 96|**0.274**|**0.301**|0.362|0.392|0.371|0.394|0.376|0.397|0.404|0.413|0.384|0.402|
> |ETTh2|96 $\to$ 336|**0.312**|**0.382**|0.399|0.444|0.382|0.432|0.426|0.452|0.487|0.495|0.510|0.518|
> |ETTh2|336 $\to$ 96|**0.241**|**0.301**|0.268|0.328|0.262|0.332|0.285|0.342|0.312|0.358|0.340|0.374|
> |Electricity|96 $\to$ 336|**0.153**|**0.258**|0.175|0.267|0.179|0.277|0.190|0.295|0.230|0.325|0.223|0.338|
> |Electricity|336 $\to$ 96|**0.128**|**0.221**|0.131|0.224|0.128|0.223|0.139|0.238|0.186|0.269|0.168|0.272|
> |Traffic|96 $\to$ 336|**0.320**|**0.223**|0.419|0.296|0.441|0.302|0.462|0.334|0.437|0.298|0.709|0.376|
> |Traffic|336 $\to$ 96|**0.284**|**0.203**|0.362|0.248|0.372|0.259|0.388|0.282|0.360|0.249|0.593|0.321|
> |Weather|96 $\to$ 336|**0.216**|**0.266**|0.285|0.299|0.262|0.300|0.292|0.318|0.310|0.334|0.320|0.349|
> |Weather|336 $\to$ 96|**0.133**|**0.195**|0.147|0.201|0.147|0.196|0.162|0.212|0.177|0.218|0.172|0.220|
>
> [1] Yong Liu, Haoran Zhang, Chenyu Li, Xiangdong Huang, Jianmin Wang, and Mingsheng Long. Timer: Generative pre-trained transformers are large time series models. In ICML, 2024c.
>
> [2] Gerald Woo, Chenghao Liu, Akshat Kumar, Caiming Xiong, Silvio Savarese, and Doyen Sahoo. Unified training of universal time series forecasting transformers. In ICML, 2024.
>
> [3] Mononito Goswami, Konrad Szafer, Arjun Choudhry, Yifu Cai, Shuo Li, and Artur Dubrawski. Moment: A family of open time-series foundation models. In ICML, 2024.
>
> >Q5: For Table 3, the performance of ETHh1 to ETHh2 should be needed.
>
> **A5:** As your suggestion, we report the comparative results on ETTh1$\to$ETTh2 in the following Table. Concretely, the average MSE is reduced by **22.7\%**, **32.8\%** and **28.1\%** compared to the existing TimeLLM, GPT4TS and PatchTST. For convenience, the following table shows the comprehensive performance (MSE) of the UTSD and the pre-existing model.
>
> |Dataset|Metric|UTSD (MSE)|UTSD (MAE)|TimeLLM (MSE)|TimeLLM (MAE)|LLMTime (MSE)|LLMTime (MAE)|GPT4TS (MSE)|GPT4TS (MAE)|DLinear (MSE)|DLinear (MAE)|PatchTST (MSE)|PatchTST (MAE)
> |:-:|:-:|:-:|:-:|:-:|:-:|:-:|:-:|:-:|:-:|:-:|:-:|:-:|:-:|
> |ETTh1$\to$ETTh2|96|**0.259**|**0.321**|0.279|0.337|0.510|0.576|0.335|0.374|0.347|0.400|0.304|0.350|
> ||192|**0.295**|**0.349**|0.351|0.374|0.523|0.586|0.412|0.417|0.447|0.460|0.386|0.400|
> ||336|**0.321**|**0.371**|0.388|0.415|0.640|0.637|0.441|0.444|0.515|0.505|0.414|0.428|
> ||720|**0.336**|**0.432**|0.391|0.420|2.296|1.034|0.438|0.452|0.665|0.589|0.419|0.443|
> ||Avg.|**0.303**|**0.368**|0.353|0.387|0.992|0.708|0.406|0.422|0.493|0.488|0.380|0.405|

---

> ### Author Response · Authors · 2024-11-20
> **To Reviewer YBBT**
>
> >Q6: The performances of the other two datasets: Solar Forecasting and Mujoco Imputation should be included since it is widely benchmarked.
>
> **A6:** Thank you for your valuable feedback. We conducted forecasting and imputation experiments on the Solar dataset and MuJoCo dataset as you suggested, respectively, based on the publicly available code provided by DiffusionTS[1], and the detailed experimental results are shown in the following table. To ensure the fairness of the experimental results, the exact same data preprocessing operation and metrics as in DiffusionTS were utilized. Specifically, our approach achieves the best performance on the Solar forecasting task, 80% masking and 90% masking interpolation tasks. Surprisingly, compared with the state-of-the-art DiffusionTS, UTSD achieves an improvement of up to 24.5% on the prediction task, which demonstrates the superior prediction capability of the proposed method.
>
> |Solar Forecasintg|UTSD|Diffusion-TS|SSSD|CSDI|TLAE|TransMAF|GP-copula|
> |:-:|:-:|:-:|:-:|:-:|:-:|:-:|:-:|
> |MSE|**2.83e2±0.51e1**|3.75e2±3.6e1|5.03e2±1.06e1|9.0e2±6.1e1|6.8e2±7.5e1|9.30E+02|9.8e2±5.2e1|
>
> |MuJoCo Imputation|UTSD|Diffusion-TS|SSSD|CSDI|NRTSI|NAOMI|Latent-ODE|NeuralCDE|ODE-RNN|RNN GRU-D|
> |:-:|:-:|:-:|:-:|:-:|:-:|:-:|:-:|:-:|:-:|:-:|
> |70% Mask|0.25(2)|0.37(3)|0.59(8)|**0.24(3)**|0.63|1.46|3|8.35|9.86|11.34|
> |80% Mask|**0.37(2)**|0.43(3)|1.00(5)|0.61(10)|1.22|2.32|2.95|10.71|12.09|14.21|
> |90% Mask|**0.48(5)**|0.73(12)|1.90(3)|4.84(2)|4.06|4.42|3.6|13.52|16.47|19.68|
>
> [1] Xinyu Yuan and Yan Qiao. Diffusion-TS: Interpretable diffusion for general time series generation. In The Twelfth International Conference on Learning Representations, 2024.
>
> Thank you very much for your constructive comments and insightful feedback. Your valuable suggestions are immensely helpful for enhancing the quality of our manuscript. We will incorporate and address all relevant points in the revision.

---

> ### Author Response · Authors · 2024-11-25
> **Further discussion**
>
> Dear Reviewer YBBT,
>
> Thank you very much for devoting time on reviewing our manuscript.  As the author-reviewer discussion process is ending soon, we are wondering whether our responses have well addressed your concerns.  If you have any further questions regarding our manuscript, please let us know and we are glad here to provide further discussion and clarification to improve the quality of this manuscript.
>
> Thank you for your contributions,
>
> Authors of Paper 2173

---

> ### Author Response · Authors · 2024-11-27
> **Official Comment by Authors**
>
> Dear Reviewer YBBT,
>
> Thank you very much for your time and valuable comments on our manuscript. It is heartening to know that a modified version is ready with the help of valuable comments from reviewers, and we would like to know if our modified version has solved your problem. Our improvements are centered around the following suggestions:
>
> 1. **Adding additional evaluation metrics**: To evaluate the quality of time series in the generation task, we refer to the metrics from DiffusionTS and the full experimental results are presented in **Table 12, Page 22** of the revision. Specifically, the generation metrics Context-FID Score, Correlational Score,  Discriminative Score and Predictive Score is reduced by 18.2%, 16.2%, 18.5% and 17.8% on ETTh dataset compared to DiffusionTS. More experimental analysis can be found in **L1167-1170**.
> 2. **Introducing additional baselines and benchmarks**: We adjust the DiffusionTS to time series forecasting task in comparison with Table 1/2 as reviewer suggested, and the results are reported in **A2,A4,A6**.
> 3. **Applying UTSD to unconditional generation**: In uncondi-UTSD, the pre-training weights in Encoder and Decoder are frozen, and an additional set of adapter block and zero convolution is introduced to align the multi-domain uniform representation space with the specified domain representation space. We show the diagram of uncondi-UTSD in the **modified Figure 2**, and uncondi-UTSD shows better unconditional generation performance on the two datasets ETTh and Energy, in **Table 12, Page 22**.
> 4. **Adding additional zero-shot prediction experiments**: As reviewer's suggestion, we report the comparative results on ETTh1 and ETTh2 in the **Table 3,9, Page 8,20**. Concretely, the average MSE is reduced by 22.7%, 32.8% and 28.1% compared to the existing TimeLLM, GPT4TS and PatchTST.
>
> Sincerely thank you for your contribution and responsibility. If you have any further concerns about our manuscript, please let us know and we are happy to provide further discussion here.
>
> Author of paper 2173

---

> ### Author Response · Authors · 2024-11-30
> **Eagerly Await Your Response**
>
> Dear Reviewer YBBT:
>
> Thank you once again for your valuable and constructive review. As the author-reviewer discussion period is coming to a close, we kindly request your feedback on our rebuttal provided above. Please let us know if our responses have sufficiently addressed your concerns. If there are any remaining issues, we would be happy to engage in further discussion, as we believe that an in-depth exchange will help strengthen our paper.
>
> Best regards!
>
> Authors

---

### Official Review · Reviewer_AAQN · 2024-11-03

**Soundness:** 3
**Presentation:** 2
**Contribution:** 2
**Rating:** 6
**Confidence:** 3

**Summary:**

This paper introduces UTSD (Unified Time Series Diffusion), a novel diffusion-based model for multi-domain time series forecasting. The key contributions include:
- A condition-denoising architecture that captures multi-scale temporal patterns
- Direct modeling in actual sequence space rather than latent space
- An improved classifier-free guidance strategy for conditional generation
- A transfer-adapter module for efficient domain adaptation
The model achieves state-of-the-art results on multiple benchmarks in different forecasting scenarios including across-domain training, zero-shot learning, and probabilistic forecasting.

**Strengths:**

This study introduce an innovative condition-denoising architecture along with an actual-space diffusion and denoising process, complemented by a classifier-free guidance strategy for conditional generation. This study represents the first establishment of a foundational time series model leveraging diffusion techniques. The model demonstrates exceptional cross-domain generalization capabilities, indicating its potential to redefine paradigms in time series analysis.And the Unified Time Series Diffusion (UTSD) model achieves state-of-the-art (SOTA) results across various benchmarks.
The originality of this model, coupled with its high-quality outcomes, clarity in methodology, and significant implications for the field, underscores the importance of this findings.

**Weaknesses:**

- There are too few types of evaluation indicators, as a model generated by the diffuison process, more indicators such as those used in the DiffusionTS paper, such as Context-FID Score, Correlational Score, Discriminative Score, Predictive Score, etc.
- Theoretical analysis of why the model works better in actual sequence space vs latent space could be stronger
- Memory and computational complexity analysis is missing
- Limited discussion of model limitations and failure cases
- Some architectural choices (like number of blocks) need better justification
- Comparison with some recent relevant works could be expanded

**Questions:**

1. Can you please add evaluation indicators other than MSE and MAE?
2. As the author said in the paper “the average MSE is reduced by 17.9%, 18.6% and 22.4% compared to the existing TimeLLM, LLM4TS and GPT4TS, which indicates that the proposed method can fully utilize a small amount of data for efficient training.” Could you please explain why a decrease in MSE means that less data can be used for training?
3. In order to train a across-domain model, a mixed dataset needs to be established. However, will the training cost increase after the dataset is enlarged?
4. How does the computational complexity compare to existing approaches?
5. What are the main limitations/failure cases of the model?
6. How sensitive is the model to hyperparameter choices?
7. Can the authors provide theoretical justification for why modeling in actual sequence space is better?
8. How does the model perform on extremely long sequences?

---

> ### Author Response · Authors · 2024-11-20
> **To Reviewer AAQN**
>
> Thank you for your thoughtful review and valuable feedback. We appreciate recognition of our innovative work in classifier-free guidance condition-denoising architecture, actual-space denoising process, and incorporating diffusion models into time series foundation models. We response your concerns as following.
>
> >Q1: Can you please add evaluation indicators other than MSE and MAE?
>
> **A1:** Thank you for your valuable suggestions. To evaluate the quality of generated time series, we refer to the metrices (i.e., *Context-FID, Correlational, Discriminative, and Predictive*) from DiffusionTS and perform the comparison experiments. The results are shown in the following tables. Specifically, the generation metrics Context-FID Score, Correlational Score, Discriminative Score and Predictive Score is reduced by **18.2\%**, **16.2\%**, **18.5\%** and **17.8\%** on ETTh dataset compared to DiffusionTS. Besides, the average improvement of all indicators of UTSD on ETTh and Energy datasets is **17.6\%** and **8.2\%**, respectively.
>
> |Dataset|Metrics (scores)|Length|UTSD|Diffusion-TS|TimeGAN|TimeVAE|Diffwave|DiffTime|Cot-GAN|Improve(%)|
> |:-:|:-:|:-:|:-:|:-:|:-:|:-:|:-:|:-:|:-:|:-:|
> |ETTh1|Context-FID|64|**0.522±.031**|0.631±.058|1.130±.102|0.827±.146|1.543±.153|1.279±.083|3.008±.277|17.2|
> |||128|**0.633±.029**|0.787±.062|1.553±.169|1.062±.134|2.354±.170|2.554±.318|2.639±.427|19.5|
> |||256|**0.347±.010**|0.423±.038|5.872±.208|0.826±.093|2.899±.289|3.524±.830|4.075±.894|17.9|
> |ETTh1|Correlational|64|**0.070±.002**|0.082±.005|0.483±.019|0.067±.006|0.186±.008|0.094±.010|0.271±.007|14.5|
> |||128|**0.072±.002**|0.088±.005|0.188±.006|0.054±.007|0.203±.006|0.113±.012|0.176±.006|18.4|
> |||256|**0.054±.003**|0.064±.007|0.522±.013|0.046±.007|0.199±.003|0.135±.006|0.222±.010|15.8|
> |ETTh1|Discriminative|64|**0.087±.017**|0.106±.048|0.227±.078|0.171±.142|0.254±.074|0.150±.003|0.296±.348|18.1|
> |||128|**0.120±.023**|0.144±.060|0.188±.074|0.154±.087|0.274±.047|0.176±.015|0.451±.080|16.8|
> |||256|**0.048±.011**|0.060±.030|0.442±.056|0.178±.076|0.304±.068|0.243±.005|0.461±.010|20.6|
> |ETTh1|Predictive|64|**0.098±.003**|0.116±.000|0.132±.008|0.118±.004|0.133±.008|0.118±.004|0.135±.003|15.6|
> |||128|**0.087±.003**|0.110±.003|0.153±.014|0.113±.005|0.129±.003|0.120±.008|0.126±.001|20.8|
> |||256|**0.090±.006**|0.109±.013|0.220±.008|0.110±.027|0.132±.001|0.118±.003|0.129±.000|17.1|
> |Energy|Context-FID|64|0.136±.014|**0.135±.017**|1.230±.070|2.662±.087|2.697±.418|0.762±.157|1.824±.144|-0.5|
> |||128|**0.084±.015**|0.087±.019|2.535±.372|3.125±.106|5.552±.528|1.344±.131|1.822±.271|3.6|
> |||256|**0.122±.019**|0.126±.024|5.032±.831|3.768±.998|5.572±.584|4.735±.729|2.533±.467|3.4|
> |Energy|Correlational|64|**0.618±.012**|0.672±.035|3.668±.106|1.653±.208|6.847±.083|1.281±.218|3.319±.062|8.1|
> |||128|**0.426±.031**|0.451±.079|4.790±.116|1.820±.329|6.663±.112|1.376±.201|3.713±.055|5.4|
> |||256|**0.341±.039**|0.361±.092|4.487±.214|1.279±.114|5.690±.102|1.800±.138|3.739±.089|5.6|
> |Energy|Discriminative|64|**0.066±.005**|0.078±.021|0.498±.001|0.499±.000|0.497±.004|0.328±.031|0.499±.001|15.0|
> |||128|**0.127±.038**|0.143±.075|0.499±.001|0.499±.000|0.499±.001|0.396±.024|0.499±.001|11.4|
> |||256|**0.252±.047**|0.290±.123|0.499±.000|0.499±.000|0.499±.000|0.437±.095|0.498±.004|13.2|
> |Energy|Predictive|64|**0.225±.001**|0.249±.000|0.291±.003|0.302±.001|0.252±.001|0.252±.000|0.262±.002|9.6|
> |||128|**0.221±.001**|0.247±.001|0.303±.002|0.318±.000|0.252±.000|0.251±.000|0.269±.002|10.7|
> |||256|**0.214±.001**|0.245±.001|0.351±.004|0.353±.003|0.251±.000|0.251±.000|0.275±.004|12.8|
>
> Detailed experimental results are shown in Table 12, page 22 of our revised manuscript.
>
> >Q2: Explain the confused statement about why a decrease in MSE means that less data can be used for training.
>
> **A2:** Thank you for your careful review, and we apologize for any confusion the original presentation may have caused. As we have discussed in the related work, TimeLLM, LLM4TS and GPT4TS are based on pre-trained large language models that are fine-tuned for application in the time series domain, and thusthese models use a huge corpus (more than **15,000,000 million** timesteps) in pre-train. In contrast, UTSD is only pre-trained on a mixed dataset (only **27.5 million** timesteps) obtained by merging 7 small datasets, while the experimental results on the right hand side of Table 2 are obtained by pre-training UTSD from scratch on a specific single small dataset (**0.12~1.52 million** timesteps). Compared to TimeLLM, which uses a huge corpus pre-training and then aligns to the temporal space, UTSD shows excellent data utilisation and can learn sufficiently data patterns from a small number of corpus for inference on downstream datasets, achieving better results on the prediction task. We will refine the expression in the revised manuscript to avoid situations that might confuse readers.

---

> ### Author Response · Authors · 2024-11-20
> **To Reviewer AAQN**
>
> >Q3: In order to train a across-domain model, a mixed dataset needs to be established. However, will the training cost increase after the dataset is enlarged?
>
> **A3:** Thank you for your insightful comments. Indeed, all data-driven models, whether they are traditional deep models or recent predictive models based on diffusion theory, are challenged with large computational overheads and training costs as the size of the dataset becomes larger. However, this is a necessary cost for all pretrain fine-tune paradigms, including Large Language Models (GPT, Llama) and Large Visual Models (SD1.5, SDXL).
>
> An encouraging news is that UTSD has made great efforts to reduce the time overhead during training and inference. Firstly, the pretrain phase occurs entirely offline, and after cross-dataset pretrained, the fine-tuned model weights can quickly converge to the desired state. In addition, UTSD's hybrid architecture supports naturally efficient fine-tuning through the ‘plug-and-play’ Adapter. Specifically, pre-training to obtain a Condition-Denoising Net with a large number of weights is completely frozen, and only a small number of weights in the Adapter component need to be optimised. The effectiveness of the fine-tuning strategy can be intuitively explained by the fact that the pre-trained Condition Net is responsible for capturing generic fluctuation patterns from observed sequences as conditional information, the Denoising Net is required to reconstruct sequence samples from noise in the target domain based on specific fluctuation patterns, and the fine-tuning-enabled Adapter is used to connect the unified representation space with the proprietary representation space.
>
> Moreover, compared to other methods (TimeLLM, LLMTime, GPT4TS), our pretrain corpus is smaller (as in the previous A1), which further illustrates the efficiency and lost cost of our UTSD.
>
> >Q4: How does the computational complexity compare to existing approaches?
>
> **A4:** Thanks to the reviewer for raising an issue that is critical to a realistic scenario. A detailed discussion of the computational complexity of UTSD is as follows:
>
> The computational complexity of UTSD is $O((\frac{(L^2+H^2)}{P_d}+k\cdot P_d\cdot(H+L))\cdot T)$, where $L$ and $H$ denote the length of the input and output sequences, $P_d$ is the dimension of the patch-wise tokenizer, $K$ denotes the depth of the encoder-decoder, and $T$ denotes the number of rounds of iterative denoising operation, respectively.
>
> To address scalability concerns, we conducted experiments on runtime and memory usage (ETTh1 dataset, lookback window=336, forecast window=96). Results, including the large foundation time series model and diffusion model, demonstrate that our model requires less time and memory compared to other approaches. Detailed results are presented in the table below.
>
> UTSD utilizes the de-Markovised DDIM algorithm, which means that UTSD only requires a smaller number of iterations, and thus the computational complexity of UTSD in the training and inference phases is far lower than that of the traditional temporal diffusion model. In addition, UTSD does not require the high-quality autoencoder, which greatly reduces the overall training overhead of UTSD.
>
> |Metrics(ETTh1-96)|Total Param. (M)|Mem. (MiB)|Speed (s/iter)|
> |:-:|:-:|:-:|:-:|
> |UniTime|439.52|2074|0.335|
> |TimeLLM|3623.71|4537|**0.184**|
> |UTSD|**313.79**|**1809**|0.270|
>
> |Metrics(ETTh1-96)|Total Param. (M)|Mem. (MiB)|Speed (s/iter)|
> |:-:|:-:|:-:|:-:|
> |CSDI|**54.29**|**983**|0.245|
> |DiffusionTS|258.64|1539|0.119|
> |UTSD|313.79|1809|**0.270**|

---

> ### Author Response · Authors · 2024-11-20
> **To Reviewer AAQN**
>
> >Q5: What are the main limitations/failure cases of the model?
>
> **A5:** We appreciate you raising this important question about UTSD's limitations. Our experiments revealed several key areas where the model faces challenges:
>
> **High arithmetic overhead and time cost.** As mentioned earlier, compared to traditional deep models, diffusion models have the defect of unstable and slow convergence during training and require multiple rounds of iterative denoising processes during inference. These drawbacks lead to the fact that diffusion-based time-series prediction models require high arithmetic overhead and time cost, both in the training and inference phases. However, UTSD exhibits lower computational overheads compared to existing time series foundation models, which due to three innovative designs that, patch-wise attention mechanism, iterative denoising in real sequence space and de-Markovised DDIM algorithm. Detailed discussion can be found in Appendix A4.
>
> **Challenges applied to discriminantive-based tasks.** Diffusion models excel in generative-based paradigms, such as forecasting, imputation, and anomaly detection. However, compared to general-purpose deep models like TimesNet and Moment, they do not show significant advantages in discriminative tasks like classification. We anticipate that the pre-trained UTSD should not only generate high-quality prediction samples but also possess strong representational learning and discriminative capabilities. This is crucial for developing a "multi-task generalized spatiotemporal diffusion model," which will be the focus of our future work.
>
> >Q6: How sensitive is the model to hyperparameter choices?
>
> **A6:** In the UTSD architecture, all components are composed four minimal blocks (encoder block, decoder block, middle block and adapter block). The basic construction of each block is illustrated in Figure 2, however, in the concrete implementation we allow consecutive **L-layer** block residues to be stacked together to form stacked-block and used to form the Condition Net, Denoising Net and Adapter. Furthermore, the number of input and output channels of the middle block is denoted as **D**, and accordingly the number of output channels in the three pairs of encoder-decoder blocks are $\frac D4$, $\frac D2$, and $D$, respectively. In order to verify the sensitivity of UTSD to hyperparameter selection, the following table shows the performance of UTSD with different parameter scales on multiple benchmarks. Where the model parameter combinations include **(L,D) = (2,128), (2,256), (3,128), (3,256), (4,128), (4, 256)**, with a fixed forecasting window of 96, and a performance metric of MSE.
>
> By default, the hyperparameters of UTSD are fixed to **L=3** and **D=256**, and all experimental results presented follow this setting. In the scratch forecasting task, the number of output channels in the three pairs of Encoder-Decoder blocks is 64,128,256, respectively, and number of input and output channels of the Middle Block is fixed to 256.
>
> |Hyperparameter|ETTh1|ETTh2|ETTm1|ETTm2|Weather|ECL|Traffic|
> |-|-|-|-|-|-|-|-|
> |(L=2,D=128)|0.352|0.307|0.335|0.244|0.215|0.153|0.311|
> |(L=2,D=256)|0.344|0.302|0.319|0.254|0.211|0.158|0.325|
> |(L=3,D=128)|0.328|**0.274**|0.321|0.239|0.203|0.154|**0.296**|
> |(L=3,D=256)|0.334|0.285|0.308|**0.233**|0.202|0.149|0.301|
> |(L=4,D=128)|**0.322**|0.280|**0.302**|0.235|**0.198**|**0.145**|0.298|
> |(L=4,D=256)|0.340|0.313|0.319|0.234|0.214|0.160|0.306|

---

> ### Author Response · Authors · 2024-11-20
> **To Reviewer AAQN**
>
> >Q7: Can the authors provide theoretical justification for why modeling in actual sequence space is better?
>
> **A7:** The recent Q-DM [1] confirms our view and proposes a develop a Timestep-aware Quantization (TaQ) method to effectively eliminate the accumulated quantization error caused by the multi-step denoising process.
>
> To further validate our view, we designed a control experiment with the Latent-UTSD model for iterative denoising in the hidden representation space, which ‘trains the self-encoder in an end-to-end manner using a mixed multidomain synthetic dataset, and subsequently reconstructs the embeddings of the target sequences directly in the hidden space’. The experimental results in the following table show that Latent-UTSD exhibits relatively poor performance, which validates our point from an experimental perspective. For implementation details, refer to the appendix section Section C. For convenience, we show the comprehensive performance of Latent-UTSD on the four datasets in the following table.
>
> |Mdoels|Metrics|ETTh1(96)|ETTh1(192)|ETTh1(336)|ETTh1(720)|ETTh1(Avg)|ETTm1(96)|ETTm1(192)|ETTm1(336)|ETTm1(720)|ETTm1(Avg)|
> |:-:|:-:|:-:|:-:|:-:|:-:|:-:|:-:|:-:|:-:|:-:|:-:|
> |UTSD|MSE|**0.274**|**0.290**|**0.383**|**0.387**|**0.334**|**0.299**|**0.304**|**0.312**|**0.317**|**0.308**|
> |Latent-UTSD|MSE|0.323|0.342|0.450|0.454|0.392|0.352|0.365|0.365|0.370|0.363|
> |UTSD|MAE|**0.301**|**0.339**|**0.424**|**0.428**|**0.383**|**0.333**|**0.358**|**0.365**|**0.368**|**0.356**|
> |Latent-UTSD|MAE|0.348|0.386|0.493|0.492|0.430|0.390|0.418|0.435|0.443|0.422|
>
> |Mdoels|Metrics|Traffic(96)|Traffic(192)|Traffic(336)|Traffic(720)|Traffic(Avg)|Weather(96)|Weather(192)|Weather(336)|Weather(720)|Weather(Avg)|
> |:-:|:-:|:-:|:-:|:-:|:-:|:-:|:-:|:-:|:-:|:-:|:-:|
> |UTSD|MSE|**0.284**|**0.293**|**0.308**|**0.319**|**0.301**|**0.133**|**0.184**|**0.207**|**0.264**|**0.202**|
> |Latent-UTSD|MSE|0.314|0.316|0.341|0.343|0.329|0.154|0.211|0.238|0.313|0.229|
> |UTSD|MAE|**0.203**|**0.211**|**0.215**|**0.223**|**0.213**|**0.195**|**0.237**|**0.258**|**0.313**|**0.246**|
> |Latent-UTSD|MAE|0.223|0.236|0.242|0.242|0.235|0.226|0.271|0.297|0.359|0.288|
>
> This is due to differences in the focus of the inductive bias to capture patterns between the two stages of training, e.g., the encoder loses some of the information critical to the reconstruction process during the projection of the original sequence space into the hidden representation space. Subsequently, in the iterative denoising process the model targets ‘incomplete hidden representations’ for reconstruction and therefore the noise reduction model is unable to capture all the fluctuating pattern distributions in each round of reconstruction, which results in a natural accumulation of errors in the latent spatial inference.
>
> [1] Li, Yanjing et al. “Q-DM: An Efficient Low-bit Quantized Diffusion Model.” Neural Information Processing Systems (2023).
>
> >Q8: How does the model perform on extremely long sequences?
>
> **A8:** Thank you for your valuable feedback. We introduce more visualisation results in the **revision** appendix section of the revised version, including the ‘extremely long term sequences’ as you suggested. **Figure 10 (Page. 24 in Revision)** shows real extremely long multi-periodic series sampled from the ECL and Traffic datasets, and in addition, **Figure 11 (Page. 25 in Revision)** shows real short period non-periodic sequences sampled from the ETT dataset. Specifically, UTSD also shows good performance in extremely long sequence prediction, which verifies that UTSD has the ability to capture extremely long term dependencies, which is crucial for practical applications. In addition, UTSD also shows satisfactory prediction results in short-term non-periodic sequences.

---

> ### Author Response · Authors · 2024-11-25
> **Further discussion**
>
> Dear Reviewer AAQN,
>
> Thank you very much for devoting time on reviewing our manuscript.  As the author-reviewer discussion process is ending soon, we are wondering whether our responses have well addressed your concerns.  If you have any further questions regarding our manuscript, please let us know and we are glad here to provide further discussion and clarification to improve the quality of this manuscript.
>
> Thank you for your contributions,
>
> Authors of Paper 2173

---

> ### Author Response · Authors · 2024-11-27
> **Official Comment by Authors**
>
> Dear Reviewer AAQN,
>
> Thank you very much for your time and valuable comments on our manuscript. It is heartening to know that a modified version is ready with the help of valuable comments from reviewers, and we would like to know if our modified version has solved your problem. Our improvements are centered around the following suggestions:
>
> 1. **Adding additional evaluation metrics**: To evaluate the quality of time series in the generation task, we refer to the metrics from DiffusionTS and the full experimental results are presented in **Table 12, Page 22** of the revision. Specifically, the generation metrics Context-FID Score, Correlational Score,  Discriminative Score and Predictive Score is reduced by 18.2%, 16.2%, 18.5% and 17.8% on ETTh dataset compared to DiffusionTS. More experimental analysis can be found in **L1167-1170**.
> 2. **Data efficiency confusion**: We rethought our description in revised **L366-368** to avoid confusion for the reader.
> 3. **Complete complexity analysis**: We provide a theoretical analysis of the computational complexity of UTSD and show the time cost and memory footprint of UTSD in **A4**. Compared with the advanced UniTime, TimeLLM, Diffusion, UTSD has obvious advantages. The full complexity analysis will be further included into the revised manuscript.
> 4. **Sensitivity analysis of model hyperparameters**: We demonstrate the model performance for different groups of residual stacking layers **L** and model dimension **D**. The results can be found in revision **Table8, Page 19**.
> 5. **More ablation experiments**: To validate our modeling strategy in real sequence space, the Latent-UTSD model is designed to iteratively reduce noise in the hidden representation space. Detailed ablation experiments are presented in revision **Table 6, Page 16**.
> 6. **More visualization experiments**: We introduce more visualisation results in the revision appendix section (**Figure 10,11 Page 24,25**), including the **extremely long term sequences** as reviewer suggested.
>
> Sincerely thank you for your contribution and responsibility. If you have any further concerns about our manuscript, please let us know and we are happy to provide further discussion here.
>
> Author of paper 2173

---

> ### Author Response · Authors · 2024-11-30
> **Eagerly Await Your Response**
>
> Dear Reviewer AAQN:
>
> Thank you once again for your valuable and constructive review. As the author-reviewer discussion period is coming to a close, we kindly request your feedback on our rebuttal provided above. Please let us know if our responses have sufficiently addressed your concerns. If there are any remaining issues, we would be happy to engage in further discussion, as we believe that an in-depth exchange will help strengthen our paper.
>
> Best regards!
>
> Authors

---

### Official Review · Reviewer_jtdw · 2024-11-04

**Soundness:** 2
**Presentation:** 2
**Contribution:** 1
**Rating:** 5
**Confidence:** 2

**Summary:**

The paper proposes a Unified Time Series Diffusion (UTSD) model for time series tasks (mainly focusing on forecasting tasks). The model utilizes a diffusion-based approach rather than traditional autoregressive methods. In particular, the conditional network, adaptor fine-tuning strategy are considered. Numerical experiments and sample codes are provided.

**Strengths:**

1. This paper considers an interesting research topic which trying to apply the diffusion model into time-series analysis domain.

2. Various experimental tasks are considered and cover diverse domains and performance comparisons against both foundation models and domain-specific models.

**Weaknesses:**

1. My primary concern lies in the suitability of employing a non-autoregressive diffusion model for forecasting tasks. Forecasting is typically a regression problem rather than a generative one, and autoregressive behavior is often observed in certain domains, such as climate and finance. Given this, the current draft would benefit from further justification regarding the choice of a non-autoregressive diffusion model in this context.

2. The architecture of the proposed model requires further discussion. Section 3 describes the proposed model as a combination of convolution and attention blocks. However, existing literature shows that models based solely on either convolutional or attention networks are already effective for forecasting tasks. Could the authors clarify why a hybrid structure is necessary here? Specifically, how does this configuration address any limitations of diffusion models in forecasting?

Minor Issue: In Appendix E.2, the benchmarks also include imputation and classification tasks, but these seem to be missing from the main draft.

**Questions:**

Please see the weakness section.

At this stage, the novelty of the work is insufficient for acceptance at a top-tier machine learning conference, so I tend to recommend rejection. However, I am open to reconsidering my evaluation based on further discussion.

---

> ### Author Response · Authors · 2024-11-20
> **To Reviewer jtdw**
>
> Thank you for your thoughtful review and valuable feedback. We response your concerns as following.
>
> >Q1: My primary concern lies in the suitability of employing a non-autoregressive diffusion model for forecasting tasks. Forecasting is typically a regression problem rather than a generative one, and autoregressive behavior is often observed in certain domains, such as climate and finance. Given this, the current draft would benefit from further justification regarding the choice of a non-autoregressive diffusion model in this context.
>
> **A1:** Thank you for your insightful feedback. We adapt the non-autoregressive diffusion model to handle substantial distribution shifts across different domains. Traditional autoregressive models, while effective at capturing temporal dependencies within individual domains, often face cumulative error challenges in long-term forecasting and struggle with cross-domain generalization due to their reliance on domain-specific conditional probabilities derived from historical patterns.
>
> Notably, several well-known time series analysis models, such as Informer and PatchTST, have successfully employed non-autoregressive architectures, achieving remarkable predictive performance. These models have become widely recognized baselines in the field, further validating the effectiveness of non-autoregressive approaches for time series forecasting. Our extensive experiments demonstrate that the UTSD model outperforms existing autoregressive models on several benchmarks, showcasing its effectiveness in both zero-shot and fine-tuned settings. This indicates the suitability of our approach for forecasting tasks that involve complex, multi-domain data.
>
> We will incorporate a more detailed justification and discussion of these points in the revised manuscript to clarify the rationale behind our model choice. Thank you again for your valuable feedback.
>
> >Q2: The architecture of the proposed model requires further discussion. Section 3 describes the proposed model as a combination of convolution and attention blocks. However, existing literature shows that models based solely on either convolutional or attention networks are already effective for forecasting tasks. Could the authors clarify why a hybrid structure is necessary here? Specifically, how does this configuration address any limitations of diffusion models in forecasting?
>
> **A2:** Thank you for your insighful comments. In our UTSD, convolutional layers are highly effective for capturing local temporal patterns and reducing noise through feature extraction. By integrating convolutional blocks, our model can efficiently process local dependencies and enhance the robustness of the representation. On the other hand, attention mechanisms excel in capturing long-range dependencies and providing dynamic context awareness. By incorporating attention blocks, our model can adaptively focus on relevant parts of the time series, which is crucial for accurate cross-domain forecasting. The diffusion model's denoising process benefits from the hybrid structure by gaining a comprehensive understanding of both local and global patterns. This enhances the model's ability to generate accurate predictions across diverse domains, where patterns and dependencies can vary significantly. Traditional diffusion models may struggle with capturing complex temporal dependencies, especially in varied domains. Our hybrid approach ensures that both local fluctuations and broader trends are effectively modeled, improving the diffusion process's stability and accuracy in generating predictions.

---

> ### Author Response · Authors · 2024-11-20
> **To Reviewer jtdw**
>
> >Q3: Minor Issue: In Appendix E.2, the benchmarks also include imputation and classification tasks, but these seem to be missing from the main draft.
>
> **A3:** Thank you for pointing out this oversight. The UTSD model is designed to be a versatile tool for various time-series tasks, following the generative-based pretrain paradigm. By learning a unified representation that captures both local and global patterns, UTSD effectively generalizes across different domains. Its ability to incorporate multi-scale conditional contexts allows it to adapt dynamically to task-specific demands. The adapter mechanism facilitates seamless fine-tuning, enabling efficient adaptation to various downstream tasks with minimal adjustments. These features position UTSD as a comprehensive model capable of addressing diverse time-series challenges.
>
> We conduct additional experiments on imputation tasks to assess UTSD's performance. The results, shown in the table below, indicate that the average MSE is reduced by **17.0%**, **20.1%**, and **38.7%** compared to GPT4TS, TimesNet, and PatchTST, respectively. This improvement demonstrates that the diffusion model, with its reconstruction paradigm, is well-suited for imputation tasks. Notably, UTSD performs exceptionally well on the ECL dataset, characterized by multi-periodic patterns, aligning with the multi-scale representation mechanism of our Condition-Denoising component.
>
> |Datasets|Mask Ratios|UTSD (MSE)|UTSD (MAE)|TimeLLM (MSE)|TimeLLM (MAE)|GPT4TS (MSE)|GPT4TS (MAE)|TimesNet (MSE)|TimesNet (MAE)|LLMTime (MSE)|LLMTime (MAE)|
> |:-:|:-:|:-:|:-:|:-:|:-:|:-:|:-:|:-:|:-:|:-:|:-:|
> |ETTm1|12.5%|0.019|**0.077**|**0.017**|0.085|0.023|0.101|0.041|0.130|0.096|0.229|
> |     |25.0%|0.022|**0.092**|**0.022**|0.096|0.025|0.104|0.047|0.139|0.100|0.234|
> |     |37.5%|**0.028**|**0.110**|0.029|0.111|0.029|0.111|0.049|0.143|0.133|0.271|
> |     |50.0%|**0.035**|**0.117**|0.040|0.128|0.036|0.124|0.055|0.151|0.186|0.323|
> |     |Avg. |**0.026**|**0.099**|0.028|0.105|0.027|0.107|0.047|0.140|0.120|0.253|
> |ETTm2|12.5%|0.018|0.079|**0.017**|**0.076**|0.018|0.080|0.026|0.094|0.108|0.239|
> |     |25.0%|**0.019**|0.082|0.020|**0.080**|0.020|0.085|0.028|0.099|0.164|0.294|
> |     |37.5%|**0.021**|**0.085**|0.022|0.087|0.023|0.091|0.030|0.104|0.237|0.356|
> |     |50.0%|**0.024**|**0.094**|0.025|0.095|0.026|0.098|0.034|0.110|0.323|0.421|
> |     |Avg. |**0.020**|0.085|0.021|**0.084**|0.022|0.088|0.029|0.102|0.208|0.327|
> |ETTh1|12.5%|**0.040**|**0.137**|0.043|0.140|0.057|0.159|0.093|0.201|0.126|0.263|
> |     |25.0%|**0.053**|**0.155**|0.054|0.156|0.069|0.178|0.107|0.217|0.169|0.304|
> |     |37.5%|**0.070**|**0.175**|0.072|0.180|0.084|0.196|0.120|0.230|0.220|0.347|
> |     |50.0%|**0.093**|**0.202**|0.107|0.216|0.102|0.215|0.141|0.248|0.293|0.402|
> |     |Avg. |**0.064**|**0.167**|0.069|0.173|0.078|0.187|0.115|0.224|0.202|0.329|
> |ETTh2|12.5%|0.040|**0.124**|**0.039**|0.125|0.040|0.130|0.057|0.152|0.187|0.319|
> |     |25.0%|**0.043**|**0.131**|0.044|0.135|0.046|0.141|0.061|0.158|0.279|0.390|
> |     |37.5%|**0.049**|**0.143**|0.051|0.147|0.052|0.151|0.067|0.166|0.400|0.465|
> |     |50.0%|**0.053**|**0.155**|0.059|0.158|0.060|0.162|0.073|0.174|0.602|0.572|
> |     |Avg. |**0.047**|**0.138**|0.048|0.141|0.049|0.146|0.065|0.163|0.367|0.436|
> |Electricity|12.5%|**0.043**|**0.129**|0.080|0.194|0.085|0.202|0.055|0.160|0.196|0.321|
> |     |25.0%|**0.049**|**0.142**|0.087|0.203|0.089|0.206|0.065|0.175|0.207|0.332|
> |     |37.5%|**0.056**|**0.151**|0.094|0.211|0.094|0.213|0.076|0.189|0.219|0.344|
> |     |50.0%|**0.065**|**0.165**|0.101|0.220|0.100|0.221|0.091|0.208|0.235|0.357|
> |     |Avg. |**0.053**|**0.147**|0.090|0.207|0.092|0.210|0.072|0.183|0.214|0.339|
> |Weather|12.5%|**0.024**|**0.040**|0.026|0.049|0.025|0.045|0.029|0.049|0.057|0.141|
> |     |25.0%|**0.026**|**0.043**|0.028|0.052|0.029|0.052|0.031|0.053|0.065|0.155|
> |     |37.5%|**0.030**|**0.047**|0.033|0.060|0.031|0.058|0.081|0.180|0.058|0.121|
> |     |50.0%|**0.033**|**0.052**|0.037|0.065|0.034|0.062|0.038|0.063|0.102|0.207|
> |     |Avg. |**0.028**|**0.046**|0.031|0.056|0.030|0.054|0.060|0.144|0.076|0.171|
>
> We have supplemented the modified version with experiments on imputation task, the detailed results of which are presented in Table 11, page. 21, in addition to further analyses of the experimental results (L1076-1137).

---

> ### Author Response · Authors · 2024-11-25
> **Further discussion**
>
> Dear Reviewer jtdw,
>
> Thank you very much for devoting time on reviewing our manuscript.  As the author-reviewer discussion process is ending soon, we are wondering whether our responses have well addressed your concerns.  If you have any further questions regarding our manuscript, please let us know and we are glad here to provide further discussion and clarification to improve the quality of this manuscript.
>
> Thank you for your contributions,
>
> Authors of Paper 2173

---

> ### Author Response · Authors · 2024-11-27
> **Official Comment by Authors**
>
> Dear Reviewer jtdw,
>
> Thank you very much for your time and valuable comments on our manuscript. It is heartening to know that a modified version is ready with the help of valuable comments from reviewers, and we would like to know if our modified version has solved your problem. Our improvements are centered around the following suggestions:
>
> 1. **Applicability of non-autoregressive diffusion models**: We have added more about the pitfalls of autoregressive mechanisms in cross-domain generalization scenarios and zero-shot tasks in **L86-93** of the revised manuscript, which will help the reader understand UTSD's motivation for establishing unified diffusion models in the time-series domain.
> 2. **Hybrid architecture addresses the limitations of diffusion models in prediction**: We have added detailed design about the innovative 1 $\times$ 1 Conv1D and attention mechanism of Adapter in revision **L332-337**. The former is utilized to align the number of tokens in the observation space and the prediction space to achieve **flexible input and output in short time**, and the latter is utilized to capture the dependencies between all tokens, establishing a connection between the context learning stage and the noise reduction reconstruction stage.
> 3. **Supplementary imputation task**: The UTSD model is intended to be a versatile tool for various time-series tasks, so we added the imputation experiments on all six benchmarks. Detailed experimental results are presented in **Table 11, Page 21** and the experimental analyses were in the revision **L1076-1137**
>
> Sincerely thank you for your contribution and responsibility. If you have any further concerns about our manuscript, please let us know and we are happy to provide further discussion here.
>
> Author of paper 2173

---

### Author Response · Authors · 2024-11-22
**Summary of Revisions**

We sincerely thank all the reviewers for their insightful reviews and valuable comments, which are instructive for improving our paper further.

In this paper, a **Unified Time Series Diffusion (UTSD)** model is established **for the first time to model the multi-domain probability distribution**, utilizing the powerful probability distribution modeling ability of Diffusion. Unlike existing models that capture the conditional probabilities of the prediction horizon to the historical sequence, we use a diffusion denoising process to model the mixture distribution of the cross-domain data and generate the prediction sequence for the target domain directly utilizing conditional sampling. The proposed UTSD contains several pivotal designs: The **condition network** captures the multi-scale fluctuation patterns from the observation sequence, which are utilized as context representations; The **adaptor-based fine-tuning strategy** is utilized for downstream tasks in target domains before multi-domain joint pretraining, and supports flexible observation and prediction horizons through token-level alignment. We conduct extensive experiments on mainstream benchmarks, and UTSD outperforms existing foundation models on all data domains, exhibiting superior zero-shot generalization ability. Empirical results validate the potential of UTSD as a time series foundational model.

The reviewers generally expressed positive opinions of our paper, noting that the method proposed "**an interesting research topic,**" coupled with "**its high-quality outcomes, clarity in methodology, innovative architecture,**" demonstrates "**significant implications for the field,**" and achieves "**state-of-the-art performance.**" Additionally, they highlighted its "**potential to serve as the time-series unified diffusion model.**"

The reviewers also raised insightful and constructive concerns. We have made every effort to address all the concerns by providing sufficient evidence and requested results. Here is the summary of the major revisions:

1. **Employing non-autoregressive diffusion model for forecasting (Reviewers jtdw, eVfc)**: Diffusion models exhibit good cross-domain generalisation by modelling integrated conditional probability distributions across multiple domains, rather than temporal dependencies within individual domains. Extensive experiments indicate the suitability of our approach for forecasting tasks that involve complex, multi-domain data.
2. **Add multi-tasking experiment results (Reviewers jtdw, YBBT)**: The UTSD model is designed to be a versatile tool for various time-series tasks, following the generative-based pretrain paradigm. We conduct additional experiments on imputation and unconditional cross-domain generation tasks to assess UTSD's performance.
3. **Add more evaluation metrics and baseline models (Reviewers AAQN, YBBT)**: To evaluate the quality of generated time series, we refer to the metrices from DiffusionTS and perform the comparison experiments.
4. **Compare our method and previous Diffusion4TS models (Reviewers YBBT, eVfc)**: We provide a comprehensive analysis of the differences of our method and other Diffusion4TS models, focusing on cross-domain generalization, efficient fine-tuning paradigm, flexible timesteps of input and output.
5. **More visualisation results for challenging time series forecasting (Reviewers AAQN, pg2F)**: We introduce more visualisation results in the revision appendix section of the revised version. UTSD also shows good performance in extremely long sequence prediction, which verifies that UTSD has the ability to capture extremely long term dependencies, which is crucial for practical applications. In addition, UTSD also shows satisfactory prediction results in short-term non-periodic sequences.

The valuable suggestions from reviewers are very helpful for us to improve our paper. **All the changes I've made in our revision (highlighted in blue)**. We'd be very happy to answer any further questions.

Looking forward to the reviewer's feedback.

---

### Comment · Area_Chair_jgVu · 2024-11-26
**Encouragement to Actively Participate in the Discussion Phase**

Dear Reviewers,

Thank you for your valuable contributions to the review process so far. As we enter the discussion phase, I encourage you to actively engage with the authors and your fellow reviewers. This is a critical opportunity to clarify any open questions, address potential misunderstandings, and ensure that all perspectives are thoroughly considered.

Your thoughtful input during this stage is greatly appreciated and is essential for maintaining the rigor and fairness of the review process.

Thank you for your efforts and dedication.

---

### Meta-Review · Area_Chair_jgVu · 2024-12-19

**Metareview:**

(a) Summary of Scientific Claims and Findings
This paper introduces the Unified Time Series Diffusion (UTSD) model for multi-domain time series tasks. UTSD leverages a diffusion denoising process to model joint probability distributions across diverse domains, bypassing the autoregressive approach. The paper proposes three key contributions:
A condition network to capture multi-scale patterns from observation sequences as context for denoising.
An adapter-based fine-tuning strategy to enhance domain adaptability while reusing universal representations learned in pretraining.
A diffusion process performed in the actual sequence space with classifier-free guidance to improve stability and performance.
Extensive experiments demonstrate that UTSD outperforms baseline models in forecasting, imputation, and other tasks, with claims of superior zero-shot and cross-domain generalization.

(b) Strengths of the Paper
Novelty: UTSD introduces a novel approach by integrating diffusion models into time series foundational models and emphasizing sequence-space denoising.
Extensive Experiments: The authors present results across multiple benchmarks and tasks, showcasing improved performance over existing baselines.
Practical Contributions: The adapter-based fine-tuning mechanism provides computational efficiency during domain-specific adaptation.
Visualization and Analysis: The paper includes visualizations and ablation studies to support claims about the model’s effectiveness in long- and short-term dependencies.

(c) Weaknesses of the Paper
Theoretical Insufficiency: The paper lacks rigorous theoretical justification for the advantages of sequence-space modeling over latent-space alternatives. While some empirical evidence is provided, the theoretical foundation remains weak.
Limited Cross-domain Generalization: The model's cross-domain performance is restricted to datasets with similar characteristics, reducing its applicability in more diverse, real-world scenarios.
Inadequate Comparisons: While comparisons to certain baselines (e.g., DiffusionTS) are included, the absence of benchmarks with other state-of-the-art models like MOIRAI or UniTS undermines the claim of being a comprehensive foundational model.
Evaluation Metrics: The initial reliance on MSE/MAE as the primary evaluation metrics is problematic for generative models. Although additional metrics were added during rebuttal, this should have been included from the outset.
Computational Overhead: While UTSD optimizes diffusion processes through design choices like de-Markovization, diffusion models inherently require higher computational resources, especially for long sequences and multiple iterative denoising rounds.
Presentation and Clarity: Some sections, particularly on the hybrid architecture and diffusion process, lack sufficient explanation or empirical backing, leading to questions about the necessity and effectiveness of the design.

(d) Reasons for Rejection
Scope and Applicability: The UTSD model is innovative but lacks generalizability to a broad range of time series tasks. Its reliance on diffusion models limits its practical impact.
Theoretical Gaps: The paper does not provide sufficient theoretical insights into the proposed approach, particularly regarding the choice of sequence-space modeling and hybrid architectures.
Benchmarking Limitations: Missing comparisons with key state-of-the-art models and insufficient diversity in datasets weaken the empirical claims.
Reviewer Concerns: Despite an active rebuttal period, several critical concerns raised by reviewers—including cross-domain generalization, theoretical contributions, and computational efficiency—remain inadequately addressed.

**Additional Comments On Reviewer Discussion:**

1. Theoretical Justification of Sequence-space Diffusion
Concern: Reviewers questioned the theoretical basis for performing diffusion in the actual sequence space rather than latent space, as well as the claimed advantages of this approach.
Author Response: The authors cited experimental comparisons showing that sequence-space modeling outperformed latent-space approaches (e.g., Latent-UTSD). However, the theoretical explanation was limited to a brief discussion of information preservation during denoising.
Evaluation: While the experimental evidence partially supported the authors’ claims, the lack of deeper theoretical insights left this concern unresolved.

2. Cross-domain Generalization
Concern: Reviewers noted that the cross-domain generalization of UTSD was constrained to datasets with similar patterns, which limits the model’s applicability to diverse real-world domains.
Author Response: The authors acknowledged this limitation and proposed extensions in future work but did not offer new experiments or solutions in the rebuttal.
Evaluation: The acknowledgment was appreciated but insufficient to address the core issue, which undermines the paper’s claims of being a universal model.

3. Benchmark Comparisons and Metrics
Concern: The reviewers criticized the lack of comprehensive comparisons with state-of-the-art models (e.g., MOIRAI, UniTS) and the initial reliance on MSE/MAE, which are insufficient for evaluating generative tasks.
Author Response: The authors added additional metrics (e.g., Context-FID, Discriminative Score) and conducted comparisons with DiffusionTS. They argued that some missing comparisons were due to unavailable code or datasets.
Evaluation: While the new metrics strengthened the empirical evaluation, the lack of broader comparisons remained a significant drawback.

4. Hybrid Architecture Justification
Concern: The necessity of combining convolutional and attention-based layers was questioned, as standalone architectures have demonstrated effectiveness in prior work.
Author Response: The authors explained that convolutional layers capture local dependencies while attention layers handle long-range patterns, enhancing the denoising process. However, they did not provide ablation studies to quantify the specific contributions of each component.
Evaluation: The explanation provided was plausible but lacked empirical support, reducing confidence in the hybrid design’s necessity.
Computational Complexity and Scalability

Despite the authors’ efforts during rebuttal, the paper requires further theoretical and empirical refinement to address these concerns adequately. For these reasons, the paper was recommended for rejection.

---

### Decision · Program_Chairs · 2025-01-22

Reject